# From primordial clocks to circadian oscillators

Warintra Pitsawong[1,4,9], Ricardo A. P. Pádua[1,9], Timothy Grant[2,5,6,9], Marc Hoemberger[1,7], Renee Otten[1,7], Niels Bradshaw[3], Nikolaus Grigorieff[2,8] & Dorothee Kern[1✉]

Circadian rhythms play an essential part in many biological processes, and only three prokaryotic proteins are required to constitute a true post-translational circadian oscillator[1]. The evolutionary history of the three Kai proteins indicates that KaiC is the oldest member and a central component of the clock[2]. Subsequent additions of KaiB and KaiA regulate the phosphorylation state of KaiC for time synchronization. The canonical KaiABC system in cyanobacteria is well understood[3–6], but little is known about more ancient systems that only possess KaiBC. However, there are reports that they might exhibit a basic, hourglass-like timekeeping mechanism[7–9]. Here we investigate the primordial circadian clock in *Rhodobacter sphaeroides*, which contains only KaiBC, to elucidate its inner workings despite missing KaiA. Using a combination of X-ray crystallography and cryogenic electron microscopy, we find a new dodecameric fold for KaiC, in which two hexamers are held together by a coiled-coil bundle of 12 helices. This interaction is formed by the carboxy-terminal extension of KaiC and serves as an ancient regulatory moiety that is later superseded by KaiA. A coiled-coil register shift between daytime and night-time conformations is connected to phosphorylation sites through a long-range allosteric network that spans over 140 Å. Our kinetic data identify the difference in the ATP-to-ADP ratio between day and night as the environmental cue that drives the clock. They also unravel mechanistic details that shed light on the evolution of self-sustained oscillators.

Circadian clocks are self-sustained biological oscillators that are ubiquitously found in prokaryotic and eukaryotic organisms. In eukaryotes, these systems are complex and highly sophisticated, whereas in prokaryotes, the core mechanism is regulated by a post-translational oscillator that can be reconstituted in vitro with ATP and three proteins (encoded by *kaiA*, *kaiB* and *kaiC*)[1]. Seminal work on the KaiABC system has resulted in a comprehensive understanding of its circadian clock. KaiC is the central component that autophosphorylates by binding to KaiA and autodephosphorylates following association with KaiB[3–6]. The interplay among these three proteins has been shown in vitro to constitute a true circadian oscillator characterized by persistence, resetting and temperature compensation. Consequently, the KaiABC system is considered an elegant and the simplest implementation of a circadian rhythm. The evolutionary history of *kai* genes established *kaiC* as the oldest member dating back around 3.5 billion years ago. Subsequent additions of *kaiB* and most recently *kaiA* formed the extant *kaiBC* and *kaiABC* clusters, respectively[2,10]. Notably, some studies of more primitive organisms that lack *kaiA* hinted that the *kaiBC*-based systems might already provide a basic, hourglass-like timekeeping mechanism[7–9]. Contrary to the self-sustained oscillators found in cyanobacteria, such a timer requires an environmental cue to drive the clock and for the daily flip of the hourglass. The central role of circadian rhythms in many biological processes, controlled by the day and night cycle on Earth, makes their evolution a fascinating topic.

Here we investigate such a primitive circadian clock through biochemical and structural studies of the KaiBC system of the purple, nonsulfur photosynthetic proteobacterium *R. sphaeroides* KD131 (hereafter, its components are referred to as KaiB$_{RS}$ and KaiC$_{RS}$). The organism shows sustained rhythms of gene expression in vivo, but whether *kaiBC* is responsible for this observation remains inconclusive in the absence of a *kaiC* knockout[11]. A previous study of the closely related bacterium *Rhodopseudomonas palustris* that used a knockout strain demonstrated causality between the proto-circadian rhythm of nitrogen fixation and expression of the *kaiC* gene[9]. Here through in vitro experiments, we discover that KaiBC$_{RS}$ is a primordial circadian clock with a mechanism that is different from the widely studied circadian oscillator in *Synechococcus elongatus* PCC 7942 (hereafter, its components are referred to as KaiA$_{SE}$, KaiB$_{SE}$ and KaiC$_{SE}$)[3–6]. We identify an environmental cue that regulates the phosphorylation state and consequently produces a 24 h clock in vivo as the switch in the ATP-to-ADP ratio between day and night. Our results from kinetic studies combined with X-ray and cryogenic electron microscopy (cryo-EM) structures of the relevant states unravel a long-range allosteric pathway that is crucial for the function of the hourglass and sheds light on the evolution of self-sustained

[1]Howard Hughes Medical Institute and Department of Biochemistry, Brandeis University, Waltham, MA, USA. [2]Janelia Research Campus, Howard Hughes Medical Institute, Ashburn, VA, USA. [3]Department of Biochemistry, Brandeis University, Waltham, MA, USA. [4]Present address: Biomolecular Discovery, Relay Therapeutics, Cambridge, MA, USA. [5]Present address: John and Jeanne Rowe Center for Research in Virology, Morgridge Institute for Research, Madison, Madison, WI, USA. [6]Present address: Department of Biochemistry, University of Wisconsin-Madison, Madison, WI, USA. [7]Present address: Treeline Biosciences, Watertown, MA, USA. [8]Present address: Howard Hughes Medical Institute, RNA Therapeutics Institute, University of Massachusetts Chan Medical School, Worcester, MA, USA. [9]These authors contributed equally: Warintra Pitsawong, Ricardo A. P. Pádua, Timothy Grant. ✉e-mail: dkern@brandeis.edu

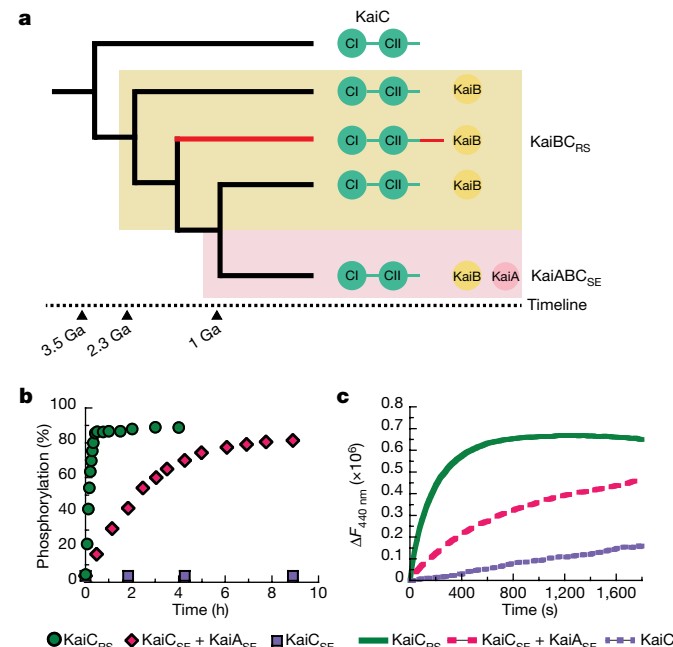

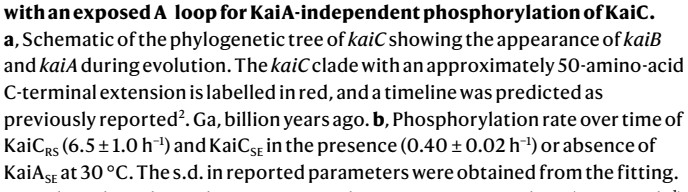

**Fig. 1 | The extended C-terminal tail of KaiC_RS forms a coiled-coil interaction with an exposed A loop for KaiA-independent phosphorylation of KaiC.**
**a**, Schematic of the phylogenetic tree of *kaiC* showing the appearance of *kaiB* and *kaiA* during evolution. The *kaiC* clade with an approximately 50-amino-acid C-terminal extension is labelled in red, and a timeline was predicted as previously reported[2]. Ga, billion years ago. **b**, Phosphorylation rate over time of KaiC_RS ($6.5 \pm 1.0$ h$^{-1}$) and KaiC_SE in the presence ($0.40 \pm 0.02$ h$^{-1}$) or absence of KaiA_SE at 30 °C. The s.d. in reported parameters were obtained from the fitting. **c**, Nucleotide exchange between ATP and mant-ATP in KaiC_RS alone ($18.0 \pm 1.5$ h$^{-1}$) compared with KaiC_SE in the presence ($4.7 \pm 0.3$ h$^{-1}$) and absence of KaiA_SE ($0.08 \pm 0.04$ h$^{-1}$) measured at 30 °C. Representative traces are shown, and the fitted parameters (mean ± s.d.) were obtained from three replicate measurements. **d**, X-ray structure of dodecameric KaiC_RS (PDB: 8DBA) coloured by hexamer A

(light green) and hexamer B (dark green). The CI, CII and coiled-coil domains are indicated, and the A loop is labelled in **e**. **e**, Superposition based on an alignment of the CII domain of KaiC_RS (green; PDB: 8DBA, chain B), KaiC_SE (purple; PDB: 1TF7, chain B)[36] and KaiC_SE-S431E/T432A (yellow; PDB: 7S65, chain A)[19] shows that KaiC_RS has an extended A loop orientation that no longer forms the inhibitory interaction with the 422 loop (KaiC_SE numbering). The conformation of the 422 loop in KaiC_RS resembles the one seen in the cryo-EM structure of the phosphomimetic KaiC_SE-S431E/T432A (yellow; PDB: 7S65)[19]. No electron density is observed for the C-terminal part of wild-type KaiC_SE and the S431E/T432A mutant owing to flexibility, and the missing 22 residues for wild-type KaiC_SE (46 for S431E/T432A) are represented by a dashed line (not shown for the mutant).

oscillators. Notably, we find a new protein fold for KaiC_RS and uncover a register shift in the coiled-coil domain that spans around 115 Å as the key regulator in this system, which shows structural similarities to dynein signalling[12].

## The C-terminal tail is a primitive regulatory moiety

To gain insight into the evolution of the *kaiBC* cluster, we constructed a phylogenetic tree of *kaiC* after the emergence of *kaiB* (Fig. 1a, Extended Data Fig. 1a and Supplementary Datasets 1 and 2). The first question we sought to answer is how KaiC_RS and other members in the clade can autophosphorylate despite having no KaiA. KaiA is known to be crucial for this function in the canonical KaiABC system at its optimum temperature. We observed a large clade that exhibits a C-terminal tail about 50 amino acids longer compared with *kaiC* in other clades (Extended Data Fig. 1b). This C-terminal extension near the A loop is predominantly found in the *kaiC2* subgroup, which was previously annotated as having two serine phosphorylation sites instead of the Thr–Ser pair found in the *kaiC1* and *kaiC3* subgroups[13–15] (Extended Data Fig. 1b). In *S. elongatus*, the binding of KaiA_SE to the A loop of KaiC_SE tethers them in an exposed conformation[16] that activates both autophosphorylation and nucleotide exchange[17]. Given the proximity of the extended C-terminal tail to the A loop, we conjectured that it could serve as the 'primitive' regulatory moiety that was made redundant with the appearance of KaiA.

To test our hypothesis, we first measured the autophosphorylation and nucleotide exchange rates in KaiC_RS, which both depend on the

presence of KaiA in the KaiABC_SE system. We observed an autophosphorylation rate for KaiC_RS that was about 16-fold higher than for KaiC_SE activated by KaiA_SE ($6.5 \pm 1.0$ h$^{-1}$ compared with $0.40 \pm 0.02$ h$^{-1}$, respectively; Fig. 1b and Extended Data Fig. 2a–e). Similarly, the nucleotide exchange rate was faster in KaiC_RS compared with KaiC_SE, even in the presence of KaiA_SE ($18.0 \pm 1.5$ h$^{-1}$ compared with $4.7 \pm 0.3$ h$^{-1}$, respectively; Fig. 1c and Extended Data Fig. 2f). Our data show that KaiC_RS can perform both autophosphorylation and nucleotide exchange on its own and does so faster than its more recently evolved counterparts.

## A coiled-coil interaction assembles a KaiC_RS dodecamer

To mechanistically assess how KaiC in *kaiA*-null systems accomplishes autophosphorylation, we turned to structural biology. The crystal structure of KaiC_RS, unlike KaiC from cyanobacteria, revealed a homo-dodecamer that consisted of two homohexameric domains joined by a 12-helical coiled-coil domain that is formed by the extended C-terminal tail (Protein Data Bank (PDB) identifier: 8DBA; Fig. 1d and Extended Data Table 1). A closer inspection of the CII domains in KaiC_RS and KaiC_SE/TE (*Thermosynechococcus elongatus* BP-1 referred to as KaiC_TE) showed an obvious difference in A loop orientations: an extended conformation in KaiC_RS compared with a buried orientation in KaiC_SE/TE (Fig. 1e). The existence of such an extended conformation following binding of KaiA has been previously proposed[18]. This hypothesis was based on the perceived hyperphosphorylation and hypophosphorylation that occurred after removing the A loop or disrupting KaiA binding, respectively[18].

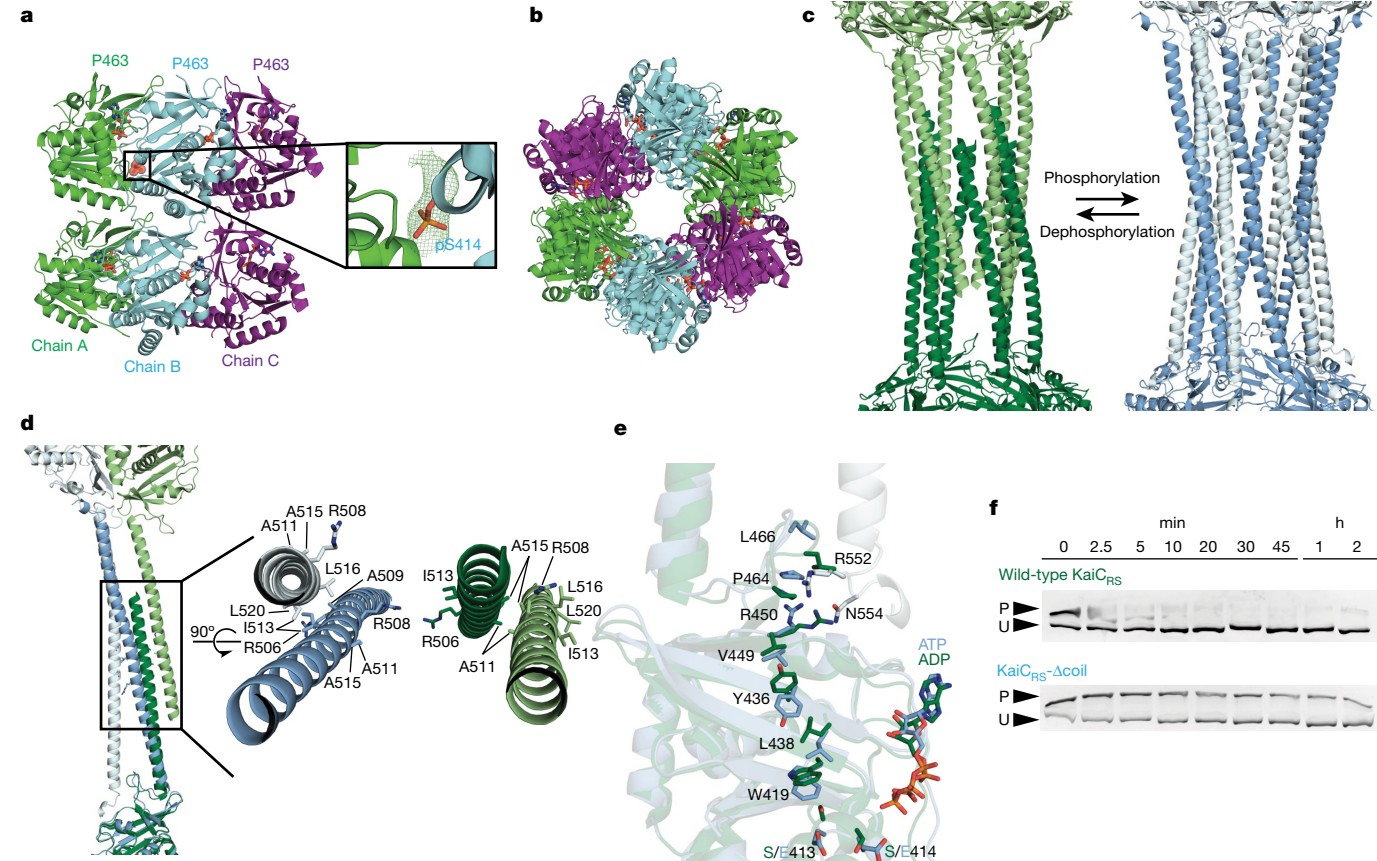

**Fig. 2 | A coiled-coil partner switch coupled to an allosteric network in the CII domain promotes autodephosphorylation. a**, X-ray structure of KaiC$_{RS}$-Δcoil was solved in the C222$_1$ space group and contained three monomers in the asymmetric unit, with ADP present in all active sites. The obtained electron density map allowed for model building up to Pro463, which indicated that the truncation at position 490 results in enhanced flexibility beyond Pro463. Phosphorylation of Ser414 (pS414) was observed in chain B (cyan) as shown by the electron density $mF_o$–$DF_c$ polder map (green mesh, 3$\sigma$ contour level). **b**, Assembly analysis using the PISA software[37] revealed a hexamer as the most probable quaternary structure (top view). **c**, Structural comparison of the coiled-coil domain for unphosphorylated KaiC$_{RS}$ (dark and light green; X-ray structure) and the KaiC$_{RS}$-S413E/S414E phosphomimetic mutant (dark and light blue; cryo-EM structure). **d**, Overlay of interacting dimers of the structures in **c** using the CII domain of chain A as a reference

(dark shades; bottom). Unphosphorylated KaiC$_{RS}$ (dark green) interacts with the opposite partner on the right (light green), whereas KaiC$_{RS}$-S413E/S414E (dark blue) interacts with the partner on the left (light blue). The hydrophobic packing in the coiled-coil domain is mediated by only the Cβ atoms of alanine and arginine residues in unphosphorylated KaiC$_{RS}$, but involves the entire side chain of leucine and isoleucine residues in the phosphomimetic structure. **e**, Allosteric network in the phosphomimetic state (blue) from the coil (light blue) propagating through the KaiC$_{RS}$ CII domain to the active site (dark blue) compared with the unphosphorylated state (dark green) (Supplementary Video 1). **f**, Autodephosphorylation of KaiC$_{RS}$ and KaiC$_{RS}$-Δcoil over time in the presence of 4 mM ADP at 30 °C. The phosphorylated (P) and unphosphorylated (U) proteins were separated by Zn$^{2+}$ Phos-tag SDS–PAGE (for gel source data, see Supplementary Fig. 1).

A recently solved cryo-EM structure of the night-time phosphomimetic KaiC$_{SE}$-S431E/T432A in its compressed state directly showed a disordered A loop that no longer interacts with the 422 loop[19], similar to the extended A loop conformation we observed in KaiC$_{RS}$ (Fig. 1e). The loss of interaction between the A loop and the 422 loop (just 10 residues apart from the phosphorylation sites) results in closer proximity between the hydroxyl group of Ser431–Thr432 and the γ-phosphate of ATP, thereby, facilitating the phosphoryl transfer step[20]. Furthermore, the sequence similarity between KaiC$_{RS}$ and KaiC$_{SE}$ is less than 30% for the A loop and residues considered important for stabilization of this loop in its buried orientation (that is, the 422 loop and residues 438–444) (Fig. 1e). Together, our structural and kinetic data support the idea that an exposed A loop is key for the KaiA-independent enhancement of nucleotide exchange and hence autophosphorylation in KaiC$_{RS}$ and perhaps other KaiBC-based systems.

We then questioned whether the purpose of the coiled-coil domain is to 'pull up' the A loop or to actively participate in nucleotide exchange and autophosphorylation of KaiC. To further understand its role, we generated a truncation at residue Glu490 based on the phylogenetic

tree and crystallographic information (KaiC$_{RS}$-Δcoil) (Extended Data Fig. 1b) to disrupt the coiled-coil interaction between the two hexamers. The crystal structure of KaiC$_{RS}$-Δcoil (PDB: 8DB3; Fig. 2a,b and Extended Data Table 1), its size-exclusion chromatogram and analytical ultracentrifugation profile (Extended Data Fig. 3a–c) showed a hexameric structure with no coiled-coil interaction. Nucleotide exchange rates in the CII domain for KaiC$_{RS}$-Δcoil and the wild-type protein were comparable (19.1 ± 0.8 h$^{-1}$ and 18.0 ± 1.5 h$^{-1}$, respectively; Extended Data Fig. 3d). The phosphorylation rates were also similar (5.5 ± 0.4 h$^{-1}$ and 7.4 ± 0.3 h$^{-1}$ for KaiC$_{RS}$-Δcoil and wild type, respectively; Extended Data Fig. 3e,f). These results indicate that the extended A loop and not the coiled-coil interaction plays a pivotal part in nucleotide exchange and autophosphorylation in KaiC$_{RS}$. The results also provide a potential mechanism of autophosphorylation in other KaiBC-based systems that lack a coiled-coil bundle. Notably, the coiled-coil bundle provides additional hexameric stability. In detail, the KaiC$_{RS}$ dodecamer is stable for extended periods of time in the presence of only ADP (Extended Data Fig. 3g,h), whereas for KaiC$_{SE}$, oligomers are not observed under these conditions[21].

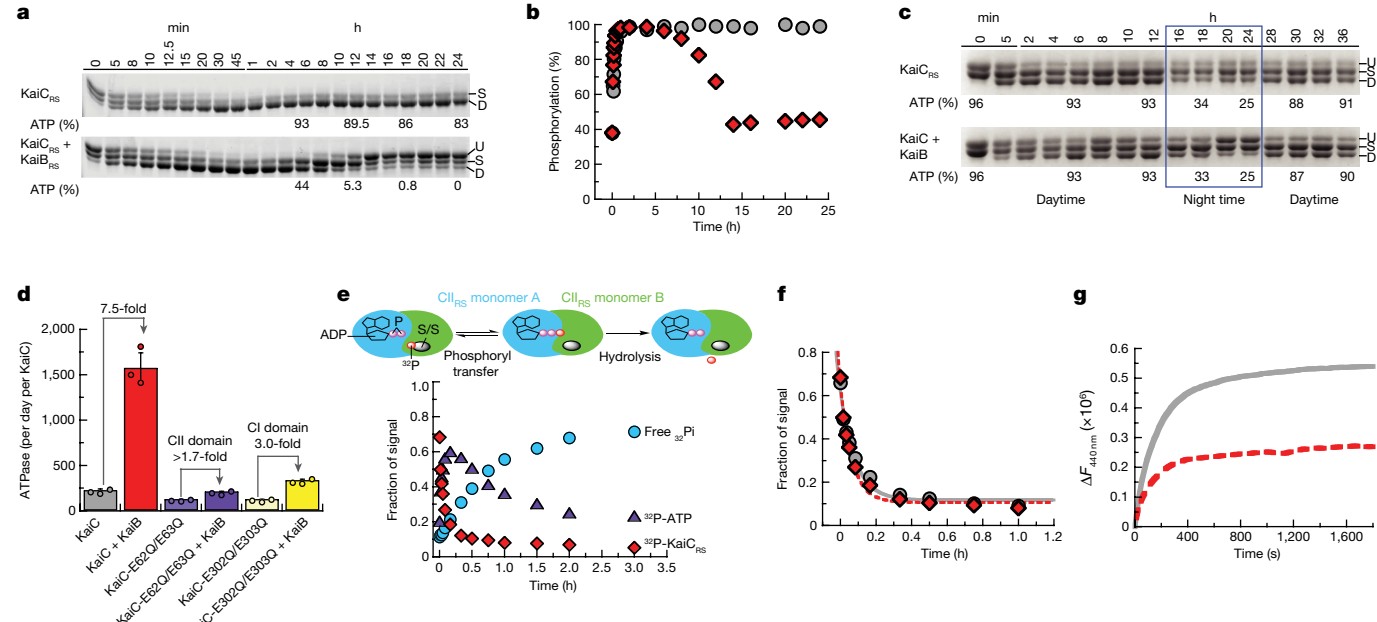

**Fig. 3 | The regulatory role of KaiB$_{RS}$ in the phosphorylation–dephosphorylation cycle of KaiC$_{RS}$. a**, SDS–PAGE gel of 3.5 μM KaiC$_{RS}$ and 4 mM ATP in the absence (top) and presence (bottom) of 3.5 μM KaiB$_{RS}$ at 35 °C, with the percentage of ATP indicated at specific time points. **b**, Phosphorylation (single and double) of KaiC$_{RS}$ during the reaction in the absence (grey circles) or presence (red diamonds) of KaiB$_{RS}$. **c**, Phosphorylation–dephosphorylation cycle of 3.5 μM phosphorylated KaiC$_{RS}$ in the absence and presence of 3.5 μM KaiB$_{RS}$ in a constant ATP-to-ADP ratio of high ATP (4 mM) to mimic daytime and about 25% ATP to mimic the night time (exact percentage of ATP indicated at specific time points) at 30 °C. U, S and D in **a** and **c** represent the unphosphorylated, single phosphorylated (at Ser413 or Ser414) and double phosphorylated state of KaiC$_{RS}$, respectively (for gel source data, see Supplementary Fig. 1). **d**, ATPase activity of wild-type KaiC$_{RS}$ in the absence and presence of KaiB$_{RS}$, KaiC$_{RS}$-E62Q/E63Q in the

absence and presence of KaiB$_{RS}$, and KaiC$_{RS}$-E302Q/E303Q in the absence and presence of KaiB$_{RS}$ at 30 °C. Bar graphs show mean ± s.d. from three replicates. **e**, Time-dependent autodephosphorylation of $^{32}$P-labelled KaiC$_{RS}$ bound with ADP in the presence of 20 μM KaiB$_{RS}$ and 4 mM ADP at 30 °C showing phosphorylated $^{32}$P-KaiC$_{RS}$, $^{32}$P-ATP and free $^{32}$Pi. The reaction products were separated by thin layer chromatography. **f**, The decay of phosphorylated $^{32}$P-KaiC$_{RS}$ bound with 4 mM ADP in the absence (grey circles) and presence (red diamonds) of KaiB$_{RS}$ at 30 °C is obtained from autoradiography quantification (Extended Data Fig. 7). **g**, The nucleotide exchange of 3.5 μM KaiC$_{RS}$ (grey trace) and 3.5 μM KaiC$_{RS}$ in complex with 30 μM KaiB$_{RS}$ (red dotted trace) in the presence of ATP with mant-ATP. Representative traces are shown, and the fitted parameters (mean ± s.d.) were obtained from three replicate measurements.

## A long-range allosteric network in KaiC$_{RS}$

The change in phosphorylation state of KaiC has been well established to be the central feature for the circadian rhythm[22,23]. Notably, when comparing the unphosphorylated form of full-length KaiC$_{RS}$ (PDB: 8DBA) and its phosphomimetic mutant (S413E/S414E; PDB: 8FWI) (Extended Data Fig. 4 and Extended Data Table 2), we observed two distinct coiled-coil interactions. Following phosphorylation, the coiled-coil pairs swap partners by interacting with the other neighbouring chain from the opposite hexamer, which resulted in a register shift that propagated around 115 Å along the entire coiled-coil (Fig. 2c and Extended Data Fig. 5). In the phosphomimetic state, the register comprised bulkier hydrophobic residues that resulted in a more stable interaction than for the dephosphorylated form (Fig. 2d and Extended Data Fig. 3g). Furthermore, the C-terminal residues of KaiC$_{RS}$-S413E/S414E interacted with the CII domain of the opposite hexamer, whereas the lack of electron density for the last 30 residues in the wild-type structure indicates more flexibility in the dephosphorylated state. We discovered that these conformational changes in the coiled-coil domain seemed to be coupled through a long-range allosteric network to the phosphorylation sites. The rotameric states of residues Ser413, Ser414, Trp419, Val421, Tyr436, Leu438, Val449 and Arg450 moved concertedly and pointed towards the nucleotide-binding site when the protein was phosphorylated or pointed away in the absence of a phosphate group (Fig. 2e, Extended Data Fig. 5d and Supplementary Video 1). We propose that the proximity of the nucleotide to the phosphorylated residue facilitated more efficient phosphoryl transfer. We therefore experimentally determined the impact of the coiled-coil

domain on the autodephosphorylation rate of KaiC$_{RS}$. The wild-type protein dephosphorylated comparatively quickly (observed rate constant = 11.5 ± 0.8 h$^{-1}$) in the presence of only ADP. By contrast, little dephosphorylation was observed for KaiC$_{RS}$-Δcoil (Fig. 2f and Extended Data Fig. 3i), for which allosteric propagation was disrupted (Extended Data Fig. 5d). Consistent with this accelerated dephosphorylation rate mediated by the coiled-coil domain, our crystallographic data showed a phosphate group on Ser414 for KaiC$_{RS}$-Δcoil but not for the wild-type protein (Fig. 2a and Extended Data Fig. 5d).

## The ATP-to-ADP ratio resets the clock

It was notable that KaiC$_{RS}$ can autodephosphorylate on its own despite being constitutively active for phosphorylation owing to its extended A loop conformation. In the canonical *kaiABC* system, the interaction between KaiB and KaiC is required to provide a new binding interface that sequesters KaiA from its activating binding site, thereby promoting autodephosphorylation at the optimum temperature of the organism[24–26]. We therefore sought to discover whether the KaiC$_{RS}$ system can oscillate and whether there is a regulatory role for KaiB$_{RS}$ in this process. Comparing the in vitro phosphorylation states of KaiC$_{RS}$ in the absence and presence of KaiB$_{RS}$ showed an initial, rapid phosphorylation followed by an oscillatory-like pattern in the presence of KaiB$_{RS}$ (hereafter referred to as KaiBC$_{RS}$), whereas KaiC$_{RS}$ alone remained phosphorylated (Fig. 3a,b). Notably, the ATP consumption during the reaction with KaiB$_{RS}$ was significantly higher than without (Fig. 3a). As noted above, KaiC$_{RS}$ will also dephosphorylate completely in the presence of only ADP (Fig. 2f). These results suggest that the phosphorylation state

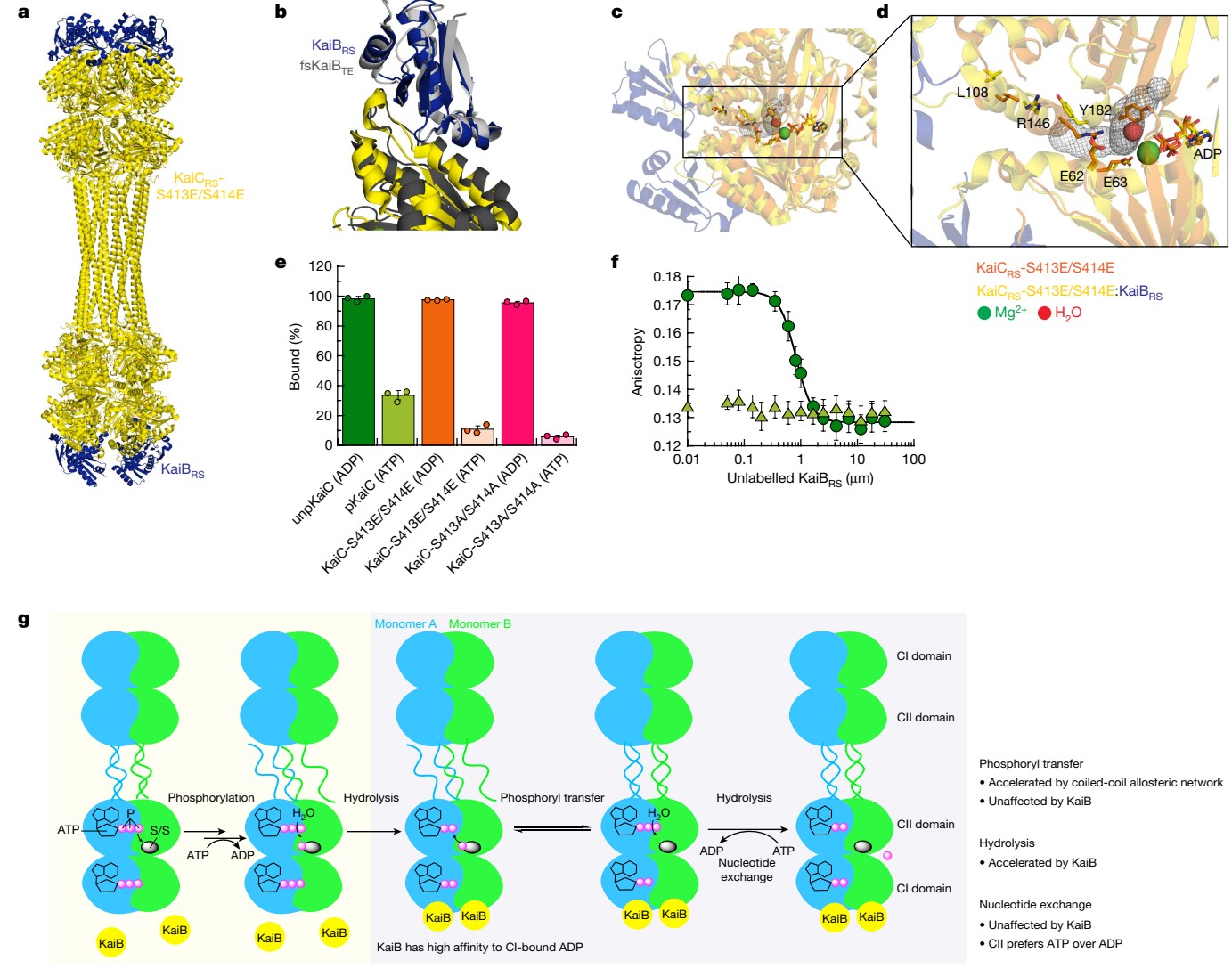

**Fig. 4 | KaiB_RS binds to the post-hydrolysis state and accelerates the ATPase activity of KaiC_RS. a**, Cryo-EM structure of KaiC_RS-S413E/S414E (yellow) in complex with KaiB_RS (blue) (PDB: 8FWJ). **b**, Superposition of KaiC_RS-S413E/S414E (yellow) bound to KaiB_RS (blue) (PDB: 8FWJ) and KaiC_TE-S413E (dark grey) bound to fsKaiB_TE (fold-switch, binding-competent state of KaiB_TE; light grey) (PDB: 5JWQ)[26]. **c,d**, Binding of KaiB_RS (blue) creates a tunnel (grey mesh) that enables water to reach the catalytic position (red sphere) for ATP hydrolysis in the CI domain. **e**, Binding of wild-type and mutant forms of KaiB_RS to His-tagged KaiB_RS in the presence of ADP or an ATP-recycling system at 25 °C. Bar graphs show mean ± s.d. from three replicates. **f**, Fluorescence anisotropy of unlabelled KaiB_RS competitively displacing KaiB_RS−6IAF (where 6IAF is the fluorophore) from unphosphorylated KaiC_RS in the presence of ADP (dark green circles) and phosphorylated KaiC_RS in the presence of the ATP-recycling system (light green triangles) at 30 °C. The average anisotropy and standard error were calculated from ten replicate measurements. **g**, Schematic of the uncovered mechanism of KaiC_RS regulated by coiled-coil interactions and KaiB_RS in the CI and CII domains.

of KaiC_RS and the observed oscillatory half-cycle (Fig. 3a,b) is probably related to a change in the ATP-to-ADP ratio. We conjectured that this could constitute the environmental cue to reset the timer. To test our hypothesis, an ATP-recycling system was added after complete dephosphorylation of KaiBC_RS. As predicted, KaiC_RS was able to restart the cycle and phosphorylate again (Extended Data Fig. 6a). We note that in vivo, the ATP-to-ADP ratio will not vary as substantially as in this in vitro experiment, as nucleotide homeostasis is tightly regulated. To mimic the day and night period for *R. sphaeroides*, we repeated the experiments while keeping the ATP-to-ADP ratio constant (mostly ATP at daytime owing to photosynthesis compared with 25:75% ATP-to-ADP during night time)[27]. In the presence of high ATP (that is, mimicking daytime), KaiC_RS remained single or double phosphorylated (Fig. 3c and Extended Data Fig. 6b) irrespective of KaiB_RS. By contrast, a constant 25:75% ATP-to-ADP ratio (that is, mimicking night time) resulted in a

much higher fraction of dephosphorylated KaiC_RS in the presence of KaiB_RS (Fig. 3c). Moreover, when the ATP-to-ADP ratio was flipped to mimic daytime, KaiC_RS was able to phosphorylate again (Fig. 3c, around the 28 h mark). Our data support the notion that the phosphorylation behaviour of KaiBC_RS strongly depends on the ATP-to-ADP ratio and demonstrate that the physical binding of KaiB_RS results in a higher level of KaiC_RS dephosphorylation at night time.

Next we investigated the accelerated ATPase activity observed in KaiC_RS after the formation of the complex. The ATPase activity reported for KaiC_SE is low (about 15 ATP molecules per day per molecule of KaiC_SE) and was proposed as a reason for the slowness of circadian oscillation[28]. KaiC_RS alone shows a significantly faster ATPase rate than KaiC_SE, which is further enhanced by binding of KaiB_RS (208 ± 19 and 1,557 ± 172 ATP molecules day per day per KaiC_RS, respectively; left two bars in Fig. 3d and Extended Data Fig. 6c–g). Furthermore, KaiC_RS does not exhibit

temperature compensation for its ATPase activity (temperature coefficient $Q_{10}$ about 1.9; Extended Data Fig. 6c), a feature that is present in KaiC$_{SE}$ and proposed to be a prerequisite for self-sustained rhythms[28]. The deviation from unity for $Q_{10}$ is consistent with our earlier observation that the KaiBC$_{RS}$ system is not a true circadian oscillator but rather an hourglass timer (Fig. 3b).

## Regulatory role of KaiB$_{RS}$

Mechanistic details of how the binding of KaiB$_{RS}$ in the CI domain allosterically affects the autodephosphorylation of KaiC$_{RS}$ in the CII domain remain unclear. There are three plausible scenarios to explain this: (1) KaiB$_{RS}$ binding stimulates the phosphoryl transfer from pSer back to ADP (Extended Data Fig. 7a); (2) KaiB$_{RS}$ binding increases the hydrolysis rate of the active-site ATP (Extended Data Fig. 8a); or (3) KaiB$_{RS}$ binding accelerates nucleotide exchange in the CII domain (Extended Data Fig. 8e). To differentiate among these possibilities, we performed radioactivity experiments to follow nucleotide interconversion. We also measured ATPase activity for wild-type KaiC$_{RS}$ and mutant forms that are incapable of ATPase activity in the CI or CII domain, and quantified nucleotide-exchange rates by measuring the fluorescence of mant-ATP. First, we detected fast, transient $^{32}$P-ATP formation in our radioactivity experiments when starting from $^{32}$P-phosphorylated KaiC$_{RS}$, which was due to its ATP synthase activity in the CII domain (Fig. 3e and Extended Data Fig. 7b–d). The observed phosphoryl-transfer rate was independent of KaiB$_{RS}$ (observed rate constant = $12.0 \pm 1.7$ h$^{-1}$ and $15.4 \pm 1.7$ h$^{-1}$ in its absence and presence, respectively; Fig. 3f) and agreed well with the rates determined from our gel electrophoresis experiments ($11.0 \pm 0.8$ h$^{-1}$ and $11.5 \pm 0.8$ h$^{-1}$ with or without KaiB$_{RS}$, respectively; Extended Data Fig. 7e,f). Our experimental data confirmed that KaiC$_{RS}$ undergoes dephosphorylation through an ATP synthase mechanism, similar to what was observed for KaiC$_{SE}$ (ref. [29]). KaiB does not expedite the actual phosphoryl-transfer reaction, which is never the rate-limiting step. As we were unable to stabilize the first phosphorylation site (Ser414) in the presence of ADP, the rates reported here correspond exclusively to dephosphorylation of Ser413. Second, to deconvolute the contributions of the CI and CII domains to the observed ATPase activity, we measured ADP production from KaiC$_{RS}$ mutants that abolish hydrolysis in either the CI domain (KaiC$_{RS}$-E62Q/E63Q) or the CII domain (KaiC$_{RS}$-E302Q/E303Q). For wild-type KaiC$_{RS}$, the binding of KaiB$_{RS}$ resulted in a 7.5-fold increase in ATPase activity, and both domains were affected and contributed additively (3-fold for CI and at least 1.7-fold for CII) to the overall effect (Fig. 3d and Extended Data Fig. 8b–d). Of note, the fold increase in the CII domain represents a lower limit as the mutations induced to generate KaiC$_{RS}$-E62Q/E63Q interfere with KaiB$_{RS}$ binding, as previously reported for KaiC$_{SE}$ (ref. [30]). Third, our measurements of nucleotide exchange showed that this rate is also unaffected by KaiB$_{RS}$ binding ($19.8 \pm 1.8$ h$^{-1}$ and $18.0 \pm 1.5$ h$^{-1}$ with or without KaiB$_{RS}$, respectively; Fig. 3g). As there is no tryptophan residue near the nucleotide-binding site in the CI domain, only the exchange rate in the CII domain could be determined. Notably, the change in fluorescence amplitude was smaller in the presence of KaiB$_{RS}$, which demonstrates that even though the binding of KaiB$_{RS}$ does not accelerate nucleotide exchange, it appears to induce a conformational rearrangement in the CII domain, especially at higher temperatures (Fig. 3g and Extended Data Fig. 8f–h).

## Structure of the KaiBC$_{RS}$ complex

To elucidate the structural underpinning of the enhanced ATPase activity of KaiC$_{RS}$ after KaiB$_{RS}$ binding, we solved the cryo-EM structures of KaiC$_{RS}$ alone (PDB: 8FWI) and in complex with KaiB$_{RS}$ (PDB: 8FWJ) (Extended Data Table 2). Twelve KaiB$_{RS}$ molecules (monomeric in solution; Extended Data Fig. 9a) bind to the CI domain of the KaiC$_{RS}$-S413E/S414E dodecamer (Fig. 4a–c and Extended Data Fig. 9b).

The bound state of KaiB$_{RS}$ adopts the same fold-switch conformation as observed for KaiB$_{TE}$ (ref. [25]) and suggests that this is the canonical binding-competent state (Fig. 4b). Following binding of KaiB$_{RS}$, the CI–CI interfaces loosen up (Fig. 4c), which enables the formation of a tunnel that connects bulk solvent to the position of the hydrolytic water in the active sites (Fig. 4d and Extended Data Fig. 9c). There are other lines of evidence for the weakened interactions within the CI domains. First, KaiB$_{RS}$ binding to either KaiC$_{RS}$-CI domain (Extended Data Fig. 10a) or KaiC$_{RS}$-Δcoil (that is, missing the C-terminal extensions; Extended Data Fig. 10b) resulted in disassembly of the hexameric KaiC$_{RS}$ structure into its monomers. By contrast, full-length KaiC$_{RS}$ maintained its oligomeric state following binding of KaiB$_{RS}$, which is probably due to the stabilization provided by the coiled-coil interaction. Second, a decrease in melting temperature ($T_m$) of KaiC$_{RS}$ was observed with increasing KaiB$_{RS}$ concentration (Extended Data Fig. 10c). There was no interaction between neighbouring KaiB$_{RS}$ molecules within the complex (Extended Data Fig. 9b), which suggests that there is a non-cooperative assembly of KaiB$_{RS}$ to KaiC$_{RS}$. This result is contrary to what has been observed for KaiBC$_{SE}$ and KaiBC$_{TE}$ complexes[31,32].

Furthermore, we noted that KaiB-bound structures in phosphomimetic variants of KaiC$_{RS}$ (Fig. 4c,d) and KaiC$_{SE}$ (ref. [26]) have ADP bound in their CI domain. This result demonstrates that the post-hydrolysis state is also the binding-competent state for KaiB$_{RS}$. To test this hypothesis, a His-tagged KaiB$_{RS}$ protein was used in pull-down assays to detect its physical interaction with wild-type and mutant forms of KaiC$_{RS}$ bound with either ADP or ATP. Nearly all KaiB$_{RS}$ was complexed to ADP-bound KaiC$_{RS}$, whereas less than 30% co-eluted in the ATP-bound form, regardless of the phosphorylation state (Fig. 4e and Extended Data Fig. 10d,e). The formation of complexes depended inversely on the ATP-to-ADP ratio (Extended Data Fig. 10f). We performed fluorescence anisotropy competition experiments to obtain a more quantitative description of the binding interaction between KaiC$_{RS}$ and KaiB$_{RS}$. Highly similar dissociation constant ($K_d$) values were obtained for unphosphorylated, wild-type KaiC$_{RS}$ (Fig. 4f) and its phosphomimetic form (Extended Data Fig. 10g) bound with ADP ($0.42 \pm 0.03$ μM and $0.79 \pm 0.06$ μM, respectively). No measurable binding curves were obtained for ATP-bound phosphorylated wild-type KaiC$_{RS}$ (Fig. 4f) or for KaiC$_{RS}$-S413E/S414E (Extended Data Fig. 10g) with ATP-recycling system, which is probably due to the small fraction of complex present. Our data show that the post-hydrolysis state in the CI domain is key for KaiB$_{RS}$ binding, whereas the phosphorylation state of KaiC$_{RS}$ has only a marginal effect.

In summary, we demonstrate that binding of KaiB$_{RS}$ at the CI domain in the post-hydrolysis state facilitates the hydrolysis of transiently formed ATP after dephosphorylation of KaiC$_{RS}$ in the CII domain (Fig. 4g). Our fluorescence experiments (Fig. 3g and Extended Data Fig. 8f) detected a conformational change in the CII domain following KaiB$_{RS}$ binding, but we did not observe major structural changes in the cryo-EM structures. Based on the temperature dependence of the fluorescence amplitudes (Extended Data Fig. 8f), we conjecture that the inability to detect conformational differences is probably because of the low temperature. As the CII domain prefers to bind ATP over ADP (Extended Data Fig. 10h), ATP hydrolysis in the CII domain stimulated by KaiB$_{RS}$ is particularly important to keep KaiC$_{RS}$ in its dephosphorylated state at night time. During this period, the exogenous ATP-to-ADP ratio remains sufficiently high to otherwise result in ATP-binding in the CII active site (Fig. 3c and Extended Data Fig. 6b).

## Discussion

The KaiBC$_{RS}$ system studied here represents a primordial, hourglass timekeeping machinery, and its mechanism provides insight into more evolved circadian oscillators such as KaiABC. The dodecameric KaiC$_{RS}$ showed constitutive kinase activity owing to its extended C-terminal tail that forms a coiled-coil bundle with the opposing hexamer. This structure elicits a conformation akin to the exposed A loop conformation

in KaiAC$_{SE}$, and autophosphorylation occurs within half an hour. In the KaiABC$_{SE}$ system, the transition from unphosphorylated to double phosphorylated KaiC takes place over about 12 h, and the fine-tuning of this first half of the circadian rhythm is accomplished by the emergence of KaiA$_{SE}$ during evolution. The second clock protein, KaiB, binds the CI domain with the same fold-switched state in both systems. The interaction is controlled by the phosphorylation state in the KaiABC$_{SE}$ system, and its sole function is to sequester KaiA$_{SE}$ from the activating binding site, whereas KaiB binding directly accelerates ATPase activity in the KaiBC$_{RS}$ system regardless of the phosphorylation state. The KaiA$_{SE}$ system requires an environmental switch in the ATP-to-ADP concentration to reset the clock. The system therefore follows the day–night schedule when nucleotide concentrations inherently fluctuate in the organism. By contrast, the self-sustained oscillator KaiABC$_{SE}$ remains functional over a wide range of nucleotide concentrations and responds to changes in the ATP-to-ADP ratio by changing its phosphorylation period and amplitude to remain entrained with the day–night cycle[33].

The newly reported structural fold of KaiC utilizes the versatile coiled-coil architecture as part of a long-range allosteric network that regulates KaiC$_{RS}$ dephosphorylation. Nature uses conformational changes in coiled-coil domains for a variety of regulatory functions[34], including the activity of the motor protein dynein in the cellular transport of cargo along the actin filament[12]. A similar register shift, although in a coiled-coil interaction formed by only two helices, is used in dynein motility. Given that this simple heptad repeat sequence emerged multiple times and is found throughout all kingdoms of life[35], it is an example of convergent evolution.

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

## Methods

### Construct of KaiC and KaiB expression vectors

The wild-type KaiC$_{RS}$ (GenBank identifier: ACM04290.1) and KaiB$_{RS}$ (GenBank: WP_002725098.1) from *R. sphaeroides* strain KD131/KCTC 12085 (equivalent: *Cereibacter sphaeroides* strain KD131) constructs used in this paper were ordered from GenScript (Supplementary Table 1). Codon-optimized plasmids for KaiC$_{RS}$ and KaiB$_{RS}$ were subcloned into NcoI/KpnI sites of the pETM-41 vector. A QuikChange II Site-Directed Mutagenesis kit (Agilent Technologies) was used to generate single mutant, double mutant and truncated versions of KaiC$_{RS}$. The truncated KaiC$_{RS}$ (KaiC$_{RS}$-Δcoil and KaiCI$_{RS}$) were generated by introducing stop codons in the KaiC$_{RS}$ wild-type plasmid. All primers were ordered from Genewiz (Supplementary Table 2). The presence of the intended KaiC$_{RS}$ mutations in the plasmid was confirmed by DNA sequencing by Genewiz using primers ordered from the same company (listed in Supplementary Table 2).

Both KaiC$_{SE}$ and KaiA$_{SE}$ plasmids were a gift from E. K. O'Shea. Expression and purification were performed according to a previously described procedure[6].

### Expression and purification of KaiC$_{RS}$ and KaiB$_{RS}$ from *R. sphaeroides*

KaiC$_{RS}$, KaiC$_{RS}$ mutants and KaiB$_{RS}$ were expressed in *Escherichia coli* BL21(DE3) cells (New England Biolabs) harbouring the plasmid pETM-41 containing the *kaiC$_{RS}$* or *kaiB$_{RS}$* gene. Three colonies from a freshly prepared transformation plate were inoculated into 1 litre of TB medium containing 50 μg ml$^{-1}$ kanamycin. This culture was grown at 25 °C with shaking at 220 r.p.m. for 48 h without IPTG induction (leaky expression). The cells were pelleted by centrifugation at 4,200 r.p.m. for 15 min at 4 °C and stored at −80 °C.

Frozen cell pellets of KaiC$_{RS}$ and KaiC$_{RS}$ mutants were resuspended into lysate buffer (buffer A$_{C-RS}$) containing 1× EDTA-free protease inhibitor cocktail (Thermo Fisher Scientific), DNAse I (Sigma Aldrich) and lysozyme (Sigma Aldrich), and the lysate was sonicated for 10–15 min (20 s on, 30 s off, output power less than 40%) on ice followed by centrifugation at 18,000 r.p.m. at 4 °C for 45 min to remove cell debris. The lysate was filtered through a 0.45 μm filter and then loaded on HisTrap HP prepacked Ni-sepharose columns (Cytiva) pre-equilibrated with buffer A$_{C-RS}$ at 0.5 ml min$^{-1}$. The column was washed with buffer A$_{C-RS}$ at 1 ml min$^{-1}$ until the UV absorbance returned to baseline. Impurities were then washed with 15% buffer B$_{C-RS}$, and the protein was eluted with 50% buffer B$_{C-RS}$. The eluted complex was diluted with 1.5-fold dialysis buffer$_{C-RS}$ then subjected to in-house prepared His-tagged TEV protease (1:10, TEVP:KaiC$_{RS}$ molar ratio) cleavage to remove the His$_6$–MBP tag from KaiC$_{RS}$ (wild-type and mutants) overnight at 4 °C in 6–8 kDa snakeskin dialysis tubing (Thermo Fisher Scientific) that was exchanged against dialysis buffer$_{C-RS}$. Cleaved KaiC$_{RS}$ was filtered through a 1 μm filter and once again loaded onto HisTrap HP prepacked Ni-sepharose columns at 0.5 ml min$^{-1}$ to remove His-tagged TEV protease, His$_6$–MBP tag and uncleaved protein. The flow through was concentrated using a Millipore Amicon Ultra-15 centrifugal filter device (10 kDa cut-off) and immediately passed through a HiPrep Sephacryl S-400 HR column (Cytiva) pre-equilibrated with buffer C$_{C-RS}$. Protein was purified to homogeneity with a single band on Bis-Tris 4–12% gradient SDS–PAGE gel (Genscript) at 62.5 kDa. All protein purification steps were done at 4 °C or on ice. Protein was aliquoted and flash-frozen before storage at −80 °C until further use. The protein concentration was measured using a Microplate BCA Protein Assay kit (Thermo Fisher Scientific) on a SpectraMax MiniMax 300 imaging cytometer using BSA as a standard curve. Typical yields of KaiC$_{RS}$ (wild-type and mutants) were 20–40 mg l$^{-1}$ of culture.

To test whether the purified KaiC$_{RS}$ had any ATPase contamination, Q-sepharose HP columns (Cytiva) pre-equilibrated with buffer D$_{C-RS}$ was used before the final HiPrep Sephacryl S-400 HR column. The protein

was eluted with 5 CV of a linear gradient from 0 to 100% buffer E$_{C-RS}$. The ATPase activity of the protein samples purified using Q-sepharose HP columns was identical to the samples without this additional purification step. Buffer A$_{C-RS}$ comprised 50 mM Tris-base (pH 7.5), 250 mM NaCl, 10 mM imidazole, 2 mM TCEP, 5 mM MgCl$_2$, 1 mM ATP and 10% glycerol (v/v). Buffer B$_{C-RS}$ comprised 50 mM Tris-base (pH 7.5), 250 mM NaCl, 500 mM imidazole, 2 mM TCEP, 5 mM MgCl$_2$, 1 mM ATP and 10% glycerol (v/v). Dialysis$_{C-RS}$ comprised 50 mM Tris-base (pH 7.0), 50 mM NaCl, 2 mM TCEP, 5 mM MgCl$_2$, 1 mM ATP and 10% glycerol (v/v). Buffer C$_{C-RS}$ comprised 50 mM Tris-base (pH 7.0), 50 mM NaCl, 2 mM TCEP, 5 mM MgCl$_2$, 1 mM ATP and 10% glycerol (v/v). Buffer D$_{C-RS}$ comprised 50 mM Tris-base (pH 7.0), 2 mM TCEP, 5 mM MgCl$_2$, 1 mM ATP and 10% glycerol (v/v). Buffer E$_{C-RS}$ comprised 50 mM Tris-base (pH 7.0), 1 M NaCl, 2 mM TCEP, 5 mM MgCl$_2$, 1 mM ATP and 10% glycerol (v/v).

The purification of KaiB$_{RS}$ was similar to KaiC$_{RS}$, but with slight modifications as outlined below. After sonication and centrifugation to remove cell debris, the lysate was filtered through a 0.22 μm filter and then passed through HisTrap HP prepacked Ni Sepharose columns, pre-equilibrated with buffer A$_{B-RS}$. The column was washed with buffer A$_{B-RS}$ until the UV absorbance returned to baseline. Impurities were washed with 5% buffer B$_{B-RS}$, and the protein was eluted with 50% buffer B$_{B-RS}$. The fusion protein was concentrated down to around 30 ml using Amicon stirred cells (Millipore Sigma) with 10 kDa cut-off. In-house prepared His-tagged TEV protease was added, and the fusion protein was cleaved overnight at 4 °C in a 3.5 kDa dialysis cassette that was exchanged against dialysis$_{B-RS}$ buffer. The cleaved KaiB$_{RS}$ was passed through HisTrap HP prepacked Ni-sepharose columns and concentrated down to about 10 ml using a Millipore Amicon Ultra-15 centrifugal filter device (3.5 kDa cut-off). The protein sample was then loaded onto a 26/60 Superdex S75 gel-filtration column (Cytiva) pre-equilibrated with buffer C$_{B-RS}$ at 4 °C. The eluted protein was loaded onto Q-sepharose HP columns pre-equilibrated with buffer D$_{B-RS}$ to remove ATPase contamination. The protein was eluted out in the flow through. A gradient from 0 to 100% buffer E$_{B-RS}$ was passed through Q-sepharose HP columns to ensure that no KaiB$_{RS}$ was bound to the columns. Protein was purified to homogeneity with the single band on Bis-Tris 4–12% gradient SDS–PAGE gels at 10.3 kDa. Protein was aliquoted and flash-frozen before storage at −80 °C until use. The protein concentration was measured using a Microplate BCA Protein Assay kit (Thermo Fisher Scientific) on a SpectraMax MiniMax 300 imaging cytometer using BSA as a standard curve. Typical yields of KaiB$_{RS}$ were 30–40 mg l$^{-1}$ of culture.

Buffer A$_{B-RS}$ comprised 50 mM Tris-base (pH 7.5), 250 mM NaCl, 10 mM imidazole, 2 mM TCEP and 10% glycerol (v/v). Buffer B$_{B-RS}$ comprised 50 mM Tris-base (pH 7.5), 250 mM NaCl, 500 mM imidazole, 2 mM TCEP and 10% glycerol (v/v). Dialysis$_{B-RS}$ comprised 50 mM Tris-base (pH 7.5), 250 mM NaCl, 2 mM TCEP and 10% glycerol (v/v). Buffer C$_{B-RS}$ comprised 50 mM Tris-base (pH 7.5), 50 mM NaCl, 2 mM TCEP and 10% glycerol (v/v). Buffer D$_{B-RS}$ comprised 50 mM Tris-base (pH 7.5), 2 mM TCEP and 10% glycerol (v/v). Buffer E$_{B-RS}$ comprised 50 mM Tris-base (pH 7.5), 1 M NaCl, 2 mM TCEP and 10% glycerol (v/v).

### Phylogenetic tree of KaiC

Protein sequences used in this study were identified in a multistep process. In the first step, a selection of sequences was identified using the BLASTP algorithm, utilizing a query based on the protein sequence for KaiC from *S. elongatus* (GenBank: WP_011242648.1)[38]. The query was run against NCBI's non-redundant protein database with the exclusion of models or uncultured and environmental sample sequences. A multiple sequence alignment of the selected 1,538 sequences was generated using MAFFT[39–41] (Supplementary Dataset 1). This alignment was used as input to generate an initial phylogenetic tree for KaiC with RAxML (v.8.2.9)[42] using the PROTGAMMALG model. The generated tree was then used to identify the emergence of KaiB to create the final tree that focused on systems containing either KaiBC or KaiABC. To do so, a BLASTP search was performed for each branch tip with the KaiB

sequence from *S. elongatus*, with the results restricted to the organism at which the branch tip was identified from. The observed branch point of emergence of KaiB agrees well with previous results in which it was shown that KaiB is mainly seen in non-archaea, non-proteobacteria[2].

To ensure the best possible sequence coverage, a BLASTP search using KaiB as query (GenBank: WP_011242647.1) was performed. The resulting sequences were then used to identify the organisms that they came from, which allowed us to create a list of organisms with an identified KaiB sequence. This list was then used to select for a subsequent BLASTP search using KaiC as query and therefore to identify only KaiC sequences for organisms that contain both KaiB and KaiC. A spot check was run to confirm that, for example, KaiA was indeed found among all cyanobacteria identified except for *Prochlorococcus marinus*. The obtained sequences were trimmed down to only include sequences with a sequence homology of 90% or less using CD-HIT[43] to arrive at a total of 401 sequences. For the calculation of the phylogenetic tree, RecA from *S. elongates* was added to serve as the outgroup. Sequences were aligned using MAFFT[39–41] (Supplementary Dataset 2) with the E-INS-I algorithm[44]. The multiple sequence alignment was then used as input for the phylogenetic tree calculation with IQ-TREE (v.1.6.beta5), using the LG-substitution matrix[45] with the freeRate model (using 10 categories; LG+R10)[46,47]. To enable determination of branch support, an aBayes test[48], a SH-aLRT test (20,000 bootstrap replicates[49]) and an ultrafast bootstrap (20,000 bootstrap replicates[50]) were performed (Supplementary Dataset 1; branch supports in order: SH-aLRT support (%)/aBayes support/ultrafast bootstrap support (%)).

### X-ray crystallography

KaiC$_{RS}$ and KaiC$_{RS}$-Δcoil crystals were obtained by sitting-drop vapour diffusion using a 96-well Intelli-Plate (102-0001-00, Art Robbins) at 291 K. Drops contained 0.5 μl crystallization solution, 0.5 μl protein at 10 mg ml$^{-1}$ in 20 mM MOPS pH 6.5, 50 mM NaCl, 2 mM TCEP, 5 mM MgCl$_2$, 3 mM ATP and 1 mM AMPPCP, and were equilibrated against 50 μl of the solution in the reservoir. The KaiC$_{RS}$ crystallization solution consisted of 200 mM magnesium chloride hexahydrate, 100 mM HEPES pH 7.5 and 30% (w/v) PEG 400. KaiC$_{RS}$-Δcoil crystals were grown using 200 mM ammonium acetate, 100 mM sodium citrate tribasic dihydrate pH 5.6 and 30% (w/v) PEG 4,000.

The PEG 400 in the KaiC$_{RS}$ crystallization solution acted as a cryoprotectant, whereas KaiC$_{RS}$-Δcoil crystals were cryoprotected in LV Cryo Oil (MiTeGen). Single crystals were cooled in liquid nitrogen, and X-ray diffraction images were collected at ALS beamline 8.2.1 at 100 K (data collection details are described in Extended Data Table 1). The data were indexed and integrated in iMosflm[51], and scaled and merged in Aimless[52].

To obtain a structural model of KaiC$_{RS}$, first the KaiC$_{RS}$-Δcoil structure was solved by molecular replacement in MRage[53] using the KaiC$_{RS}$ sequence (residues 1–490) as input to search for homologues in the PDB database. The initial KaiC$_{RS}$-Δcoil structure based on the KaiC$_{SE}$ (PDB: 1TF7 (ref. [36])) was manually rebuilt in Coot (v.0.9.81)[54] and refined in Phenix (v.1.20.1-4487)[55]. Finally, the KaiC$_{RS}$-Δcoil structure was used as the molecular replacement search model in Phaser[56] to solve the full-length KaiC$_{RS}$ structure.

The assigned space groups were validated in Zanuda[57], and the position of the asymmetric unit in the unit cell was standardized using Achesym[58]. KaiC$_{RS}$ coiled-coil registers were analysed using SamCC-Turbo (v.0.0.2) with the default socket cut-off value of 7.4 (ref. [59]). The images of protein structures were rendered using PyMOL (v.2.6.0)[60].

Tunnel detection and calculation were performed using CAVER 3.0.2 PyMOL plugin[61], with the minimum probe radius varying between 0.9 and 1.1. Default values were used for all other parameters. All atoms except waters were used in the calculation. The residue selection for starting point consisted of Glu62, Glu63 and ADP602. The catalytic position of the water in the CI domain was modelled from the crystal structure of the transition-state analogue-bound F$_1$-ATPase (PDB: 1w0j, water 2,064 from chain D[62]).

### Cryo-EM and image processing

For preparation of EM grids, 3–4 μl of 4.3 mg ml$^{-1}$ (per monomer concentration) of sample in 20 mM MOPS pH 6.50, 50 mM NaCl, 2 mM TCEP, 10 mM MgCl$_2$ and 2 mM ATP was applied to glow-discharged 1.2/1.3 400 mesh C-flat carbon-coated copper grids (Protochips). The grids were frozen using a Vitrobot Mark IV (ThermoFisher) at 4 °C and 95% humidity, with a blot time of 4 s. All datasets were collected on a Titan Krios operated at an acceleration voltage of 300 keV, with a GIF quantum energy filter (Gatan) and a GATAN K2 Summit direct electron detector controlled by SerialEM[63].

Inspection of the raw cryo-EM images revealed some heterogeneity in the relative orientations between individual hexamers of the dodecameric particles, presumably due to inherent flexibility in the coiled-coil regions, which limited the resolution to 3.3–3.4 Å. To obtain higher resolution reconstructions, the dodecamers were split and processed as individual hexamers, with C6 symmetry being applied throughout processing. To reconstitute the full dodecamer reconstruction, two copies of the hexamer reconstruction were overlaid on top of each other using the 'fit in map' function in Chimera[64] to fit one hexamer into the lower resolution end density of the other. The overlaid hexamers were then combined, creating a new map in which each voxel takes the value from the hexamer with highest absolute value.

For KaiC$_{RS}$-S413E/S414E alone, a dataset of approximately 2,500 movies was collected. The movies were recorded with a pixel size of 1.074 Å, including 70 frames and with an exposure rate of 1.31 e$^-$ per Å$^2$ per frame. Approximately 825,000 particles were picked, and after 2D classification, around 320,000 particles from good class averages were carried forward for further processing. The final measured resolution of the reconstruction was 2.9 Å (Extended Data Fig. 4a).

For the KaiC$_{RS}$-S413E/S414E:KaiB$_{RS}$ complex, a dataset of around 2,000 movies was collected. The movies were recorded with a pixel size of 1.023 Å, including 70 frames and with an exposure rate of 1.35 e$^-$ per Å$^2$ per frame. About 440,000 particles were picked, and after 2D classification around 190,000 particles from good class averages were carried forward for further processing. The final measured resolution of the reconstruction was 2.7 Å (Extended Data Fig. 4b).

All data processing was carried out using *cis*TEM (v.2.0.0)[65], and followed the workflow of motion correction, CTF parameter estimation, particle picking, 2D classification, ab initio 3D map generation, 3D refinement, 3D classification, per-particle CTF refinement and B-factor sharpening. The highest resolution of 3D refinement used was 4 Å for both reconstructions, and final resolutions were estimated using the *cis*TEM PartFSC and a threshold of 0.143.

To validate the combined dodecamer structure, we also processed both datasets as full D6 symmetric dodecamers (Extended Data Fig. 4c,d). This was accomplished by extracting the picked hexamers into a large box size and performing 2D classification with automatic centring. Clear dodecamer class averages were then selected and re-extracted from the original images, with picking coordinates that were adjusted by the translation required to match the centred class average. After this centring, duplicate picks were removed to obtain the final dodecamer particle stacks. These stacks were processed as described above, with the highest resolution of 3D refinement used as 4.25 Å.

In an attempt to find deviations from D6 symmetry, we also calculated reconstructions for both structures assuming C1 symmetry, starting from the ab initio 3D step. The resulting refined C1 structures did not exhibit detectable departures from D6 symmetry (Extended Data Fig. 4e). We therefore present symmetrized volumes as our final result.

The cryo-EM structures were built using the KaiC$_{RS}$ model obtained by X-ray crystallography and fold-switch-stabilized KaiB$_{TE}$ (PDB: 5JWO (ref. [26])) as starting points. The models were constructed using Coot

(v.0.9.81)[54], and refinement was carried out using Phenix (v.1.20.1-4487)[55].

## Preparation of unphosphorylated KaiC$_{RS}$

Purified KaiC$_{RS}$ (about 20 μM) from −80 °C was dialysed in 20 mM MOPS (pH 6.5), 50 mM NaCl, 2 mM TCEP, 10 mM MgCl$_2$ and 0.1 mM ADP overnight at 4 °C to remove glycerol and to replace ATP with ADP. The dialysed KaiC$_{RS}$ was then heated at 30 °C for 4 h to obtain fully unphosphorylated KaiC$_{RS}$ bound with ADP, and the sample was then passed through 0.22 μm Spin-X centrifuge tube filters (Corning). The sample was concentrated to a higher concentration (less than 100 μM) at 4 °C. The protein concentration was measured using a BCA assay.

## In vitro KaiBC$_{RS}$ reaction

**Kinetics of KaiC$_{RS}$ autophosphorylation in the presence and absence of KaiB$_{RS}$.** Unphosphorylated KaiC$_{RS}$ (3.5 μM, prepared as described above) in the presence or absence of KaiB$_{RS}$ (3.5 μM and 35 μM) was preincubated at 20, 25, 30 and 35 °C for 1 h in 20 mM MOPS (pH 6.5), 50 mM NaCl, 2 mM TCEP, 10 mM MgCl$_2$ and 0.1 mM ADP. The reactions were started by adding 3.9 mM ATP to obtain a final concentration of 4 mM nucleotide in the presence of 2 U ml$^{-1}$ pyruvate kinase (Millipore Sigma) and 10 mM phosphoenolpyruvate (Millipore Sigma) to regenerate ATP during the reaction. The reaction samples were sampled by hand at specific time points and mixed with an equal amount of loading dye (stock concentration of 0.1 M Tris-base (pH 7.5), 4% SDS, 0.2% bromophenol blue, 30% glycerol and 0.5 M 2-mercaptoethanol). The mixed samples were then stored at −20 °C until further use.

**Kinetics of KaiC$_{RS}$ autodephosphorylation in the presence and absence of KaiB$_{RS}$.** Purified KaiC$_{RS}$ (around 20 μM) was dialysed in reaction buffer containing 20 mM MOPS (pH 6.5), 50 mM NaCl, 2 mM TCEP, 10 mM MgCl$_2$ and 0.1 mM ADP overnight at 4 °C to remove glycerol and to generate KaiC$_{RS}$ bound with ADP. After dialysis at 4 °C, KaiC$_{RS}$ exists in two states: 50% unphosphorylated and 50% single phosphorylated at Ser413 (pSer413), which were confirmed by tandem mass spectrometry (data not shown). The autodephosphorylation reaction was started by adding KaiC$_{RS}$ or KaiC$_{RS}$ in the presence of KaiB$_{RS}$ (3.5 μM) into reaction buffer pre-equilibrated at 30 °C.

**Oscillation of KaiBC$_{RS}$.** Dialysed KaiC$_{RS}$ (3.5 μM) was preincubated at 35 °C for 30 min in 20 mM MOPS (pH 6.5), 50 mM NaCl, 2 mM TCEP, 10 mM MgCl$_2$ and 0.1 mM ADP in the presence or absence of KaiB$_{RS}$ (3.5 μM). The reactions were started by adding 4 mM ATP and reaction samples were collected at specific time points for 10% SDS–PAGE and HPLC analysis to identify phosphorylation state of KaiC$_{RS}$ and amount of nucleotide at each time point, respectively.

**Controlling ATP-to-ADP ratio to mimic daytime and night time.** KaiC$_{RS}$ was dialysed in reaction buffer containing 20 mM MOPS (pH 6.5), 50 mM NaCl, 2 mM TCEP, 10 mM MgCl$_2$ and 1 mM ATP overnight at 4 °C. To start the reaction as shown in Fig. 3c, KaiC$_{RS}$ (3.5 μM) in the absence or presence of KaiB$_{RS}$ (3.5 μM) was mixed with additional ATP (final 4 mM to mimic daytime), and the reaction samples (500 μl) were added into a D-Tube Dialyzer (midi 3.5 kDa cut-off, EMD Millipore) that was exchanged against 4 mM ATP buffer (400 ml). After the 12-h time point, the reaction samples were transferred into preincubated 25% ATP/ADP buffer (400 ml) that mimics the night time. After the 24-h time point, the same samples were changed into preincubated 4 mM ATP to mimic the daytime again.

To start the experiment as shown in Extended Data Fig. 6a, KaiC$_{RS}$ (35 μM) in the presence of 3 mM ATP in 20 mM MOPS (pH 6.5), 50 mM NaCl, 2 mM TCEP and 10 mM MgCl$_2$ was heated at 35 °C for 25 min to generate fully phosphorylated KaiC$_{RS}$. The KaiC$_{RS}$ sample was then diluted 10-fold into 25% ATP/ADP buffer pre-equilibrated at 30 °C to final concentration of KaiC$_{RS}$ (3.5 μM) and KaiB$_{RS}$ (3.5 μM or 35 μM).

The reaction samples (300 μl) were added into a D-Tube Dialyzer (midi 3.5 kDa cut-off, EMD Millipore) that was exchanged against 25% ATP/ADP buffer (300 ml).

During the reaction, the samples were gently shaken in a 30 °C incubator, and reaction samples were collected at specific time points for 10% SDS–PAGE and HPLC analysis to identify the phosphorylation state of KaiC$_{RS}$ and the amount of nucleotide at each time point, respectively.

The rationale for the ATP-to-ADP ratio at daytime and night time comes from two earlier literature reports. The change in ATP-to-ADP ratio at daytime and night time were directly measured in vivo in the strain *R. sphaeroides*[27], in which ATP is 2.0–2.4 mM during day and drops to 0.5–0.6 during night, and it is well known that the total nucleotide concentration stays constant. We chose the total nucleotide concentration of 4 mM in our in vitro work to be identical to the described in vitro experiments performed for the canonical KaiC$_{SE}$. Because of photosynthesis in daylight, virtually all nucleotide is ATP[33]. We note that a slightly higher amount of ATP will not affect our results, as the affinity of KaiC$_{RS}$ for ATP is higher than for ADP.

## Separation of unphosphorylated, single and double phosphorylated KaiC$_{RS}$ by SDS–PAGE

Unphosphorylated, single phosphorylated and double phosphorylated KaiC$_{RS}$ were separated by 10% SDS–PAGE with 37.5:1 acrylamide:bis-acrylamide (Bio-Rad), 18 cm × 16 cm × 1 mm Tris-HCl gel with 1× Tris-glycine SDS running buffer (Invitrogen). The samples were heated at 95 °C for 3 min, and 400 ng of material was loaded onto the Tris-HCl gel. The gel was run with a constant current of 35 mA, 150 W, and the voltage was greater than 700 V for 5.5 h in a cold room, with a water bath set to 12 °C using a Hoefer SE600 electrophoresis unit.

Unphosphorylated and phosphorylated KaiC$_{RS}$-Δcoil were separated by Zn$^{2+}$ Phos-tag SDS–PAGE with 10% acrylamide gel containing 50 μM Phos-tag acrylamide (Wako). The gel was run with a constant current of 30 mA for 5 h 30 min in a cold-room, with 1 μg per well protein samples pre-heated at 95 °C for 3 min.

The gels were stained overnight at room temperature with Instant-Blue protein gel stain (Expedeon) with gentle shaking and destained with distilled water until bands were clearly visible. The gels were imaged on a ChemiDoc Imager (Bio-Rad), and Image Lab software (Bio-Rad) was used for analysis.

## Statistics and reproducibility for gel electrophoresis

Data shown in main text figures and Extended Data figures are representative SDS–PAGE gels for at least three independent biological replicates ($n$ = 3), except for experiments presented in Fig. 3a,c, which were performed in duplicate.

## Oligomerization state of KaiC$_{RS}$ and KaiB$_{RS}$

**Gel-filtration chromatography.** KaiC$_{RS}$, unphosphorylated KaiC$_{RS}$ and all KaiC$_{RS}$ mutants with a concentration of around 40–80 μM were loaded with a flow rate of 0.2 ml min$^{-1}$ onto a prepacked Superdex-200 10/300 GL (GE Healthcare) pre-equilibrated with 20 mM MOPS (pH 6.5), 50 mM NaCl, 2 mM TCEP, 10 mM MgCl$_2$ and 1 mM ATP (0.1 mM ADP for unphosphorylated KaiC$_{RS}$) at 4 °C using an ÄKTA Pure system (GE Healthcare). KaiB$_{RS}$ (0.5 mM) was loaded onto Superdex-75 10/300 GL (GE Healthcare) pre-equilibrated with 20 mM MOPS (pH 6.5), 50 mM NaCl and 2 mM TCEP at 4 °C. The eluate was collected in fractions of 1 ml each and subjected to SDS–PAGE analysis. A standard curve (that is, molecular weight versus elution time) was determined for the column using molecular weight protein standards (Bio-Rad) run in the same buffer and flow rate. The protein standard mixture contained thyroglobulin (670 kDa), γ-globulin (158 kDa), ovalbumin (44 kDa), myoglobin (17 kDa) and vitamin B$_{12}$ (1.35 kDa).

**Analytical ultracentrifugation.** Sedimentation velocity centrifugation experiments were run at 50,000 r.p.m. (for KaiB$_{RS}$) and 30,000

r.p.m. (for KaiC$_{RS}$ wild-type and mutants and KaiC$_{SE}$), with continuous scans from 5.8 to 7.3 cm at 0.005 cm intervals at 20 °C on a Beckman Optima XL-A (Beckman-Coulter) equipped with absorption optics and a four-hole An60Ti rotor. Measurements were set up at 280 nm (for KaiB$_{RS}$) and 295 nm (for KaiC$_{RS}$ and KaiC$_{SE}$) to avoid interference from ATP. The software package SEDFIT (v.14.1) was used for data evaluation[66]. KaiC$_{RS}$ and KaiC$_{RS}$-Δcoil (100 μM) were prepared in 20 mM MOPS (pH 6.5), 50 mM NaCl, 2 mM TCEP, 10 mM MgCl$_2$ and 1 mM ATP. KaiC$_{SE}$ (100 μM) was prepared in 20 mM MOPS (pH 8.0), 150 mM NaCl, 2 mM TCEP, 5 mM MgCl$_2$, and 1 mM ATP. KaiB$_{RS}$ (500 μM) was prepared in 20 mM MOPS (pH 6.5), 50 mM NaCl and 2 mM TCEP.

## ATPase activity

Purified KaiC$_{RS}$ (both wild-type and mutant forms (around 20 μM)) and KaiB$_{RS}$ (about 90 μM) were dialysed in 20 mM MOPS (pH 6.5), 50 mM NaCl, 2 mM TCEP, 10 mM MgCl$_2$ and 1 mM ATP (reaction buffer) overnight at 4 °C. The samples were passed through 0.22 μm Spin-X centrifuge tube filters, and concentrations were measured using a BCA assay before setting up the reactions. Typical KaiC$_{RS}$ or KaiBC$_{RS}$ reactions contained 3.5 μM KaiC$_{RS}$ (wild type and mutants) and 3.5 μM KaiB$_{RS}$ in reaction buffer with a final concentration of 4 mM ATP. The samples were incubated at the indicated temperatures and were sampled by hand at specific time points. Next 10 μl of sample was quenched with 10 μl of 10% trichloroacetic acid (Millipore Sigma), and the mixture was passed through a 0.22 μm Spin-X centrifuge tube filter to remove the precipitated protein. The flow through sample was then re-adjusted to pH 6.2 for nucleotide separation by adding 10 μl of 0.75 M HEPES, pH 8.0. The final samples were kept at −20 °C until HPLC analysis.

Three microlitres of each sample were injected with a high-precision autosampler (injection error of <0.1 μl, resulting in a maximum systemic error of about 6%) to a reverse-phase HPLC instrument with an ACE 5 μm particle size, C18-AR and 100 Å pore size column (Advanced Chromatography Technologies). The instrument was pre-equilibrated with 100 mM potassium phosphate pH 6.2 with a flow rate of 0.4 ml min$^{-1}$. Using pure nucleotide samples, the retention times of ATP, ADP and AMP were determined to be 2.6, 3.1 and 4.4 min, respectively. The concentration of each nucleotide was calculated from the relative ratio of the peak areas and the total nucleotide concentration. To determine ATPase activity rates, the observed rate constants were determined from at least five data points for each temperature using initial rate analysis and least-squares linear regression (Extended Data Figs. 6c,d and 8b–d). The mean values and uncertainties (s.d.) shown in Fig. 3d, Extended Data Figs. 6 and 8 were derived from three replicate experiments. KaleidaGraph (v.4.5.3; Synergy) was used for data analysis and plotting.

## Nucleotide exchange

KaiC$_{RS}$ (wild type or mutants) and KaiB$_{RS}$ were dialysed into 20 mM MOPS (pH 6.5), 50 mM NaCl, 2 mM TCEP, 10 mM MgCl$_2$ and 50 μM ATP overnight at 4 °C. The samples were passed through 0.22 μm Spin-X centrifuge tube filters, and the protein concentration was measured using a BCA assay. The reaction contained 3.5 μM of KaiC$_{RS}$-S413E and/or 35 μM of KaiB$_{RS}$, and samples were incubated at 20, 25, 30 and 35 °C for 16–24 h in the presence of an ATP-recycling system. The reactions were started by adding 250 μM of mant-ATP (Jena Bioscience). The spectrum was measured using the fluorescence energy transfer from tryptophan residues in KaiC$_{RS}$ to mant-ATP by exciting the sample at 290 nm (2.5 nm bandwidth) and collecting the emission intensity from 320 nm to 550 nm (5 nm bandwidth) in increments of 2 nm. To measure the nucleotide exchange rate, the maximum change in fluorescence intensity at 440 nm (Δ$F_{440\,nm}$) was followed for a total time of 1,800 s in 15 s increments with anti-photobleaching mode on FluoroMax-4 spectrofluorometer (Horiba Scientific) equipped with a water bath to control the temperature. There are two tryptophan residues within 5 Å from the nucleotide-binding site in the KaiCII$_{RS}$ domain and no

tryptophan residue close to the nucleotide-binding site in the KaiCI$_{RS}$ domain, so the nucleotide exchange observed in the experiments are for the KaiCII$_{RS}$ domain. To ensure that the exchange rate observed in the experiments are from nucleotide exchange in the KaiCII$_{RS}$ domain, KaiCI$_{RS}$ (which only contains the CI domain) was tested; no change in fluorescence was observed following the addition of mant-ATP (Extended Data Fig. 8h).

The experiments for KaiC$_{SE}$ alone and with KaiC$_{SE}$ mixed in and for KaiA$_{SE}$ were performed in a similar way, except that KaiC$_{SE}$ and KaiA$_{SE}$ were dialysed in 20 mM MOPS (pH 8.0), 150 mM NaCl, 2 mM TCEP, 10 mM MgCl$_2$ and 50 μM ATP overnight at 4 °C. KaiA$_{SE}$ was incubated with KaiC$_{SE}$ for 1 h at 30 °C before adding 250 μM mant-ATP.

For the nucleotide preference experiment, KaiC$_{RS}$-S413E/S414E and KaiC$_{RS}$-S413A/S414A were dialysed in 20 mM MOPS (pH 6.5), 50 mM NaCl, 2 mM TCEP, 10 mM MgCl$_2$ and 20 μM ADP overnight at 4 °C. KaiC$_{RS}$-S413E/S414E or KaiC$_{RS}$-S413A/S414A (3.5 μM) was first mixed with mant-ATPγS or mant-ADP (150 μM), and the kinetic trace at 440 nm was recorded at 30 °C. After the fluorescence trace at 440 nm reached a plateau, which indicates that the nucleotide analogue was fully bound to the protein, a 27-fold excess of ATP (4 mM) was added to displace the bound nucleotide analogue, and the decay of fluorescence intensity was recorded at 440 nm at 30 °C. The experiments were run in triplicate, and results were averaged and fitted to a single exponential decay.

Analysis was performed by fitting individual traces to an exponential equation using KinTek Explorer software[67,68], and error bars denote the standard errors as obtained from triplicate experiments. KaleidaGraph (v.4.5.3; Synergy) was used for data plotting.

## $^{32}$P-ATP radioactive labelling and experiment

$^{32}$P-labelled KaiC$_{RS}$ was prepared by mixing unphosphorylated KaiC$_{RS}$ (10 μM) with 0.46 μM [γ-$^{32}$P]ATP (3,000 Ci mmol$^{-1}$, PerkinElmer) and 500 μM ATP in 20 mM MOPS (pH 6.5), 50 mM NaCl, 2 mM TCEP and 10 mM MgCl$_2$ at 35 °C for 30 min and then immediately switched to 4 °C to prevent dephosphorylation of KaiC$_{RS}$. The $^{32}$P-labelled KaiC$_{RS}$ sample was passed through Zeba spin desalting columns (Thermo Fisher Scientific) pre-equilibrated with 20 mM MOPS (pH 6.5), 50 mM NaCl, 2 mM TCEP and 10 mM MgCl$_2$ twice at 4 °C. The sample was incubated with buffer containing 1 mM ADP overnight at 4 °C to obtain $^{32}$P-labelled KaiC$_{RS}$ bound with ADP. The sample was passed through a final Zeba desalting column pre-equilibrated with 20 mM MOPS (pH 6.5), 50 mM NaCl, 2 mM TCEP and 10 mM MgCl$_2$ at 4 °C, and the solution was then incubated with 8 mM ADP in the presence or absence of KaiB$_{RS}$ at 4 °C for 1 h. The samples were then diluted in 20 mM MOPS (pH 6.5), 50 mM NaCl, 2 mM TCEP and 10 mM MgCl$_2$ preincubated at 30 °C to obtain a final concentration of $^{32}$P-labelled KaiC$_{RS}$ (5 μM), KaiB$_{RS}$ (20 μM) and ADP (4 mM). The reactions were incubated at 30 °C, and at different time points, aliquots were taken (1.5 μl). The reactions were stopped by adding 1.5 μl Laemmli sample buffer (62.5 mM Tris (pH 6.8), 2% SDS, 25% glycerol and 0.01% bromophenol blue) supplemented with 5% (v/v) 2-mercaptoethanol.

The samples were spotted onto a TLC plate (PEI-cellulose F plates, Merck) and quickly dried with a blow-dryer for 30 s. The TLC plates were run first with distilled water as a mobile phase. After TLC plates were completely dried, 0.75 M KH$_2$PO$_4$ was used as the mobile phase to separate $^{32}$P-labelled KaiC$_{RS}$, [γ-$^{32}$P]ATP and inorganic phosphate ($^{32}$P), as previously shown[29]. The phosphor-screens were scanned on an Amersham Typhoon (GE Healthcare) at a resolution of 100 μm. ImageQuant TL 7.0 software was used for analysis.

## Fluorescence anisotropy competition

Fluorescence anisotropy competition experiments were carried out using a FluoroMax-4 spectrofluorometer (Horiba Scientific) at 30 °C. Excitation and emission wavelengths for KaiB$_{RS}$ labelled with 6-iodoacetamidofluorescein (6-IAF, Thermo Fisher Scientific) at Cys29 were set at 492 nm (5 nm bandwidth) and 520 nm (5 nm bandwidth),

respectively, with fixed G factor (G factor of $KaiB_{RS}$–6IAF alone) to eliminate instrumental bias. The average anisotropy and standard error were calculated from ten replicate measurements.

$KaiB_{RS}$–6IAF was prepared by mixing degassed $KaiB_{RS}$ (100 μM) with 20-fold excess of 6-IAF (stock 10 mM in 50% DMSO) in 20 mM Tris-base (pH 7.0), 50 mM NaCl and 1 mM TCEP (degassed). The reaction was incubated at room temperature for 4 h and dialysed against 20 mM Tris-base (pH 7.0), 50 mM NaCl and 1 mM TCEP in a 3.5 kDa dialysis cassette overnight at 4 °C to remove unreacted 6-IAF and small amounts of DMSO. The crosslinked sample was passed through a 0.22 μm Spin-X centrifuge tube filter and loaded onto Superdex-75 10/300 GL pre-equilibrated with 20 mM MOPS (pH 7.0), 50 mM NaCl and 1 mM TCEP with a 0.2 ml min$^{-1}$ flow rate to remove leftover unreacted 6-IAF. The sample was aliquoted and flash-frozen in liquid nitrogen and stored at −80 °C until use. All the crosslinked reactions were performed in the dark.

For the fluorescence anisotropy competitive binding experiment, $KaiB_{RS}$–6IAF (0.2–0.4 μM) was first incubated with wild-type or mutant $KaiC_{RS}$ (1 μM, 60% increase in anisotropy in comparison to $KaiB_{RS}$–6IAF alone) in 20 mM MOPS (pH 6.5), 50 mM NaCl, 2 mM TCEP and 10 mM $MgCl_2$ in the presence of 4 mM ADP or 4 mM ATP with an ATP-recycling system (2 U ml$^{-1}$ pyruvate kinase and 10 mM phosphoenolpyruvate), then unlabelled $KaiB_{RS}$ (0–50 μM) was added to the samples. The samples were incubated at 30 °C for 4 h or 12 h before measurement.

The decrease in fluorescence anisotropy (FA) versus concentration of unlabelled $KaiB_{RS}$ was fitted to equation (1) using the Levenberg–Marquardt nonlinear fitting algorithm included in KaleidaGraph (Synergy Software) to obtain the half-maximum inhibitory concentration ($IC_{50}$) value. The $K_d$ value can then be calculated from the $IC_{50}$ value using equation (2) as previously described[69].

$$FA = m_1 + \frac{(m_2 - m_1)}{1 + 10^{\log x - \log(IC_{50})}} \tag{1}$$

$$K_d = \frac{IC_{50}}{1 + \frac{[\text{labelled KaiB}_{RS}]}{K_d}} \tag{2}$$

## Pull-down assays

Pull-down assays probe the interaction between $KaiC_{RS}$ (wild type and mutants) with $KaiB_{RS}$ ($His_6$-MBP-TEV-$KaiB_{RS}$). The $His_6$-MBP-TEV-$KaiB_{RS}$ was expressed and purified as described above but without the TEV protease cleavage step. Wild-type or mutant $KaiC_{RS}$ (3.5 μM) was mixed with $KaiB_{RS}$-tag (3.5 μM) in 20 mM MOPS (pH 6.5), 50 mM NaCl, 2 mM TCEP and 10 mM $MgCl_2$ in the presence of 4 mM ADP or 4 mM ATP with an ATP-recycling system in a final volume of 400 μl. The samples were incubated at 25 °C for 4 h or 24 h before loading onto a 500 μl spin column with 200 μl (prepared from 400 μl of 50% slurry) Talon beads (Takara) pre-equilibrated with sample buffer. The samples were incubated with the Talon beads for 30 min with gentle shaking, after which the flow through was collected by gravity into 1 ml Eppendorf tubes. The beads were washed three times with 400 μl sample buffer by gravity, then the samples were eluted with 200 μl of 0.5 M imidazole buffer by centrifugation at 1,000g for 1 min. The protein mixture, flow through, wash and eluted samples were run on a Bis-Tris 4–12% gradient SDS–PAGE gel with a molecular weight marker. The gels were stained with Coomassie blue and were imaged on a ChemiDoc Imager (Bio-Rad). Image Lab Software (Bio-Rad) was used for analysis.

The following control experiments were performed: (1) fusion $KaiB_{RS}$ protein in the absence of $KaiC_{RS}$; and (2) $KaiC_{RS}$ in the absence of fusion $KaiB_{RS}$. All the fusion $KaiB_{RS}$ proteins came out only in the elution buffer in the first control experiment, which indicated that fusion $KaiB_{RS}$ binds to Talon beads and the amount of $KaiB_{RS}$ used did not overload the column. All $KaiC_{RS}$ protein came out in the flow through in the second control experiment, which indicated there is no specific binding between $KaiC_{RS}$ and the Talon beads.

## Thermofluor assay

$KaiC_{RS}$ and SYPRO Orange (Thermo Fisher Scientific) were used at final concentration of 3 μM and 10×, respectively. The experiments were carried out in 20 mM MOPS (pH 6.5), 50 mM NaCl, 2 mM TCEP, 10 mM $MgCl_2$ and 4 mM ADP or ATP. The samples were prepared to a final volume of 20 μl in a MicroAmp Fast Optical 96-well reaction plate (Applied Biosystems Life Technologies), and the plate was sealed with Axygen UltraClear sealing film (Corning). The assay plate was run in a StepOne Real-Time PCR instrument (Applied Biosystems Life Technologies) with melt curve set up. The temperature was continuously increased from 25 °C to 95 °C by 0.3 °C every 15 s. The data were fit with nonlinear fitting in KaleidaGraph (Synergy Software) to a Boltzmann sigmoidal curve (equation (3)).

$$Y = \text{Bottom} + \frac{(\text{Top} - \text{Bottom})}{1 + \exp\left(\frac{T_m - T}{c}\right)} \tag{3}$$

where $Y$ is the fluorescence intensity at temperature $T$, $T$ is the temperature in degrees Celsius, Bottom is the baseline fluorescence at low temperature, Top is the maximum fluorescence at the top of the truncated data, $c$ is the slope or steepness of the curve, and $T_m$ is the melting temperature of the protein.

## Reporting summary

Further information on research design is available in the Nature Portfolio Reporting Summary linked to this article.

## Data availability

Structure factors and refined models obtained using X-ray crystallography have been deposited into PDB under accession codes 8DBA (wild-type $KaiC_{RS}$) and 8DB3 ($KaiC_{RS}$-Δcoil). Cryo-EM maps and refined models have been deposited into the Electron Microscopy Data Bank (EMDB) and PDB, respectively. The composite map and model for the $KaiC_{RS}$-S413E/S414E dodecamer reconstruction are submitted under entries EMD-29505 and 8FWI, respectively. The composite map and model for the KaiCRS-S413E/S414E–$KaiB_{RS}$ dodecamer reconstruction are submitted under entries EMD-29506 and 8FWJ, respectively. The focused $KaiC_{RS}$-S413E/S414E hexamer refinement map is available under accession EMD-29507 and the focused $KaiC_{RS}$-S413E/S414E–$KaiB_{RS}$ hexamer refinement map is available under accession EMD-29508. The full $KaiC_{RS}$-S413E/S414E dodecamer refinement is available under accession EMD-29509 and the full $KaiC_{RS}$-S413E/S414E–$KaiB_{RS}$ dodecamer refinement is available under accession EMD-29510. Other datasets used are all publicly available in public community or discipline-specific repositories (for example, PDB identifiers 5JWQ, 1W0J, 1TF7 and 7S65). The accession codes for protein sequences, sequence alignments and phylogeny are listed in Supplementary Datasets 1 and 2.

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

**Acknowledgements** D.K. and N.G. are supported by the Howard Hughes Medical Institute (HHMI). We would like to thank M. Rigney for assistance with negative-stain data collection at the Brandeis University Electron Microscopy Facility, and Z. Yu and staff at the Janelia Research Campus cryo-EM facility for advice and assistance with data collection. The Berkeley Center for Structural Biology is supported in part by HHMI. Beamline 8.2.1 of the Advanced Light Source, a US DOE Office of Science User Facility under contract no. DE-AC02-05CH11231, is supported in part by the ALS-ENABLE programme funded by the National Institutes of Health, National Institute of General Medical Sciences, grant P30 GM124169-01. Mass spectral data were obtained at the University of Massachusetts Mass Spectrometry Core Facility (RRID:SCR_019063).

**Author contributions** W.P., R.A.P.P. and D.K. conceived the project and designed experiments. W.P. performed and analysed all biochemical data. W.P. and R.A.P.P. set up the crystal trays. R.A.P.P. collected and analysed the X-ray crystallographic data. W.P. prepared the samples for the cryo-EM studies and collected negative-stain images to screen for optimal sample conditions. T.G. collected and processed all cryo-EM data and reconstructed the cryo-EM maps under supervision of N.G. R.A.P.P. built and interpreted the structural models. W.P. and N.B. performed and analysed experiments with radioactively labelled KaiC. M.H. built the KaiC phylogeny tree. W.P., R.O. and D.K. wrote the paper. All authors commented on the manuscript and contributed to data interpretation.

**Competing interests** D.K. is co-founder of Relay Therapeutics and MOMA Therapeutics. The remaining authors declare no competing interests.

**Additional information**
**Correspondence and requests for materials** should be addressed to Dorothee Kern.

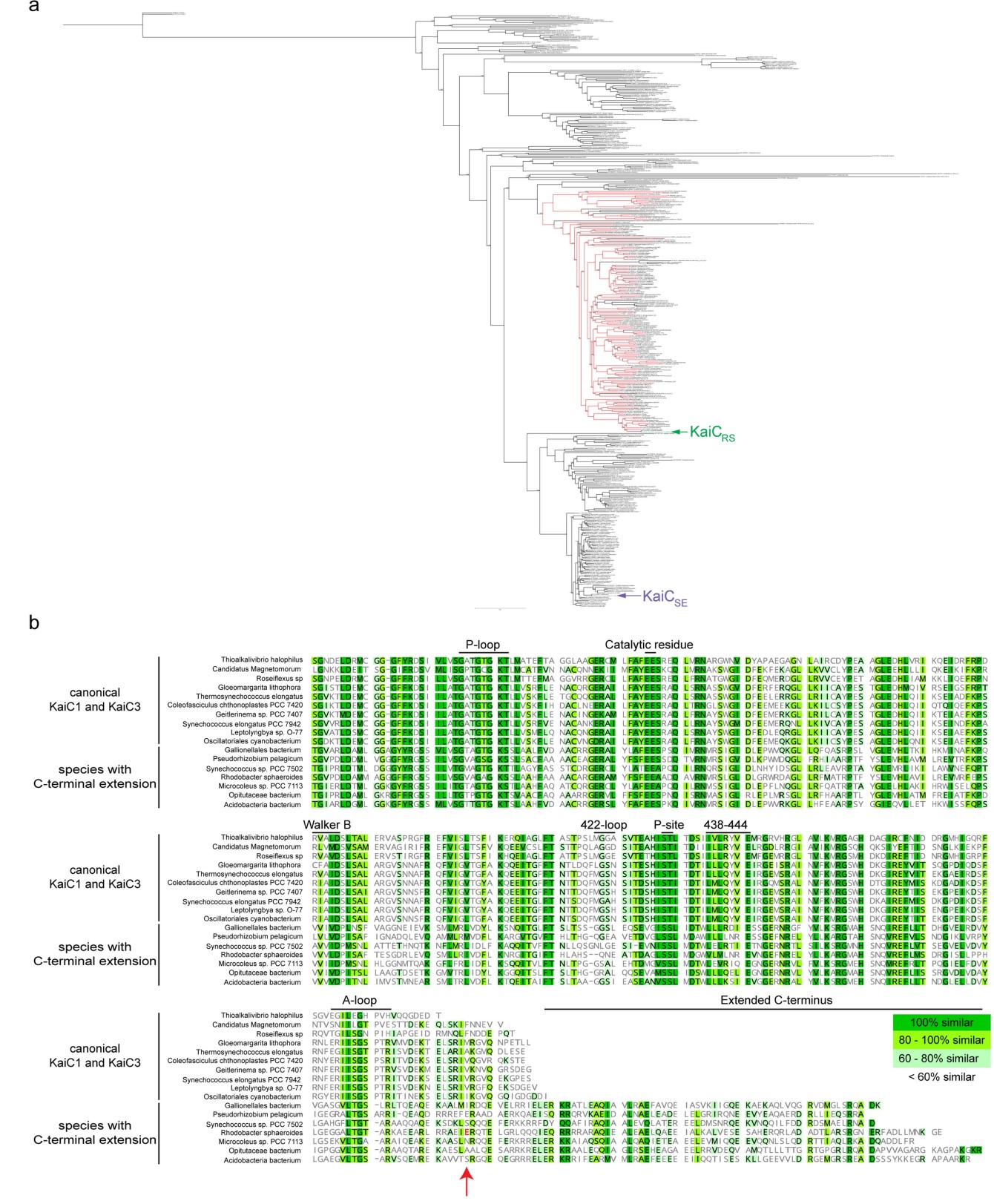

**Extended Data Fig. 1 | Evolution of *kaiC* and sequence alignment of *kaiC* subgroups. (a)** Phylogenetic tree of *kaiC* homologs, where *kaiC* genes that have an approximately 50 amino acids C-terminal extension are labeled in red. *Rhodobacter sphaeroides* strain KD131 studied here and *Synechococcus elongatus* PCC 7942 (widely studied in the literature) are highlighted in green and pink, respectively. The accession code and organism are shown at the tip of the branches, the numbers at each node represent the aBayes bootstrap values[48], and the legend for branch length is shown (see also Supplementary Datasets 1 and 2). **(b)** A sequence alignment of the CII domain of the *kaiC* subgroups annotated with its sequence similarity. Residue Glu490, the position where the stop codon was introduced in the truncated KaiC$_{RS}$-Δcoil construct, is shown in red and marked with an arrow.

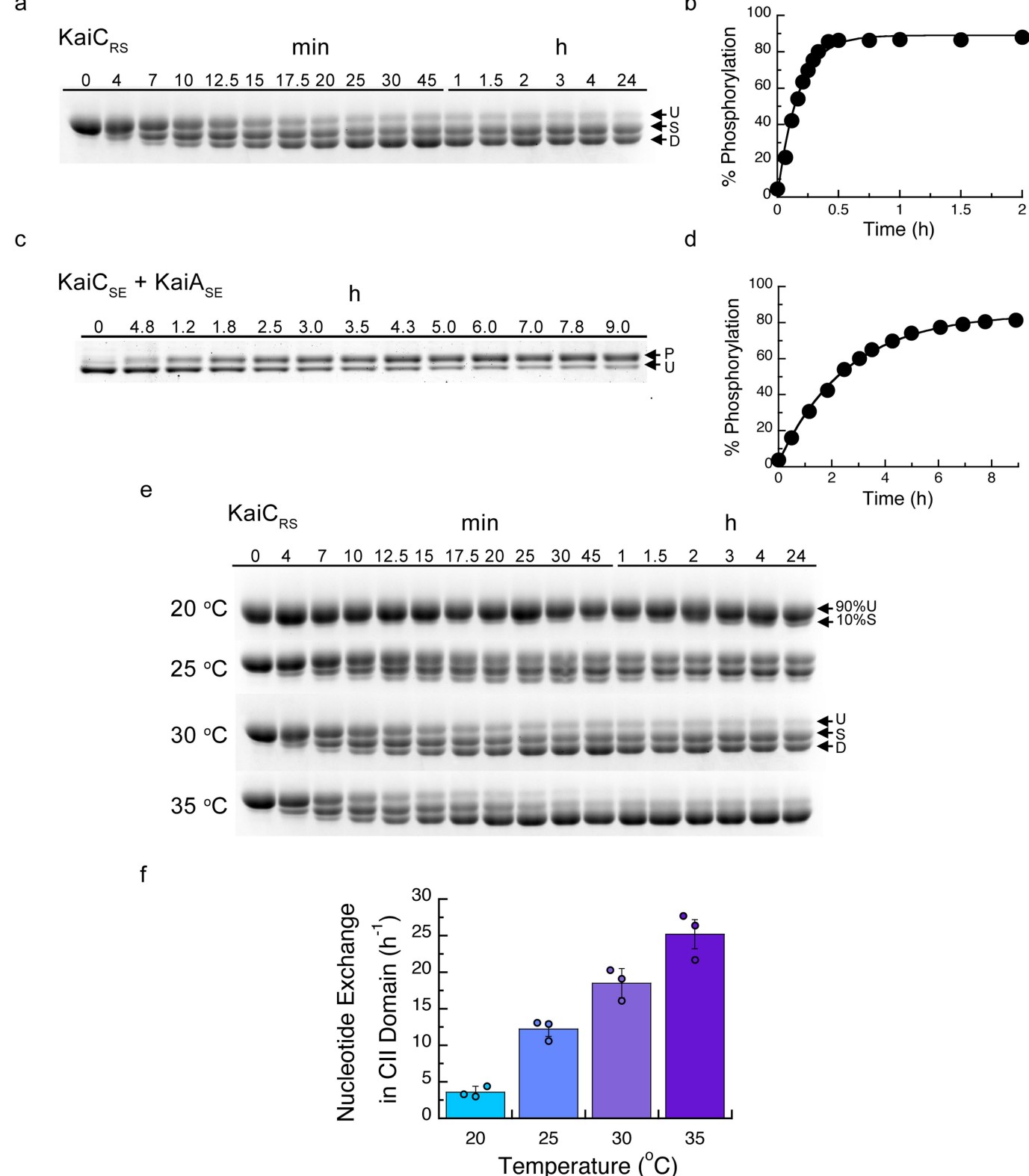

**Extended Data Fig. 2** | See next page for caption.

**Extended Data Fig. 2 | Auto-phosphorylation and nucleotide exchange rates of KaiC. (a)** 10% SDS-PAGE gel of 3.5 µM KaiC$_{RS}$ in the presence of 4 mM ATP and using an ATP-recycling system at 30 °C. U, S, and D represent unphosphorylated, singly, and doubly phosphorylated KaiC$_{RS}$, respectively. **(b)** Densitometric analysis of auto-phosphorylation (single + double phosphorylation) from panel (a) over time yields a rate of 6.5 ± 1.0 h$^{-1}$. **(c)** 6.5% SDS-PAGE gel of 3.5 µM KaiC$_{SE}$ in the presence of 1.2 µM KaiA$_{SE}$ and 4 mM ATP at 30 °C. U and P represent unphosphorylated and phosphorylated KaiC$_{SE}$, respectively. **(d)** Densitometric analysis of auto-phosphorylation of KaiC$_{SE}$ activated by KaiA$_{SE}$ (panel (c)) shows a rate of 0.40 ± 0.02 h$^{-1}$ and is substantially slower than for KaiC$_{RS}$. The standard deviation for parameters in (b) and (d) were obtained from data fitting. **(e)** 10% SDS-PAGE gels for experiments with 3.5 µM KaiC$_{RS}$ in the presence of 4 mM ATP and using an ATP-recycling system between 20 and 35 °C show that the level of phosphorylation increases with temperature. U, S, and D represent unphosphorylated, singly, and doubly phosphorylated KaiC$_{RS}$, respectively. For gel source data in (a), (c), and (e), see Supplementary Figure 2. **(f)** Bar graphs indicating the nucleotide exchange rate in the CII domain of KaiC$_{RS}$ incubated with 50 µM ATP in the presence of an ATP-recycling system, and then mixed with 250 µM mant-ATP. An increase in fluorescence intensity at 440 nm was recorded and the single-exponential time traces were fitted to obtain the exchange rate constants: 3.6 ± 0.8 h$^{-1}$ (20 °C), 12.2 ± 1.0 h$^{-1}$ (25 °C), 18.5 ± 1.5 h$^{-1}$ (30 °C), and 25.2 ± 0.2 h$^{-1}$ (35 °C). Experiments were performed in triplicate and data are presented as mean values ± s.d.

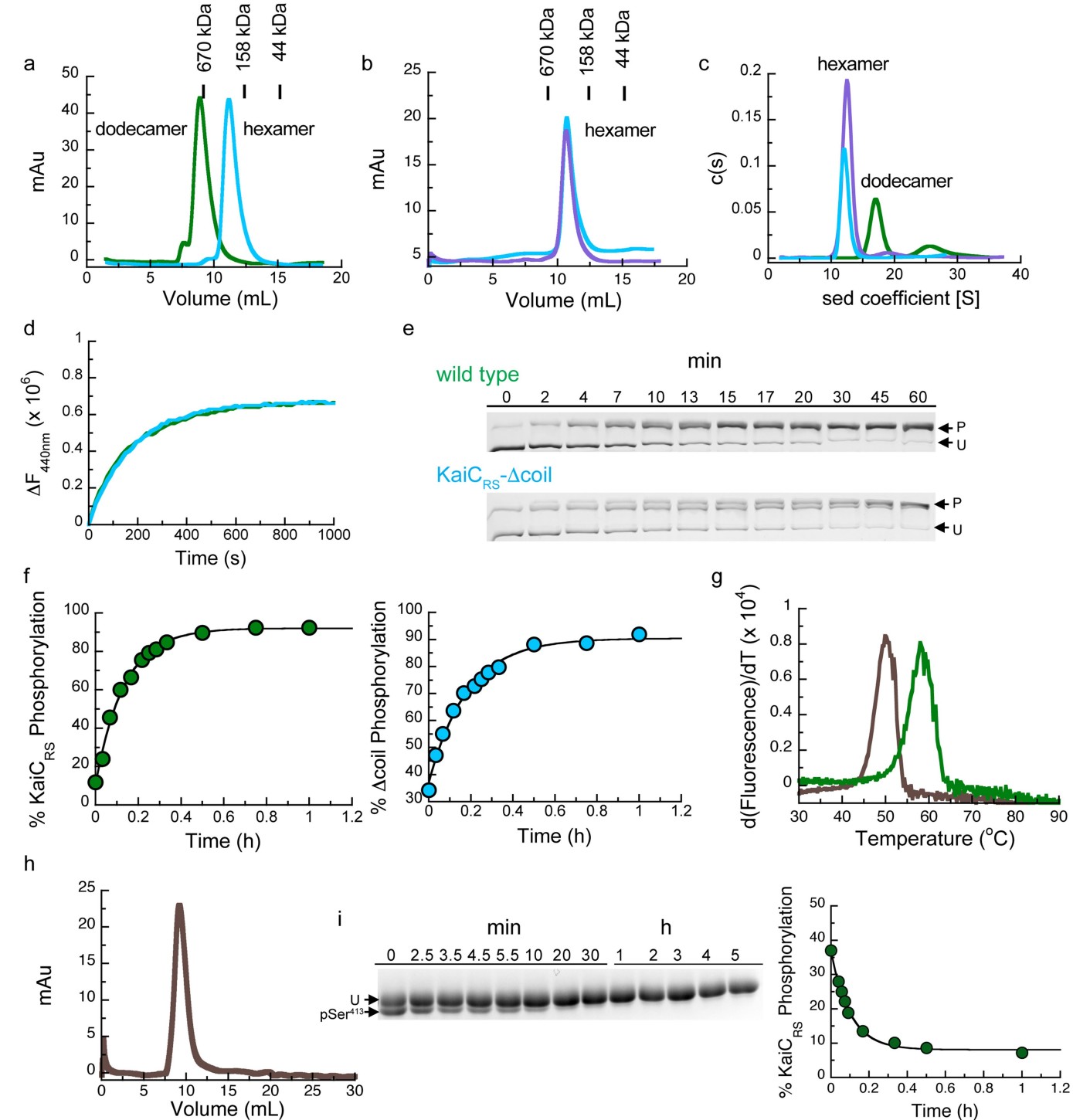

**Extended Data Fig. 3** | See next page for caption.

**Extended Data Fig. 3 | Oligomeric states of KaiC$_{RS}$ and effect of coiled-coil domain on rates of nucleotide exchange and auto-phosphorylation.**
**(a)** Oligomerization analysis of KaiC$_{RS}$ (dodecamer, green line) and truncated KaiC$_{RS}$-$\Delta$coil (hexamer, cyan line) by analytical gel-filtration chromatography. The protein size markers are indicated at the top. **(b)** Comparison of the elution profiles of KaiC$_{RS}$-$\Delta$coil (cyan line) and KaiC$_{SE}$ (purple line) from size-exclusion chromatography shows a hexameric state for both KaiC$_{RS}$-$\Delta$coil and KaiC$_{SE}$.
**(c)** Oligomeric states of KaiC$_{RS}$ (dodecamer, green line), KaiC$_{RS}$-$\Delta$coil (hexamer, cyan line), and KaiC$_{SE}$ (hexamer, purple line) were also measured by analytical ultracentrifugation (sedimentation velocity at 30,000 rpm and 20 °C) and the results agree with the data shown in panels (a) and (b). The graph in panel (c) represents the sedimentation coefficient distribution [c(s)]. **(d)** The change in fluorescence at 440 nm ($\Delta F_{440nm}$) represents the nucleotide exchange between ATP and mant-ATP at 30 °C for KaiC$_{RS}$ (green trace, 18.0 ± 1.5 h$^{-1}$) and KaiC$_{RS}$-$\Delta$coil (cyan trace, 19.1 ± 0.8 h$^{-1}$). Representative traces are shown and the fitted parameters (mean ± s.d.) were obtained from three replicate measurements.
**(e)** Zn$^{2+}$ Phos-tag$^{TM}$ SDS-PAGE gel shows the level of phosphorylation over time of KaiC$_{RS}$ (upper gel) and KaiC$_{RS}$-$\Delta$coil (lower gel) at 35 °C. P and U represent

phosphorylated and unphosphorylated protein, respectively. **(f)** Phosphorylation level over time of KaiC$_{RS}$ (green circles, 7.4 ± 0.3 h$^{-1}$) and KaiC$_{RS}$-$\Delta$coil (cyan circles, 5.5 ± 0.4 h$^{-1}$) analyzed by densitometric analysis of Zn$^{2+}$ Phos-tag$^{TM}$ SDS-PAGE gel in (e). **(g)** First derivative of thermal-stability curves measured for unphosphorylated KaiC$_{RS}$ bound with ADP (brown line) and phosphorylated KaiC$_{RS}$ bound with ATP (green line). The extracted temperatures of denaturation are 50 °C (unphosphorylated KaiC$_{RS}$ in the presence of 1 mM ADP) and 58 °C (phosphorylated KaiC$_{RS}$ in the presence of 1 mM ATP), respectively. **(h)** Dodecameric state of unphosphorylated KaiC$_{RS}$ (40 µM) bound with ADP measured by size-exclusion chromatography. **(i)** SDS-PAGE gel shows dephosphorylation of Ser413 over time at 30 °C in the presence of 4 mM ADP (U and pSer413 represent unphosphorylated and Ser413-phosphorylated KaiC$_{RS}$, respectively) with the corresponding kinetics shown in the right panel (confirmed by MS/MS) with a rate constant of 11.5 ± 0.8 h$^{-1}$. This result suggests that the coiled-coil domain promotes KaiC$_{RS}$ dephosphorylation. For gel source data in (e) and (i), see Supplementary Figure 2. The standard deviation for parameters in (f) and (i) were obtained from data fitting.

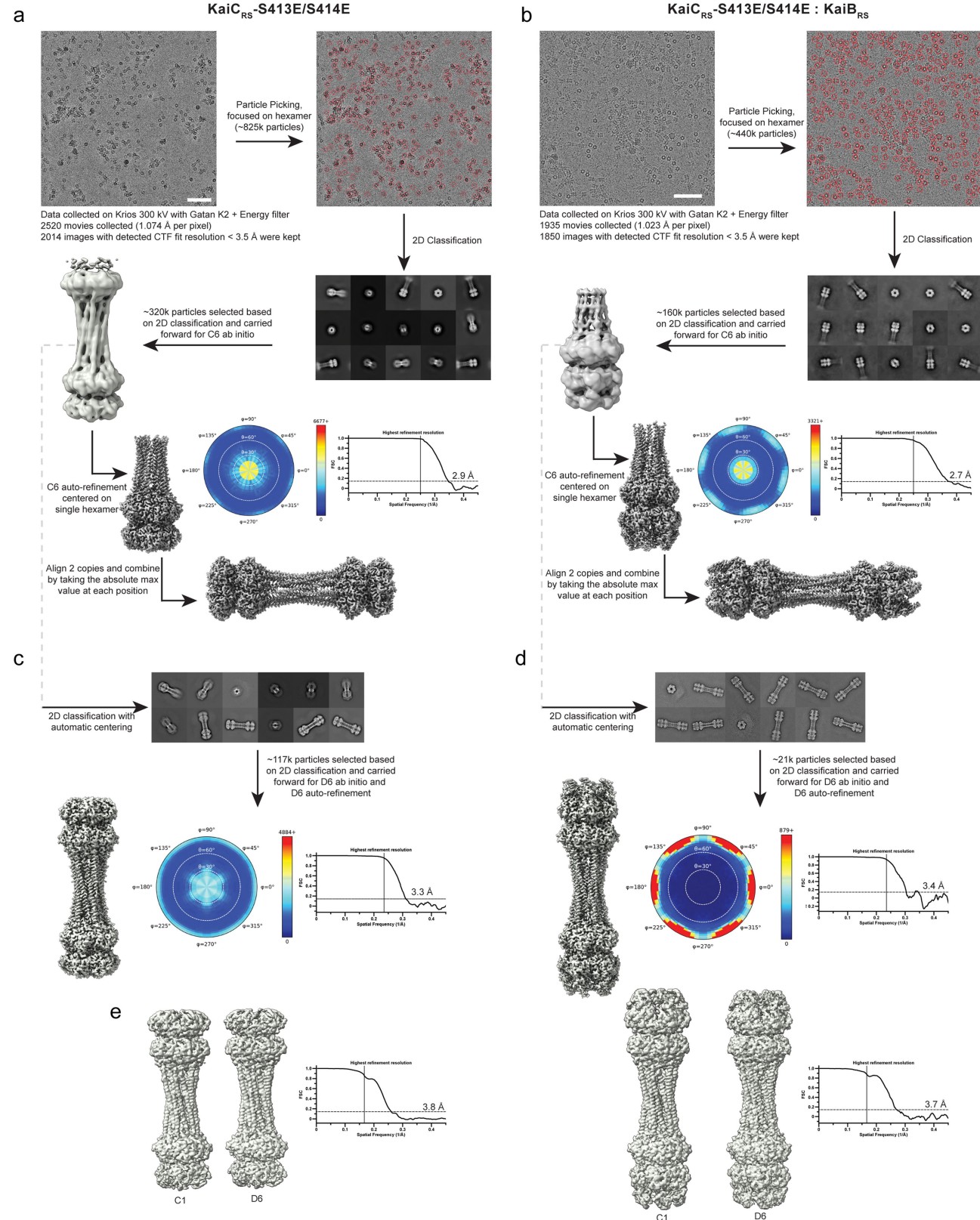

**Extended Data Fig. 4** | See next page for caption.

**Extended Data Fig. 4 | Graphical description of the cryo-EM processing workflow and validation of the final dodecamer structures.** The workflow (see description in Methods section) demonstrates a typical image (scale bar: 60 nm) and representative good class averages. The ab initio and final reconstructions are shown. Shown alongside the final reconstruction is the angular plot demonstrating the distribution of particle views and the Fourier shell correlation curve used for the global resolution estimation **(a)** KaiC$_{RS}$-S413E/S414E alone and **(b)** KaiC$_{RS}$-S413E/S414E:KaiB$_{RS}$ complex. To validate the final combined dodecamer structures, the data were reprocessed for the full dodecamer. The figure shows representative good class averages, the final reconstruction, angular distribution and Fourier shell correlation curve for the **(c)** KaiC$_{RS}$-S413E/S414E alone and **(d)** KaiC$_{RS}$-S413E/S414E:KaiB$_{RS}$ complex dodecamers. **(e)** Comparison of the $C1$ and $D6$ reconstructions of KaiC$_{RS}$-S413E/S414E alone and KaiC$_{RS}$-S413E/S414E:KaiB$_{RS}$, and Fourier shell correlation curves for the $C1$ reconstructions. The $C1/D6$ comparisons do not reveal discernable differences, suggesting that these complexes have $D6$ symmetry.

**Extended Data Fig. 5 | Correlation between the coiled-coil register shift and phosphorylation, and model for consecutive phosphorylation/ dephosphorylation events in CII domain of KaiC$_{RS}$. (a)** Structural comparison between KaiC$_{RS}$ (green) and KaiC$_{RS}$-S413E/S414E (orange, single-chain for clarity) reveals that the coiled-coil in the phosphomimetic structure points outwards, with an angle of about 20° relative to the KaiC$_{RS}$ coiled-coil. **(b)** The conformational change in the coiled-coil domain affects the dimer interface due to partner swaps with the opposite hexamer (see also Fig. 2). From an "outside perspective": the C-terminal helix in KaiC$_{RS}$ interacts with the right chain from the opposite hexamer, whereas in KaiC$_{RS}$-S413E/S414E the interaction is with the chain on the left. **(c)** Coiled-coil diagrams describe the heptad register shift that accompanies this structural rearrangement. **(d)** Based on the overlay of our structures, we propose the following model for the phosphorylation/dephosphorylation events. First, the phosphorylation cycle starts with the transfer of the γ-phosphate of ATP to the hydroxyl group of Ser414 (1; green arrow) in unphosphorylated KaiC$_{RS}$ (green) or KaiC$_{RS}$-Δcoil (cyan). Secondly, pSer414 of KaiC$_{RS}$-Δcoil (purple, singly phosphorylated) moves away from the active site placing the hydroxyl group of Ser413 closer to the γ-phosphate of ATP for the second phosphorylation (2; purple arrow). Thirdly, the doubly phosphomimetic state (KaiC$_{RS}$-S413E/S414E, orange) reveals that the phosphoryl group of pSer414 moves back towards the active site for dephosphorylation (3; orange arrow). Lastly, we hypothesize that the indole group of Trp419 "pushes" pSer413 into the active site for the second dephosphorylation event (dashed arrow), in agreement with the slower dephosphorylation rate observed in the KaiC$_{RS}$-Δcoil construct (*cf*. Fig. 2d).

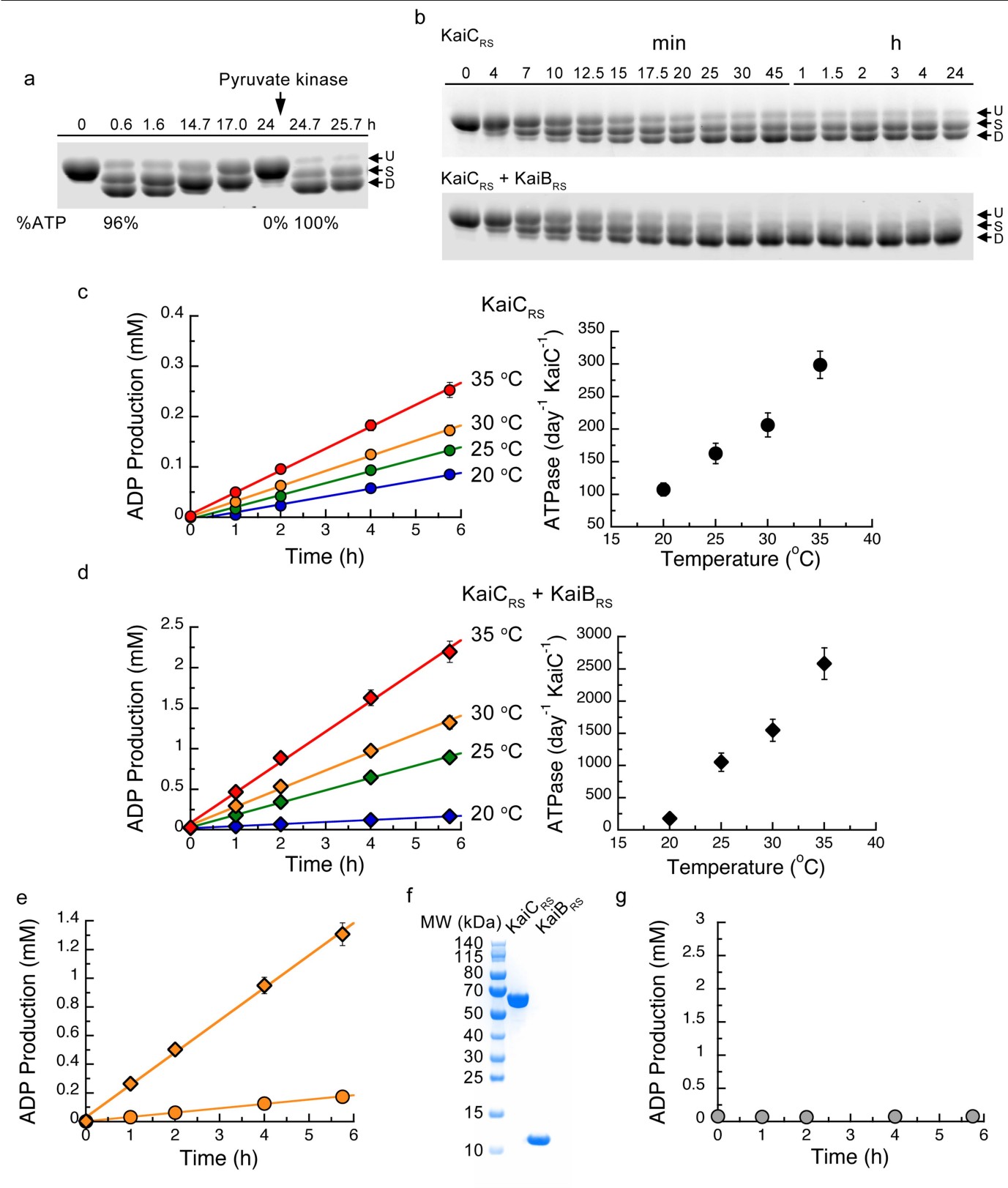

**Extended Data Fig. 6** | See next page for caption.

**Extended Data Fig. 6 | Effect of ATP-to-ADP ratio on KaiC$_{RS}$ auto-dephosphorylation and the dependence of temperature and KaiB$_{RS}$ binding on the ATPase activity of KaiC$_{RS}$.** **(a)** 10% SDS-PAGE gel of 3.5 μM KaiC$_{RS}$ and 3.5 μM KaiB$_{RS}$ in the presence of 4 mM ATP and 10 mM 2-phosphoenolpyruvate at 30 °C shows that the auto-phosphorylation cycle restarts upon regeneration of ATP by the addition of 2 U ml$^{-1}$ pyruvate kinase at the 24-hour time mark. **(b)** 10% SDS-PAGE gel of 3.5 μM KaiC$_{RS}$ without (upper panel) and with (lower panel) 3.5 μM KaiB$_{RS}$ in the presence of 4 mM ATP with an ATP-recycling system added from the beginning showing that under these conditions KaiB does not accelerate dephosphorylation. For gel source data in (a) and (b), see Supplementary Figure 2. **(c)** Representative curves for ADP production of KaiC$_{RS}$ (3.5 μM) alone and **(d)** in the presence of KaiB$_{RS}$ (3.5 μM) in 4 mM ATP measured by HPLC. The data were analysed as described in the Methods section and result in ATPase activities of 108 ± 10 day$^{-1}$ KaiC$^{-1}$ (with KaiB$_{RS}$ = 176 ± 29 day$^{-1}$ KaiC$^{-1}$) at 20 °C, 163 ± 16 day$^{-1}$ KaiC$^{-1}$ (with KaiB$_{RS}$ = 1052 ± 143 day$^{-1}$ KaiC$^{-1}$) at 25 °C, 208 ± 19 day$^{-1}$ KaiC$^{-1}$ (with KaiB$_{RS}$ = 1557 ± 172 day$^{-1}$ KaiC$^{-1}$) at 30 °C, and 300 ± 21 day$^{-1}$ KaiC$^{-1}$ (with

KaiB$_{RS}$ = 2584 ± 245 day$^{-1}$ KaiC$^{-1}$) at 35 °C. The temperature coefficient, $Q_{10}$, was calculated using the data obtained at 25 °C and 35 °C and yields a value of ~ 1.9. The standard deviations of ATPase activity at each temperature (right panels) were obtained from three replicate measurements and data are presented as mean values ± s.d. **(e)** The comparison of ADP production of KaiC$_{RS}$ in the absence (orange circles, data from panel (c)) and presence (orange diamonds, data from panel (d)) of KaiB$_{RS}$ at 30 °C indicate a 7.5-fold increase in ATPase activity for the complex. The binding of KaiB$_{RS}$ accelerates the ATPase activity of KaiC$_{RS}$ in both the CI and CII domains (see also Extended Data Fig. 8b, c). **(f)** The SDS-PAGE gel of KaiC$_{RS}$ (10 μg) and KaiB$_{RS}$ (10 μg) shows that both proteins were purified to homogeneity and the measured ATPase activity is, therefore, not due to impurities. **(g)** ADP production of KaiB$_{RS}$ in 4 mM ATP at 30 °C shows, as expected, no ATPase activity for KaiB$_{RS}$ alone and confirms the increase in ATPase activity shown in panel (d) is due to complex formation. The standard deviation for the representative curves shown in panels (c-d, left), e, and g was set to 6% assuming the largest systematic error originates from the injector.

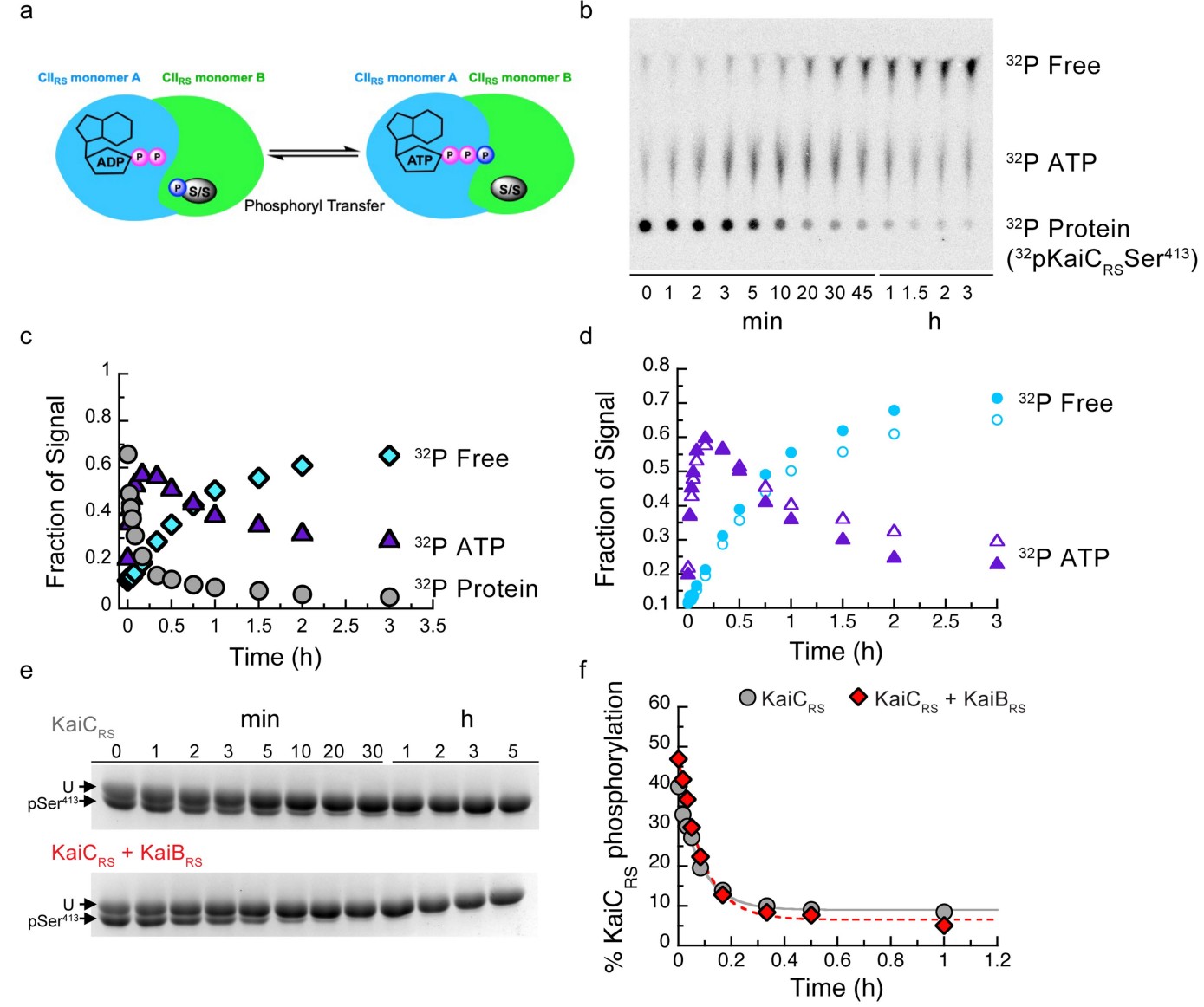

**Extended Data Fig. 7 | Dephosphorylation of KaiC$_{RS}$ occurs via an ATP-synthase mechanism and the phosphoryl-transfer step is unaffected by the binding of KaiB$_{RS}$. (a)** Possible mechanisms how KaiB$_{RS}$ could accelerate KaiC$_{RS}$ dephosphorylation at nighttime. Binding of KaiB$_{RS}$ on the CI$_{RS}$ domain directly accelerates the phosphoryl transfer from pSer to bound ADP to generate transiently bound ATP. The cartoon represents the interface between two monomers in the CII$_{RS}$ domain. **(b)** Autoradiograph of separation of $^{32}$P-KaiC$_{RS}$ at Ser413, transiently formed $^{32}$P-ATP, and free $^{32}$Pi via thin-layer chromatography (TLC) with 4 mM ADP at 30 °C, with the corresponding kinetics shown in **(c)** where gray circle, purple triangle, and cyan diamonds represent the relative concentrations of phosphorylated $^{32}$P-KaiC$_{RS}$, $^{32}$P-ATP, and free $^{32}$Pi, respectively. **(d)** Comparison of transient $^{32}$P-ATP formation and

decay in the absence (open triangle) and presence (solid triangle) of KaiB$_{RS}$ and free $^{32}$P formation in the absence (open circles) and presence of KaiB$_{RS}$ (solid circles). Faster decay of transient $^{32}$P-ATP together with higher free $^{32}$P production in the presence of KaiB$_{RS}$ indicated that KaiB$_{RS}$ accelerates hydrolysis in KaiC$_{RS}$. **(e)** SDS-PAGE gel (10%) of dephosphorylation of phosphorylated 3.5 μM KaiC$_{RS}$ at Ser413 without (upper gel) and with (lower gel) 3.5 μM KaiB$_{RS}$ in the presence of 4 mM ADP at 30 °C (for gel source data, see Supplementary Figure 2). **(f)** Densitometric analysis of data in panel (e) shows the decay of total KaiC$_{RS}$ phosphorylation in the absence (gray circles) and presence (red diamonds) of KaiB$_{RS}$ and yields rates of 11.5 ± 0.8 h$^{-1}$ and 11.0 ± 0.8 h$^{-1}$, respectively. This result indicates that binding of KaiB$_{RS}$ does not accelerate the phosphoryl-transfer step in KaiC$_{RS}$.

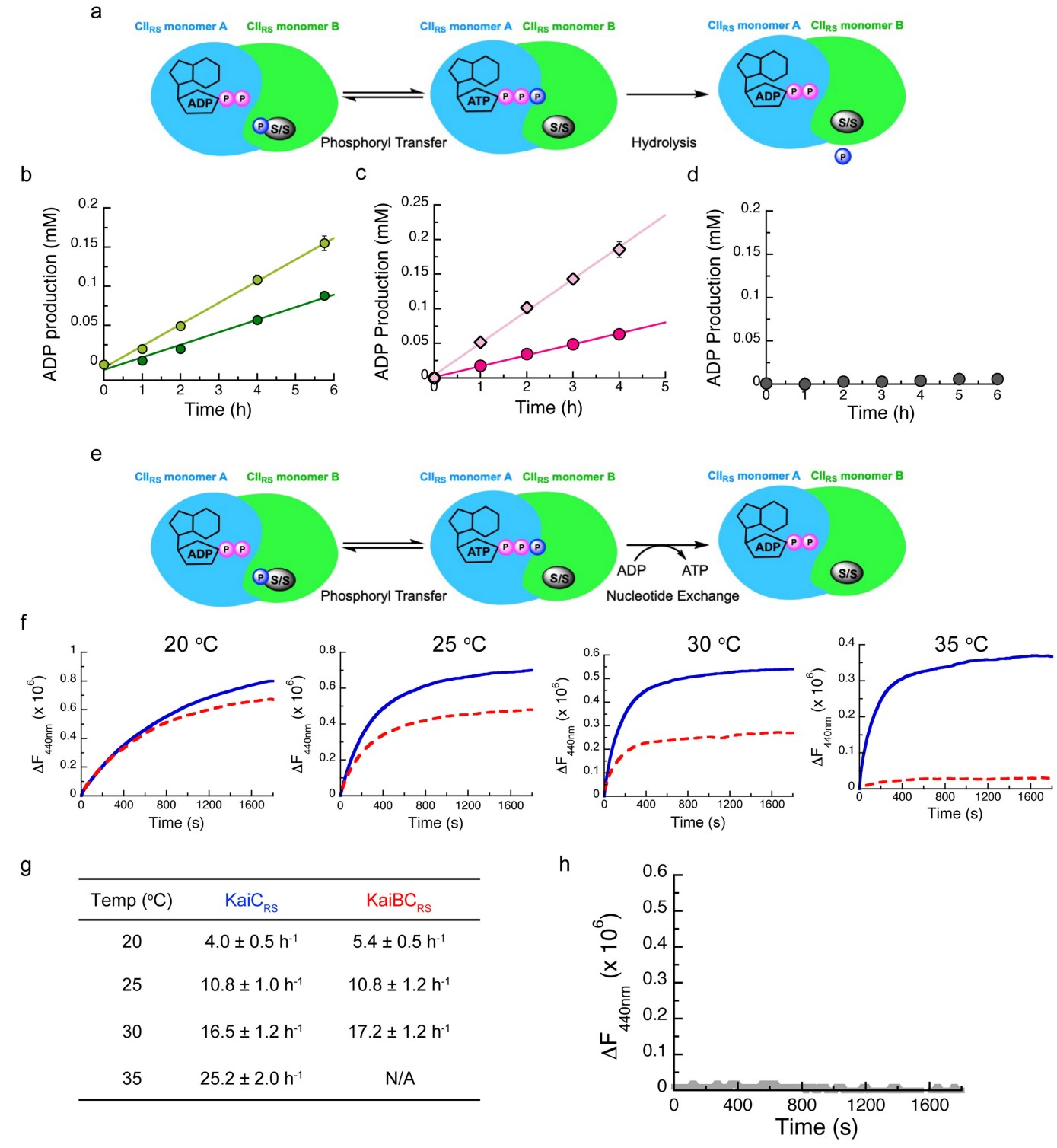

**Extended Data Fig. 8** | See next page for caption.

**Extended Data Fig. 8 | Effect of KaiB$_{RS}$ binding on ATPase activity and nucleotide exchange in the CII domain of KaiC$_{RS}$.** **(a)** Second possible mechanism to explain how KaiB$_{RS}$ accelerates KaiC$_{RS}$ dephosphorylation at nighttime: binding of KaiB$_{RS}$ to the CI$_{RS}$ domain could increase the hydrolysis rate in the CII$_{RS}$ domain and, thereby, prevent the phosphoryl transfer back from transiently formed or external ATP back to serine residues. **(b)** Representative curves for ADP production of phosphorylated KaiC$_{RS}$ with catalytic mutations in the CI domain (KaiC$_{RS}$-E62Q/E63Q, 3.5 μM) in the absence (dark green circles) and presence (light green circles) of 3.5 μM KaiB$_{RS}$ at 30 °C with 4 mM ATP was quantified using HPLC. From these data an ATPase activity in the CII domain of 112 ± 8 day$^{-1}$ KaiC$^{-1}$ and 195 ± 16 day$^{-1}$ KaiC$^{-1}$ in the presence of KaiB$_{RS}$ was determined. **(c)** Representative curves for ADP production measured by HPLC as in panel (b) of KaiC$_{RS}$ but with catalytic mutations in the CII domain (KaiC$_{RS}$-E302Q/E303Q, 3.5 μM) in the absence (dark pink circles) and presence (light pink diamonds) of 3.5 μM KaiB$_{RS}$. The corresponding ATPase activities in the CI domain are 110 ± 12 day$^{-1}$ KaiC$^{-1}$ in the absence and 320 ± 22 day$^{-1}$ KaiC$^{-1}$ in the presence of KaiB$_{RS}$. **(d)** Representative curves for ADP production of KaiCI$_{RS}$-E62Q/E63Q (construct of only CI domain with catalytic mutations) in 4 mM ATP at 30 °C shows no ATPase activity indicating that Glu62 and Glu63 are the only two residues that are responsible for ATPase activity in CI domain of KaiC$_{RS}$ and confirms the ATPase activity shown in panel b is due to ATPase activity in CII domain of KaiC$_{RS}$. The standard deviation for the representative curves shown in panels (b-d) was set to 6% assuming the largest systematic error originates from the injector. The experiments in panel (b-d) were performed in triplicate and ATPase rate given in the legend for panels (b) and (c) are presented as mean values ± s.d. **(e)** Third possible mechanism to explain how KaiB$_{RS}$ accelerates KaiC$_{RS}$ dephosphorylation at nighttime: binding of KaiB$_{RS}$ to the CI$_{RS}$ domain could promote faster nucleotide exchange in the CII$_{RS}$ domain to displace transient ATP by ADP. **(f)** Time course of fluorescence intensity at 440 nm due to binding of mant-ATP to KaiC$_{RS}$-S413E bound with ATP in the absence (solid blue trace) and presence (red dotted trace) of KaiB$_{RS}$. KaiC$_{RS}$-S413E (3.5 μM) was pre-incubated with 3.5 μM KaiB$_{RS}$ for 16 h at 20, 25, 30, and 35 °C in the presence of 50 μM ATP and an ATP-recycling system and then mixed with 250 μM mant-ATP. The observed exchange rates at each temperature are listed in the table **(g)**. **(h)** Nucleotide exchange of KaiCI$_{RS}$ (i.e., only CI$_{RS}$ domain) cannot be measured since there is no tryptophan residue in close proximity of the nucleotide binding site. In summary, KaiB$_{RS}$ accelerates KaiC$_{RS}$ dephosphorylation by increasing the hydrolysis rate in the CI and CII domains and does not affect the nucleotide exchange rate. Representative traces are shown in (f) and (h) and the fitted parameters (g; mean ± s.d.) were obtained from three replicate measurements.

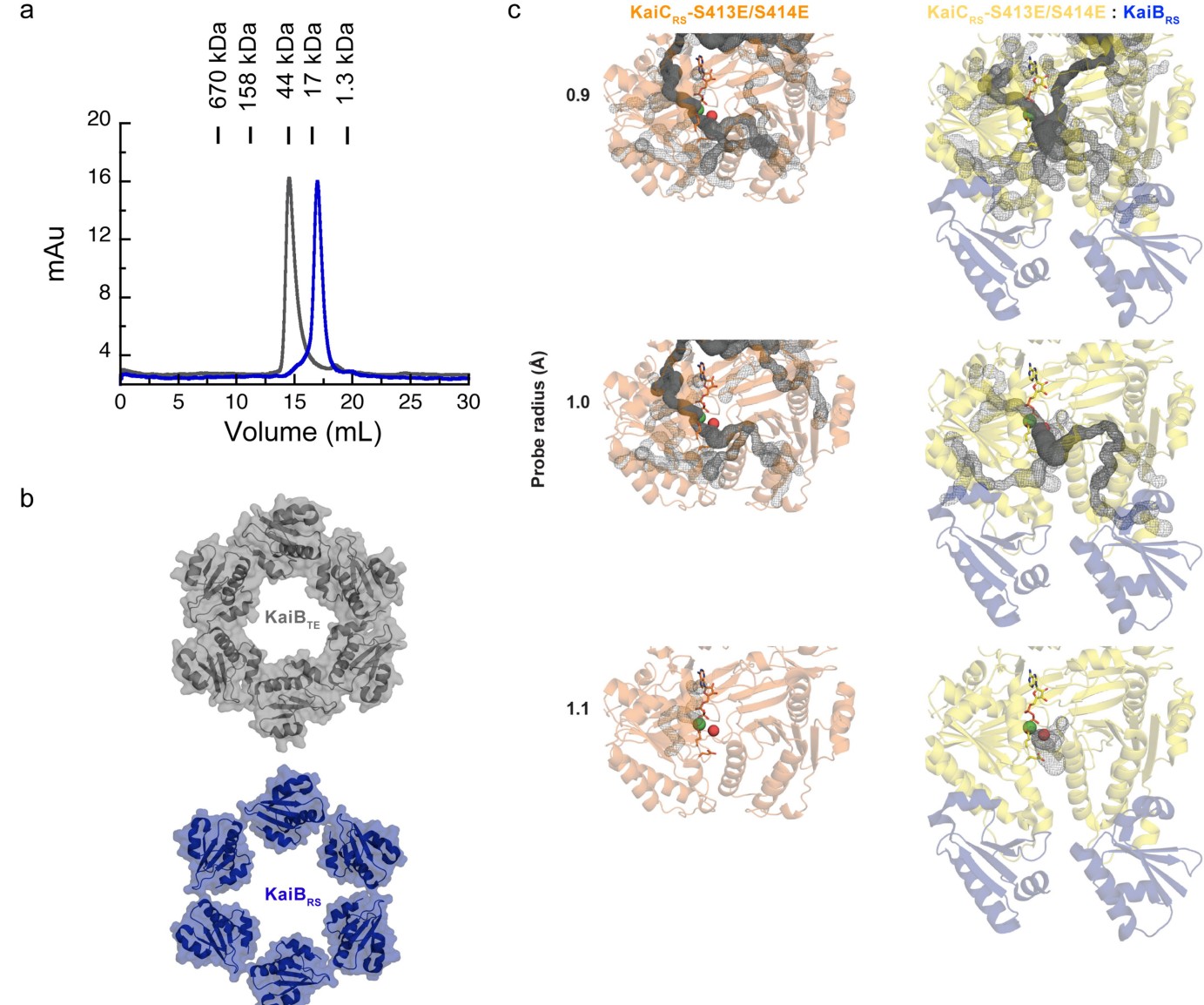

**Extended Data Fig. 9 | KaiB-KaiB interface in the KaiBC_RS complex affects the solvent accessibility into the active site of KaiC_RS-CI.** (**a**) Size-exclusion chromatography of KaiB_RS (blue) shows that it is monomeric in solution in contrast to KaiB_SE (gray), which elutes as a tetramer. Molecular-weight standards are shown above the chromatogram. (**b**) Structural comparison of KaiB_TE (gray, PDB 5jwq[26]) and KaiB_RS (blue) when bound to their corresponding KaiC hexamers. The PISA software package[37] determines that for the KaiBC_TE complex the interface between the KaiB_TE monomers is 255 Å², whereas the average interface between KaiB_RS monomers is only 45 Å² in the KaiBC_RS complex. (**c**) To understand how KaiB_RS binding to KaiC_RS-CI domain increases the hydrolysis rate, we investigated whether conformational changes modulated substrate access to the active site. The CAVER software[61] was used to calculate tunnels (gray mesh)

leading into the active site of the CI domain of KaiC_RS-S413E/S414E alone (orange) and the KaiC_RS-S413E/S414E:KaiB_RS complex (yellow:blue) with varying probe radii. In both structures, the active site was occupied by ADP:Mg²⁺ (sticks and green sphere, respectively). The crystal structure of bovine F1-ATPase in complex with a transition-state analogue (PDB 1w0j, chain D)[62] was used as a reference to determine the position of the catalytic water molecule in the active site (shown as a red sphere). The calculated tunnels connect bulk solvent to the catalytic water when KaiB_RS is bound for probe radii larger than the default value of 0.9 Å, but never in its absence. These results suggest that KaiB_RS facilitates the access of water into the active site of KaiC_RS-CI via long-range conformational changes and thus enhances ATP hydrolysis.

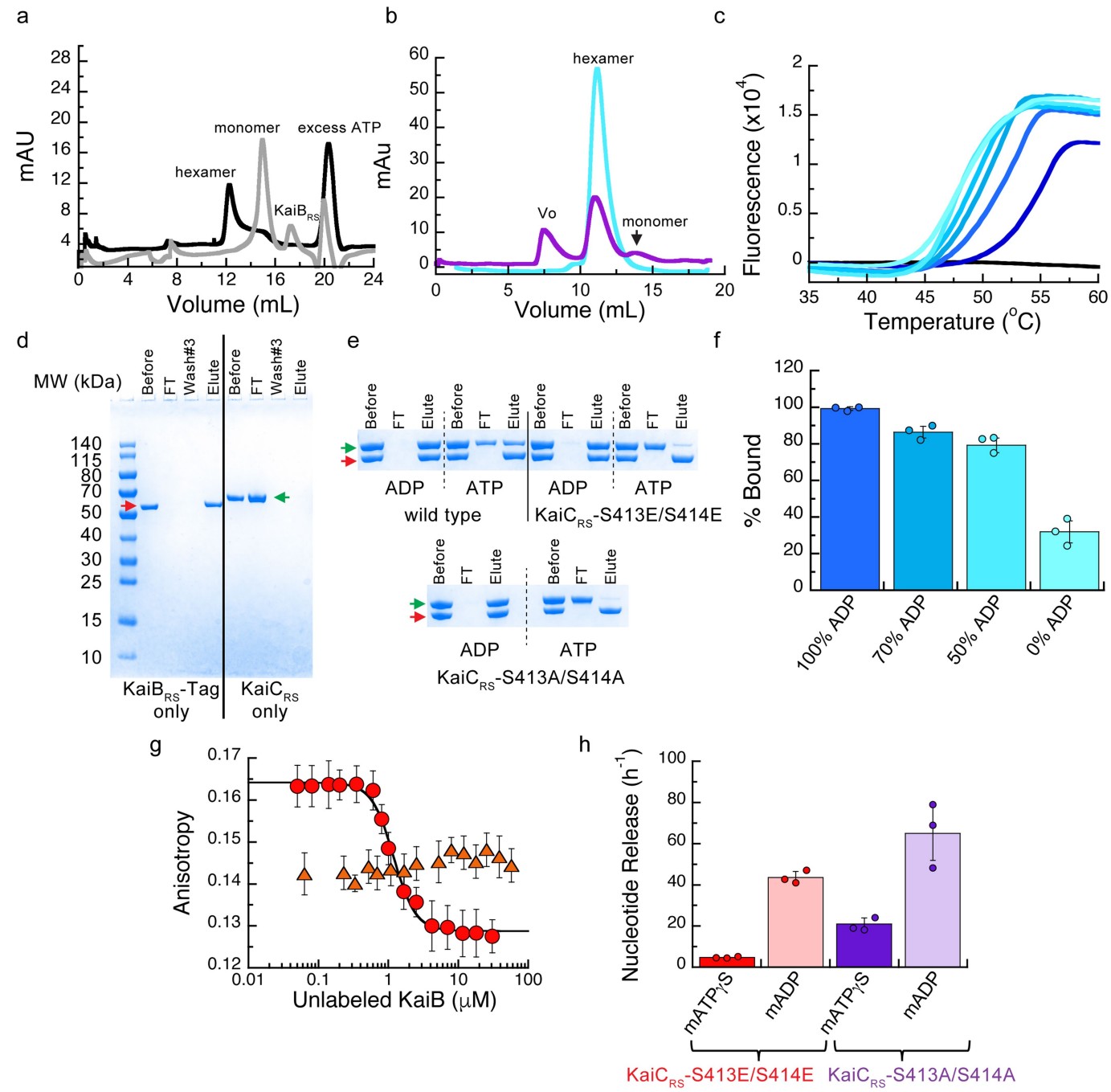

**Extended Data Fig. 10** | See next page for caption.

**Extended Data Fig. 10 | KaiB$_{RS}$ binds preferentially to the post-hydrolysis state of KaiC$_{RS}$ and affects its stability. (a)** Size-exclusion chromatography of 50 μM KaiC$_{RS}$-CI (CI$_{RS}$ domain) in the absence (black line, hexamer) and presence (gray line, monomer) of 50 μM KaiB$_{RS}$ in 1 mM ATP buffer. **(b)** Size-exclusion chromatography of 50 μM KaiC$_{RS}$-Δcoil in the presence of 50 μM KaiB$_{RS}$ (purple). The reference sample (50 μM KaiC$_{RS}$-Δcoil) is a hexamer in solution (cyan) and after the addition of 50 μM KaiB$_{RS}$ the mixture was incubated at 30 °C for 3.5 h (purple) before running the samples again on a Superdex-200 10/300 GL column at 4 °C. These data show that binding of KaiB$_{RS}$ results in (i) disassembly of the hexameric KaiC$_{RS}$-Δcoil structure into its monomers and (ii) aggregation as detected by the elution in the void volume of the column (v$_0$). **(c)** Thermal denaturation profiles for KaiC$_{RS}$-S413E/S414E in the presence of 1 mM ADP are shown from dark to light blue for increasing concentrations of KaiB$_{RS}$ (between 0 – 4 μM). The black line represents KaiB$_{RS}$ alone (15 μM), which shows no fluorescence signal as it does not bind to SYPRO Orange due to a lack of a hydrophobic core. The T$_m$ decreases upon the addition of KaiB$_{RS}$, indicating that binding of KaiB$_{RS}$ destabilizes the KaiC$_{RS}$ dodecamer. Likely due to loosening up of interface and the KaiC$_{RS}$ structure, thereby allowing for the formation of a tunnel that connects bulk solvent to the position of the hydrolytic water in the active site (see Extended Data Fig. 9). **(d)** SDS-PAGE analysis showing the control experiment for pull-down assay. The first four lanes after the molecular weight marker are KaiB$_{RS}$-Tag samples (red arrow) and show that KaiB$_{RS}$-Tag binds tightly to the column. The last four lanes are control pull-down assay experiments for KaiC$_{RS}$ (green arrow) and show that KaiC$_{RS}$ alone is unable to bind to the column. The lanes represent the initial sample used in pull-down assay (Before), flow-through after loading sample onto the column (FT), flow-through after washing the column three times with the binding buffer (Wash #3), and sample after elution with imidazole (Elute). **(e)** SDS-PAGE analysis of pull-down assay to measure the complex formation between KaiB$_{RS}$-Tag and wild-type KaiC$_{RS}$, KaiC$_{RS}$-S413E/S414E, or KaiC$_{RS}$-S413A/S414A in the presence of 4 mM ADP or ATP (with an ATP-recycling system). For gel source data, see Supplementary Figure 2. **(f)** Percentage of wild-type KaiC$_{RS}$ bound to KaiB$_{RS}$-Tag protein for different ATP-to-ADP ratios (4 mM total nucleotide concentration) at 25 °C as measured from pull-down assays. **(g)** Fluorescence anisotropy at 30 °C of unlabeled KaiB$_{RS}$ competitively replacing the fluorophore-labeled KaiB$_{RS}$ (KaiB$_{RS}$-6IAF) from KaiC$_{RS}$-S413E/S414E in the presence of 4 mM ADP (red circles, $K_D$ value of 0.79 ± 0.06 μM) and 4 mM ATP with an ATP-recycling system (orange triangles). In the latter experiment no change in anisotropy is observed, indicating that only a small fraction of KaiB$_{RS}$-6IAF is bound under these conditions. The average anisotropy and standard error were calculated from ten replicate measurements. **(h)** The mant-ATPγS or mant-ADP release is shown as bar graphs with observed rates of 4.8 ± 0.2 h$^{-1}$ and 43.6 ± 3.0 h$^{-1}$ for mant-ATPγS and mant-ADP releasing from KaiC$_{RS}$-S413E/S414E, respectively, and 21.0 ± 3.0 h$^{-1}$ and 65 ± 15 h$^{-1}$ for mant-ATPγS and mant-ADP releasing from KaiC$_{RS}$-S413A/S414A, respectively. The result shows that CII domain of KaiC$_{RS}$ prefers binding of ATP over ADP. Experiments in panels (f) and (h) were performed in triplicate and data are presented as mean values ± s.d.

**Extended Data Table 1 | X-ray crystallography data collection and refinement statistics**

|  | KaiC$_{RS}$<br>PDB: 8dba<br>(oligomeric state:<br>dodecamer) | KaiC$_{RS}$-$\Delta$coil<br>PDB: 8db3<br>(oligomeric state:<br>hexamer) |
|---|---|---|
| **Data collection** |  |  |
| Space group | P1 | C222$_1$ |
| Cell dimensions |  |  |
| $a, b, c$ (Å) | 105.11, 136.09, 146.2 | 94.06, 197.28, 150.19 |
| $\alpha, \beta, \gamma$ (°) | 93.13, 94.43, 108.09 | 90, 90, 90, |
| Resolution (Å) | 47.65 - 3.5 (3.625 - 3.5)* | 53.89 - 2.9 (3.004 - 2.9)* |
| $R_{sym}$ or $R_{merge}$ | 0.2213 (1.372) | 0.4267 (2.994) |
| $I / \sigma I$ | 2.57 (0.73) | 5.54 (1.15) |
| Completeness (%) | 97.62 (94.08) | 99.89 (99.97) |
| Redundancy | 1.9 (2.0) | 7.2 (7.3) |
|  |  |  |
| **Refinement** |  |  |
| Resolution (Å) | 3.5 | 2.9 |
| No. reflections | 94151 (9047) | 31344 (3102) |
| $R_{work}$ / $R_{free}$ | 0.2323/ 0.2775 | 0.2308/0.2892 |
| No. atoms | 47510 | 10293 |
| Protein | 46772 | 10131 |
| Ligand/ion | 948 | 231 |
| Water | 67 | 0 |
| $B$-factors |  |  |
| Protein | 102.48 | 56.06 |
| Ligand/ion | 89.08 | 44.62 |
| Water | 72.52 | - |
| R.m.s. deviations |  |  |
| Bond lengths (Å) | 0.002 | 0.004 |
| Bond angles (°) | 0.62 | 0.73 |

A single crystal was used for each structure. *Values in parentheses are for the highest-resolution shell.

**Extended Data Table 2 | Cryo-EM data collection, refinement and validation statistics**

| | KaiC$_{RS}$-S413E/S414E (EMD-29505) (PDB 8FWI) | KaiC$_{RS}$-S413E/S414E:KaiB$_{RS}$ (EMD-29506) (PDB 8FWJ) |
|---|---|---|
| **Data collection and processing** | | |
| Magnification | 79,000 | 79,000 |
| Voltage (kV) | 300 | 300 |
| Electron exposure (e–/Å$^2$) | 100 | 100 |
| Defocus range (µm) | -0.5 to 2.5 | -0.5 to 2.5 |
| Pixel size (Å) | 1.074 | 1.023 |
| Symmetry imposed | C6 | C6 |
| Initial particle images (no.) | 825,000 | 440,000 |
| Final particle images (no.) | 320,000 | 160,000 |
| Map resolution (Å) | 2.9 | 2.7 |
| FSC threshold | 0.143 | 0.143 |
| | | |
| **Refinement** | | |
| Initial model used (PDB code) | 8DBA | 8DBA |
| Model resolution (Å) | 3.1 | 2.9 |
| FSC threshold | 0.5 | 0.5 |
| Model composition | | |
| Non-hydrogen atoms | 103236 | 60492 |
| Protein residues | 6576 | 7644 |
| Ligands | 48 | 48 |
| B factors (Å$^2$) | | |
| Protein | 63.44 | 104.75 |
| Ligand | 52.39 | 79.22 |
| R.m.s. deviations | | |
| Bond lengths (Å) | 0.007 | 0.003 |
| Bond angles (°) | 1.262 | 0.720 |
| Validation | | |
| MolProbity score | 1.82 | 2.52 |
| Clashscore | 11.86 | 31.00 |
| Poor rotamers (%) | 2.04 | 3.45 |
| Ramachandran plot | | |
| Favored (%) | 98.16 | 97.31 |
| Allowed (%) | 1.84 | 2.69 |
| Disallowed (%) | 0.0 | 0.0 |

# Reporting Summary

## Statistics

For all statistical analyses, confirm that the following items are present in the figure legend, table legend, main text, or Methods section.

| n/a | Confirmed | |
|---|---|---|
| ☐ | ☒ | The exact sample size (*n*) for each experimental group/condition, given as a discrete number and unit of measurement |
| ☐ | ☒ | A statement on whether measurements were taken from distinct samples or whether the same sample was measured repeatedly |
| ☒ | ☐ | The statistical test(s) used AND whether they are one- or two-sided *Only common tests should be described solely by name; describe more complex techniques in the Methods section.* |
| ☒ | ☐ | A description of all covariates tested |
| ☒ | ☐ | A description of any assumptions or corrections, such as tests of normality and adjustment for multiple comparisons |
| ☐ | ☒ | A full description of the statistical parameters including central tendency (e.g. means) or other basic estimates (e.g. regression coefficient) AND variation (e.g. standard deviation) or associated estimates of uncertainty (e.g. confidence intervals) |
| ☒ | ☐ | For null hypothesis testing, the test statistic (e.g. *F*, *t*, *r*) with confidence intervals, effect sizes, degrees of freedom and *P* value noted *Give P values as exact values whenever suitable.* |
| ☒ | ☐ | For Bayesian analysis, information on the choice of priors and Markov chain Monte Carlo settings |
| ☒ | ☐ | For hierarchical and complex designs, identification of the appropriate level for tests and full reporting of outcomes |
| ☒ | ☐ | Estimates of effect sizes (e.g. Cohen's *d*, Pearson's *r*), indicating how they were calculated |

*Our web collection on statistics for biologists contains articles on many of the points above.*

## Software and code

Policy information about availability of computer code

| Data collection | All data collection was performed using the software provided with the respective instruments:<br>- Cryo-EM: SerialEM v3.6<br>- Fluorescence (anisotropy): FluorEssence v3.5 (HORIBA Scientific)<br>- SDS page gels scannner: ChemiDoc Imager v 6.0.1 (Bio-Rad)<br>- TLC plate scanner: Amersham Typhoon v5 (GE Healthcare) |
|---|---|
| Data analysis | Most of the software used for the analysis described in this paper are freely/publicly available:<br>- X-ray Crystallographic data was analyzed using COOT 0.9.8.1, programs in the PHENIX 1.20.1-4487 software suite (e.g., MRage and Phaser). Additional software packages include Zanuda available in the software package CCP4 7.4, Achesym (web server, no version information available), SamCC-Turbo 0.0.2, PyMOL 2.6.0, and CAVER 3. The Supplementary Video was created in part by using UCSF Chimera 1.15.<br>- cryo-EM data was analyzed using cisTEM 2.0.0-alpha, COOT 0.9.8.1, and PHENIX 1.20.1-4487.<br><br>Commercially available software used:<br>- KinTek Explorer 10 for the analysis of kinetic experiments<br>- Kaleidagraph 4.5.3 (Synergy) for the fitting/plotting of all biochemical experiments<br>- Gel densitometry: Image Lab v 6.0.1 (Bio-Rad)<br>- TLC plates scanner: ImageQuant TL 7.0 software |

For manuscripts utilizing custom algorithms or software that are central to the research but not yet described in published literature, software must be made available to editors and reviewers. We strongly encourage code deposition in a community repository (e.g. GitHub). See the Nature Portfolio guidelines for submitting code & software for further information.

# Data

Policy information about availability of data

All manuscripts must include a data availability statement. This statement should provide the following information, where applicable:

- Accession codes, unique identifiers, or web links for publicly available datasets
- A description of any restrictions on data availability
- For clinical datasets or third party data, please ensure that the statement adheres to our policy

Data availability

Structure factors and refined models obtained using X-ray crystallography are deposited in the Protein Data Bank (PDB) under accession codes 8dba (wild-type KaiCRS) and 8db3 (KaiCRS-Δcoil).

Cryo-EM maps and refined models are deposited in the Electron Microscopy Data Bank (EMDB) and Protein Data Bank (PDB), respectively. The composite map and model for the KaiCRS-S413E/S414E dodecamer reconstruction are submitted under entries EMD-29505 and 8fwi, respectively. The composite map and model for the KaiCRS-S413E/S414E:KaiBRS dodecamer reconstruction are submitted under entries EMD-29506 and 8fwj, respectively. The focused KaiCRS-S413E/S414E hexamer refinement map is available under accession EMD-29507 and the focused KaiCRS-S413E/S414E:KaiBRS hexamer refinement map is available under accession EMD-29508. The full KaiCRS-S413E/S414E dodecamer refinement is available under accession EMD-29509 and the full KaiCRS-S413E/S414E:KaiBRS dodecamer refinement is available under accession EMD-29510.

Other data sets used are all publicly available in public community/discipline-specific repositories: PDB 5jwq, PDB 1w0j, PDB 1tf7, PDB 7s65, the accession codes for protein sequences used the sequence alignments/phylogeny are listed in Supplementary Datasets 1 and 2.

# Human research participants

Policy information about studies involving human research participants and Sex and Gender in Research.

| Reporting on sex and gender | N/A |
|---|---|
| Population characteristics | N/A |
| Recruitment | N/A |
| Ethics oversight | N/A |

Note that full information on the approval of the study protocol must also be provided in the manuscript.

# Field-specific reporting

Please select the one below that is the best fit for your research. If you are not sure, read the appropriate sections before making your selection.

☒ Life sciences        ☐ Behavioural & social sciences        ☐ Ecological, evolutionary & environmental sciences

For a reference copy of the document with all sections, see nature.com/documents/nr-reporting-summary-flat.pdf

# Life sciences study design

All studies must disclose on these points even when the disclosure is negative.

| Sample size | The sample size has been described in the figure legends where applicable. No statistical method has been used to predetermine the sample size. Experiments were typically repeated in triplicates independently to ensure the robustness of the conclusions. |
|---|---|
| Data exclusions | For the phylogenetic tree, sequences with a sequence homology above 90% were excluded from the final analysis. For all other experiments no data were excluded from the analyses. |
| Replication | The replicate number for experiments is described in the figure legends. Typically, experiments were repeated as biological triplicates to ensure the robustness of the conclusions. All attempts at replication were successful. |
| Randomization | Not applicable as there were no groups to be allocated. |
| Blinding | No allocation into groups was performed (see above), so "blinding" of investigators is not applicable. |

# Reporting for specific materials, systems and methods

We require information from authors about some types of materials, experimental systems and methods used in many studies. Here, indicate whether each material, system or method listed is relevant to your study. If you are not sure if a list item applies to your research, read the appropriate section before selecting a response.

## Materials & experimental systems

| n/a | Involved in the study |
|-----|----------------------|
| ☒ | Antibodies |
| ☒ | Eukaryotic cell lines |
| ☒ | Palaeontology and archaeology |
| ☒ | Animals and other organisms |
| ☒ | Clinical data |
| ☒ | Dual use research of concern |

## Methods

| n/a | Involved in the study |
|-----|----------------------|
| ☒ | ChIP-seq |
| ☒ | Flow cytometry |
| ☒ | MRI-based neuroimaging |

