## [Peer Review File · Nature]

Manuscript Title: From primordial clocks to circadian oscillators

Reviewer Comments & Author Rebuttals

Reviewer Reports on the Initial Version:

Referees' comments:

Referee #1 (Remarks to the Author):

This manuscript presents structures and biochemical studies of clock protein homologs KaiB and KaiC from the purple, non-sulfur photosynthetic proteobacterium *Rhodobacter sphaeroides* (RS). Unlike the KaiABC system from the cyanobacterium *S. elongatus*, which creates a self-sustained and temperature compensated oscillator in vitro, the RS KaiBC proteins from exhibit changes in their structure and activity that may allow them to create a less robust, hourglass-like timer in the absence of KaiA. There are several key differences in the RS Kai proteins, including a coiled-coil extension of the KaiC C-terminus that allows for the KaiC hexamer to form a homododecamer. Intriguingly, phosphomimetic mutations in the KaiC CII domain lead to a large register shift in the coiled coil complex that may be linked to promotion of autodephosphorylation via an allosteric network leading to the CII active site, as deletion of the coiled coils stabilizes phosphoKaiC. Differences in the structure of the A-loop, located near the phosphorylation sites in KaiC, likely lead to an increased basal rate of nucleotide exchange and phosphorylation compared to *S. elongatus* KaiC. KaiB binding enhances ATPase activity at the KaiC CI and CII domains, the latter of which promotes dephosphorylation of the KaiC CII domain independent of nucleotide exchange. Consistent with this, changes in the ATP/ADP ratio serve as an environmental cue in this system, as has been previously demonstrated for the cyanobacterial clock. The experiments here are rigorously performed and analyzed. Overall, this study represents an exciting step forward in discerning the possible mechanism of what could be an ATP/ADP-sensitive hourglass clock. However, the addition of one or two experiments could provide a missing link uniting several of the observations that appear to be unique to the RS system here and establish that it does indeed form an hourglass-like oscillator.

Major comments:

1. The authors state that the proposal that the *kaiBC* gene cluster is responsible for circadian gene expression in *R. sphaeroides* is controversial as no decisive experimental evidence (e.g., *kaiC* knockout strain or in vitro experiments) has been presented in prior publications. Unfortunately, these data aren't presented here either. Although the authors present compelling evidence of an "oscillatory-like pattern" in the presence of KaiB in vitro (Fig. 3), their conclusion that the KaiBCRS system is sufficient to make an hourglass-like timer would be strengthened by either (a) an in vitro assay with 12-hr buffer exchanges into day or night-like ATP/ADP ratios for at least 2 full cycles to demonstrate bona fide oscillations in KaiC phosphorylation or KaiB recruitment or (b) genetic deletion of KaiB and/or KaiC followed by rhythm assays in *R. sphaeroides*.

2. The authors have implicated both changes in the coiled coil architecture of KaiC in the S413E/S414E phosphomimetic mutant and KaiB binding to the KaiC CI domain in promoting KaiC autodephosphorylation at the CII domain—does this activity of KaiB depend on the presence of the coiled coils? Identifying if/how they collaborate to regulate KaiC phosphorylation state would help to further develop the mechanism highlighted in Fig. 4g.

Minor comments

1. The following statement in the abstract is a bit confusing, as it could be interpreted that the cyanobacterial KaiBC proteins could give rise to an hourglass timer. One suggested is: “The canonical KaiABC system in cyanobacteria is well understood, but little is known about more ancient systems [that possess just KaiBC], except for controversial reports that [they] exhibit a basic, hourglass-like timekeeping mechanism.”

2. The statement on line 49 that “... in prokaryotes the circadian rhythm is regulated by only three proteins (kaiA, kaiB, and kaiC)” is not quite true—there is ample evidence that other proteins regulate cyanobacterial rhythms in addition to the Kai proteins (e.g., SasA, CikA, LdpA, Pex, KidA).

3. The statement “The first obvious question is how KaiCRS and other members in the clade can auto-phosphorylate despite having no KaiA, which is known to be crucial for this function in the canonical KaiABC system” is true but a little bit misleading—KaiA is needed for oscillations in KaiC phosphorylation, but KaiC can autophosphorylate on its own, and this activity is preferentially enhanced (over autodephosphorylation) by decreasing temperatures (e.g., 4 °C). At 30 °C, as in Fig. 1b, autodephosphorylation of KaiCSE is favored (Nishiwaki et al. 2007, EMBO J.). The statement on lines 167-170 could also be edited a bit to reflect these data better.

4. Typo on line 213: phopsphorylated.

5. It is quite interesting that KaiBRS is a monomer in solution, unlike the tetrameric KaiB found in KaiABC systems. Is it known if the KaiBRS monomer takes on the “fold-switch” conformation in the absence of KaiC?

6. A reference to Snijder et al. (2014, PNAS) should be added to line 258, as they were the first to demonstrate cooperativity in KaiB binding to the KaiCSE hexamer.

7. It was previously demonstrated, not implied, that ATPase activity (or the post-hydrolysis state) in the KaiC CI domain is critical for KaiB binding (Phong et al., 2012, PNAS).

Referee #2 (Remarks to the Author):

A. The paper “From primordial clocks to circadian oscillators” by Pitsawong, Padua et al. includes a wealth of biochemical and biophysical data to try shading light into the mechanism that makes KaiCB an hour glass in those organisms that lacks KaiA. In short the authors describe structural and regulatory mechanisms of a primitive circadian clock in a fairly exhaustive way. The

themes/questions that this article addresses, the definition of circadian systems, hour-glass biological mechanism and its evolution, are of general interest for a broad audience of readers. Moreover the molecular details addressed in the article are very relevant for any structural biologist interested in the molecular switches that determine fine regulation and auto regulation in particular.

In the primitive KaiBC system from *Rhodospirillum rubrum* the authors identify the presence of a large KaiC C-terminal extension that fold into coiled-coil domains and hold together two KaiC hexamers and confer stability. The authors claim that the presence of these long coiled-coils and their positioning (different in the phosphorylated vs the non-phosphorylated form) defines a long-range allosteric network that correlates with auto-phosphorylation and dephosphorylation. This is only partially convincing for me as - the comparisons of the different structures is not clear from the text/ what is X-ray what cryo-EM in terms of oligomeric states/what is crystal packing what in solution dodecameric structure/deletion of the coiled-coil is a bit rough-what about point mutations or deletion of only the last 30aa?

Having shown that the KaiC from RS is able of both auto phosphorylation and dephosphorylation the authors go on showing how the ATP-ADP ratios characteristic of the day and night status modulates the phosphorylation of KaiC. Moreover, they go on investigating the mechanism by which KaiB helps maintaining a night de-phosphorylated state. They consider 3 hypothesis and thoroughly test them (nice and clear ED Figs 7 and 8) thus finding a mechanism of phosphate transfer, similar to the ATP-synthase, followed by KaiB-accelerated ATP hydrolysis. I find this part of the results very interesting and solid. Finally the authors also solve the structure of the Kai-BC complex and suggests a mechanism for increased ATPase activity via higher solvent accessibility in the complex.

B. The work, to my knowledge, is quite novel as it presents a primordial circadian clock made only of KaiBC and thus giving information about basis of cyclic systems. I am not an expert in the field, but, for what I can see, the authors referenced well the existing literature, mainly based on KaiABC, including the studies available on pre-print.

C. This work includes a constant interplay and corroboration steps between biochemical and structural results, which makes it solid. However, while it is really impressive that the authors determined so many different structures, I think that their analysis should be more thorough and better presented in the manuscript to make a really strong structural contribution to the work. For example, one or two extended figures describing the cryo-EM procedures and images of the 2D classes and the maps obtained should be prepared. The comparison between the unphosphorylated (U) and phosphomimic seems to be done between the crystal structure (dodecamer?) with the cryo-EM map phosphomimic. However, is the crystal structure a dodecamer in the asymmetric unit or in the crystal packing? Information about the oligomeric state could be added to the X-ray and cryo-EM tables. In the material and methods section the authors explain that the cryo-EM dodecameric structures were actually obtained upon combination of more refined C6 symmetries hexamers because the flexibility of the coiled-coil mediated C-terminal interaction limited the attainable resolution of the dodecameric structure. Now several questions arise:

1. What happens if and no symmetry is applied, even at the level of hexamer analysis? How does a C1 reconstruction of KaiCs compare with the recent reconstruction by Swan et al.

(<https://www.biorxiv.org/content/10.1101/2021.09.14.460370v1.full>)? Any interesting insights about the ATP pockets in C1 reconstructions of the U versus phosphomimic?

2. The authors say that “the hexamer reconstruction were overlaid on top of each other and combined by taking the voxel with the maximum absolute value from the two overlaid copies”. How was this done? Did they use the max voxel procedure in chimera? Or with map combination using Phenix? In any case, a combination a posteriori of the two hexamers might make the analysis of the specific changes in the coiled-coil shift of the phosphorylated (cryo-EM with hexamer combination) vs the unphosphorylated (X-ray crystal packing?) states less strong. It is therefore absolutely required that the authors show in supplementary data the look of the naturally occurring dodecameric structures even if at low resolution (how low? How interpretable at the helical level?).

3. Even at low resolution what about scanning the conformational spaces using cryoDRG or 3D variability for the dodecameric assemblies? Then I would expect that multiple possible variants with the two hexamers oriented differently compared to one another should be done. This could be very interesting especially by looking at the effect of different bending of the coiled-coils on the A-loops. In summary, further structural analysis or at least a better presentation of it in the extended and supplementary materials would strengthen the results about the allosteric network at the moment only validated by coiled-coiled deletion. I would argue that a full deletion is a harsh thing to do and that some point-mutant analysis should be done as complement or at least the deletion of only the last 30 aa.

4. An additional clarification is required: the crystal structures were used instead of an ab-initio? So what happens if an ab initio is generated from the by splitted hexamers with the coiled-coil most likely “floating” around or less resolved?

D. I have partially answered this in the section above regarding the cryo-EM data. Would be important to have a workflow with numbers of particles and pictures of the different structured obtained and how. Regarding the biochemical assays, I noticed that often there are no multi points or sigma bars nor is stated in the figure captions how many measurements were performed. I assume that at least triplicates were done for each experiment presented, but this must be stated each time.

E/F. Overall I think that this work is fairly robust and interesting, I would however required the revisions/experiment/presentation of data suggested in sections C and D.

G. To my knowledge the authors take onto consideration the existing literature.

H. The title does not really represent a condensation of the findings described in the manuscript. The abstract is clear. The “temperature factor”, i mean the reference to the temperature compensation aspect is mentioned here and there in the text. I find it confusing, and I do not understand its relevance. It might be more obvious for people in the field. Some reference could be added in the introduction such as after “mechanism that is different from the widely studied circadian oscillator in *Synechococcus elongatus* PCC 7942” or after the referral to the dynein coiled-coil regulation.

The text could be clearer for readers outside the field in the part where the extended A-loop conformation is explained. What is the importance of the 422 loop?

Line 185: When defining the ATP:ADP ratio there is a typo. The regulatory role of the KaiB is very well explained in the ED Figs 7 and 8, while in the main text is not as clear, more specifically the conclusions are not spelt out clearly. In the Discussion, this sentence "The dodecameric KaiCRS shows constitutive kinase-activity due to its extended C-terminal tail that forms a coiled-coil bundle with the opposing hexamer, and auto-phosphorylation occurs within half an hour" should be tuned down as stated as it is now it almost says that the coiled-coil itself has kinase activity. The comparison with the KaiABC system is very clear. The conclusions about the role of the coiled-coil in the allosteric network is not very obvious and convincing to me (from the data), even though likely and tempting to imagine. I would leave it more as an open question that requires further studies. Probably further analysis of the dodecamers in the cryo-EM datasets could give more insights as well as point mutations e.g. for W419. Maybe, in the discussion, a sentence explaining more in details the analogy with the dynein register shift could help.

Comments on figures:

In many figures the spatial order of the panels is strange and not always the same. I would expect to have always the b panel following a on the top right not bottom left.

Fig. 1e: It would be good to indicate the 422-loop also in the zoomed view. Moreover not all KaiC seem to have the same 422-loop.

Fig. 2a: Are the colours chosen ok for color-blind people?

Fig. 2b: I find this panel a bit confusing. I think it would be better to show one picture as it is now in the left but the details of the phosphorylated and unphosphorylated forms separately.

Fig. 3a: Please state here all the meaning of the acronyms U, S, D used in the figure.

Fig. 3d: Here is the only time where it is stated that two replicates were done. Should not be three replicates for all the biochemical experiments?

Fig. 3f: Unclear; where are the point values coming from?

Fig. 4g: Would be good to have here also an additional summarising schematic showing all that is happening in CI CII and the coiled coils upon binding of KaiB and with ATP/ADP ratios changing.

ED Fig. 2a: Why in this experiment up to 4 mM ATP is used? Looks very high to me.

ED Fig. 4f: If the authors have a gel with more separated bands they should use it instead.

ED Fig. 5d: A movie might help the visualisation (might, it is not a requirement). This figure seem to suggest that a single mutant of W419 could be made and characterised to strengthen the

mechanism.

ED Fig. 10d: Unclear to me. I do not understand the point the authors want to make.

Referee #3 (Remarks to the Author):

The manuscript from Dorothee Hern's group is a tour-de-force work from protein sequence mining to the biochemistry and structural biology of ancient KaiBC proteins. The breadth and depth of work are unparalleled, contributing to the fundamental mechanistic question in biological timing – what do these circadian (in this case, Kai) proteins do before they came together to produce temperature-compensated circadian oscillators?

The biochemical experiments along with various deletions and mutations examining autophosphorylation of KaiC and the role of KaiB in dephosphorylation are meticulously carried out. In addition, x-ray crystal structure and CryoEM add volume to understanding the regulation of phosphorylation and the phosphorylation-dependent changes in interactions. These are seminal contributions.

My major issue with the manuscript is the interpretation and the title. I will go over it in the next few paragraphs.

The authors argue that the strength of the manuscript lies in the result interpretation that the KaiBC system in R.s. is a primordial clock and is a precursor to the self-sustaining KaiABC-based circadian oscillator system. The authors claim the self-sustained circadian gene expression profile in RS may not be due to KaiBC, because there is no definitive genetic experiment knocking out the Kai component or biochemical explanation, yet in the very following sentence, they argue their KaiBC system may explain a rudimentary hourglass like the behavior of the organism. The authors don't present any genetic data, and their biochemical explanation is insufficient.

The authors mostly present a series of in vitro experiments with wildly varying conditions (temperature, ATP conc., protein conc.) that demonstrate the KaiBC system shows some aspects of phosphorylation-dephosphorylation that are not self-sustaining and yet can be modulated by an abrupt change in ATP concentration.

The kinase activity of KaiC-RS is almost an order of magnitude higher than that of KaiC-SE and is not temperature compensated. Therefore, it is possible that at a certain stoichiometry of KaiBC complex, ATP-ADP ratio, and temperature, this kinetics can be slow enough for an unstable or ~12h rhythm. However, the authors don't make enough attempts to address this fundamental aspect of the KaiBC primordial clock.

Previous reports have shown the R.s. has a primitive circadian clock mechanism. This clock can be entrained to 24 h light-dark and temperature cycles. Upon release to constant darkness and constant temperature, a self-sustained 24 h rhythm in gene expression under aerobic condition and a 10-12h rhythm in gene expression under anaerobic condition is observed. Hence, a clock in this species is likely to respond to changes that mimic natural conditions, such as temperature, light, and the resultant changes in the intracellular biochemical environment. Therefore, to illustrate that the KaiBC system in Rs underlies a primitive clock, at least two conditions should be met – the phosphorylation-dephosphorylation cycle should respond to a temperature cycle and be self-sustained – albeit with a non-24h periodicity. The authors argue that the change in ATP concentration may represent such an environmental resetting cue. However, in the absence of any

in vivo measurements of ATP/ADP concentration in the organism under light/dark or high-low temperature conditions, the results are preliminary at best.

The authors show the phosphorylation changes to the KaiBC complex when the ATP levels fall, while there is no experiment on what happens when the ATP-ADP ratio flips to ATP levels begin to rise again. This is a crucial experiment to support the argument that the KaiBC system can respond to changes in ATP concentration in both directions.

In summary, the manuscript is misleading, as is with the current title. The readers expect to see enough experimental evidence that the KaiBC system in R.s. is a primordial clock or hourglass that can respond to light:dark, high:low temperature, or ATP-ADP cycle in both directions. At least such a set of experimental data, along with the extensive structure-function studies, will pave the way for future experiments to “evolve” the KaiBC-RS to KaiABC-S.E. clock.

Minor comments.

The authors acknowledge this phosphorylation-dephosphorylation is also observed in the KaiBC-SE system. However, the authors claim those reports are controversial without explaining why these past reports were controversial and how their experiments rise above those controversies.

Ext Figure 1. This is an interesting phylogenetic tree with a lot of information. But unfortunately, the font size is less than 1. I hope the authors provide a high-resolution figure and the journal enables downloading such a high res figure. In addition, it will be useful to provide a table of accession numbers that were used in this phylogenetic tree.

Extended Figure 3 should be incorporated into the main figure 1 or 2.

Ext Figure 6. The author state the nighttime ATP to ADP ratio is 25% ATP but does not cite how they measured the nighttime total ATP+ADP and percentage of ATP in vivo. This information is critical. Ideally, this should be measured in the lab from the same strain from where the KaiB and KaiC were cloned for this study.

Extended figure 7 beautifully shows the potential ATP synthase mechanism underlying dephosphorylation of KaiC. Part of the Ext Figure 7 (a and b) should be accommodated in the main figures.

There are some figures where mean and SEM are shown, but the replicate number is only 2. I don't see the utility of presenting the mean and SEM for these experiments.

Author Rebuttals to Initial Comments:

Referee #1:

Remarks to the Author:

This manuscript presents structures and biochemical studies of clock protein homologs KaiB and KaiC from the purple, non-sulfur photosynthetic proteobacterium *Rhodobacter sphaeroides* (RS). Unlike the KaiABC system from the cyanobacterium *S. elongatus*, which creates a self-sustained and temperature compensated oscillator in vitro, the RS KaiBC proteins from exhibit changes in their structure and activity that may allow them to create a less robust, hourglass-like timer in the absence of KaiA. There are several key differences in the RS Kai proteins, including a coiled-coil extension of the KaiC C-terminus that allows for the KaiC hexamer to form a homododecamer. Intriguingly, phosphomimetic mutations in the KaiC CII domain lead to a large register shift in the coiled coil complex that may be linked to promotion of autodephosphorylation via an allosteric network leading to the CII active site, as deletion of the coiled coils stabilizes phosphoKaiC. Differences in the structure of the A-loop, located near the phosphorylation sites in KaiC, likely lead to an increased basal rate of nucleotide exchange and phosphorylation compared to *S. elongatus* KaiC. KaiB binding enhances ATPase activity at the KaiC CI and CII domains, the latter of which promotes dephosphorylation of the KaiC CII domain independent of nucleotide exchange. Consistent with this, changes in the ATP/ADP ratio serve as an environmental cue in this system, as has been previously demonstrated for the cyanobacterial clock. The experiments here are rigorously performed and analyzed. Overall, this study represents an exciting step forward in discerning the possible mechanism of what could be an ATP/ADP-sensitive hourglass clock. However, the addition of one or two experiments could provide a missing link uniting several of the observations that appear to be unique to the RS system here and establish that it does indeed form an hourglass-like oscillator.

We are excited about this excellent summary of the findings and impact, and the reviewer's strong endorsement. We thank the reviewer for this scientific thorough review leading to superb suggestions that definitely made the manuscript more impactful and accessible to the broad readership (see below).

Major comments:

1. The authors state that the proposal that the kaiBC gene cluster is responsible for circadian gene expression in *R. sphaeroides* is controversial as no decisive experimental evidence (e.g., kaiC knockout strain or in vitro experiments) has been presented in prior publications. Unfortunately, these data aren't presented here either. Although the authors present compelling evidence of an "oscillatory-like pattern" in the presence of KaiB in vitro (Fig. 3), their conclusion that the KaiBCRS system is sufficient to make an hourglass-like timer would be strengthened by either (a) an in vitro assay with 12-hr buffer exchanges into day or night-like ATP/ADP ratios for at least 2 full cycles to demonstrate bona fide oscillations in KaiC phosphorylation or KaiB recruitment or (b) genetic deletion of KaiB and/or KaiC followed by rhythm assays in *R. sphaeroides*.

We thank the referee for this suggestion. We had intended to show the "hourglass, oscillatory-like pattern" by our data initially presented in Fig. 3b,c (i.e., one full 24-hour

oscillation plus the resetting for the next cycle with the change in ATP-to-ADP ratio). However, the suggested experiment in (a) is indeed powerful, and we therefore performed this more direct *in vitro* assay employing dialysis to change and maintain the ATP-to-ADP ratios as present during day and night, and acquired time points for 48 hours. Protein stability becomes an issue at longer time points thus we were only able to extend the data points up to 36 hours. However, our new experiments now clearly demonstrate that KaiBC_{RS} is required and sufficient for an hourglass-like timer: First, during day time, with high ATP concentrations, fast and efficient phosphorylation is observed. Second, under night conditions (25% ATP), KaiB_{RS} is needed for the dephosphorylation of KaiC_{RS} and results in a much higher fraction of the dephosphorylated state than in the absence of KaiB_{RS} (*cf.* Fig. 3c; around 20- to 24-hour time points). Third, when the ATP-to-ADP ratio is flipped again to mimic the daytime, KaiC_{RS} is able to phosphorylate again (*cf.* Fig. 3c; around the 28-hour mark). We replaced the panel of Fig. 3c with our newly obtained data from the 36-hour experiment where the ATP-to-ADP ratio was changed every 12 hours to mimic the day- and night-time using buffer dialysis, following the excellent experimental suggestion by the reviewer.

2. The authors have implicated both changes in the coiled coil architecture of KaiC in the S413E/S414E phosphomimetic mutant and KaiB binding to the KaiC CI domain in promoting KaiC autodephosphorylation at the CII domain—does this activity of KaiB depend on the presence of the coiled coils? Identifying if/how they collaborate to regulate KaiC phosphorylation state would help to further develop the mechanism highlighted in Fig. 4g.

Indeed, we show that changes in the coiled-coil domain and binding of KaiB_{RS} accelerate the dephosphorylation in KaiC_{RS} (*cf.* Figs. 2f and 3c, respectively). Whether or not there is *synergy* between both regulatory mechanisms is indeed a very interesting question, but unfortunately intractable to assess experimentally for the following reason: KaiB_{RS} binding to the C-terminal truncated version (KaiC_{RS}- Δ coil) results in dissociation of the hexameric complex into monomeric KaiC_{RS} which is considerably less stable (*cf.* Extended Data Figure 10b). More importantly, monomeric KaiC_{RS} does not contain a nucleotide-binding site (which is in the interface between two monomers) and thus cannot autodephosphorylate as the mechanism requires the P-transfer from the protein to the bound ADP (see Figures 3e and 4g). We note that we edited Fig. 4g to illustrate the proposed the mechanism of KaiBC_{RS} better.

Minor comments

1. The following statement in the abstract is a bit confusing, as it could be interpreted that the cyanobacterial KaiBC proteins could give rise to an hourglass timer. One suggested is: “The canonical KaiABC system in cyanobacteria is well understood, but little is known about more ancient systems [that possess just KaiBC], except for controversial reports that [they] exhibit a basic, hourglass-like timekeeping mechanism.”

Agreed - we appreciate your suggestion and have changed the sentence to: “The canonical KaiABC system in cyanobacteria is well understood, but little is known about more ancient systems that possess just KaiBC, except for reports that they might exhibit a basic, hourglass-like timekeeping mechanism.”

2. The statement on line 49 that "... in prokaryotes the circadian rhythm is regulated by only three proteins (kaiA, kaiB, and kaiC)" is not quite true—there is ample evidence that other proteins regulate cyanobacterial rhythms in addition to the Kai proteins (e.g., SasA, CikA, LdpA, Pex, KidA).

Correct, that sentence is indeed ambiguous and to clarify we have changed the text as follows: "In eukaryotes these systems are complex and very sophisticated, whereas in prokaryotes the core mechanism is regulated by a posttranslational oscillator that can be reconstituted *in vitro* with three proteins (kaiA, kaiB, and kaiC) and ATP."

3. The statement "The first obvious question is how KaiCRS and other members in the clade can auto-phosphorylate despite having no KaiA, which is known to be crucial for this function in the canonical KaiABC system" is true but a little bit misleading—KaiA is needed for oscillations in KaiC phosphorylation, but KaiC can autophosphorylate on its own, and this activity is preferentially enhanced (over autodephosphorylation) by decreasing temperatures (e.g., 4 °C). At 30 °C, as in Fig. 1b, autodephosphorylation of KaiCSE is favored (Nishiwaki et al. 2007, EMBO J.). The statement on lines 167-170 could also be edited a bit to reflect these data better.

Yes, this is correct! We changed the text to specify that KaiA is needed for autophosphorylation at the optimum temperature for cyanobacteria.

4. Typo on line 213: phopsphorylated.

Thank you for catching that mistake – corrected now.

5. It is quite interesting that KaiBRS is a monomer in solution, unlike the tetrameric KaiB found in KaiABC systems. Is it known if the KaiBRS monomer takes on the "fold-switch" conformation in the absence of KaiC?

Another excellent question, and this is indeed the project of my new NMR postdoc and new graduate student to characterize this equilibrium in detail.

[This has been redacted]

6. A reference to Snijder et al. (2014, PNAS) should be added to line 258, as they were the first to demonstrate cooperativity in KaiB binding to the KaiC_{SE} hexamer.

Thank you and we apologize for the oversight: the suggested reference has been added.

7. It was previously demonstrated, not implied, that ATPase activity (or the post-hydrolysis state) in the KaiC CI domain is critical for KaiB binding (Phong et al., 2012, PNAS).

Thank you for pointing that out, we changed the wording accordingly.

Referee #2:

Remarks to the Author:

A. The paper "From primordial clocks to circadian oscillators" by Pitsawong, Padua et al. includes a wealth of biochemical and biophysical data to try shading light into the mechanism that makes KaiCB an hour glass in those organisms that lacks KaiA. In short the authors describe structural and regulatory mechanisms of a primitive circadian clock in a fairly exhaustive way. The themes/questions that this article addresses, the definition of circadian systems, hour-glass biological mechanism and its evolution, are of general interest for a broad audience of readers. Moreover the molecular details addressed in the article are very relevant for any structural biologist interested in the molecular switches that determine fine regulation and auto regulation in particular.

We thank the reviewer for the enthusiasm/endorsement of our manuscript, and pointing out its relevance to the field of structural biology in understanding regulation. Importantly, we appreciate the thorough review and outstanding suggestions for revisions that further improved our manuscript!

In the primitive KaiBC system from *Rhodospirillum rubrum* spheroides the authors identify the presence of a large KaiC C-terminal extension that fold into coiled-coil domains and hold together two KaiC hexamers and confer stability. The authors claim that the presence of these long coiled-coils and their positioning (different in the phosphorylated vs the non-phosphorylated form) defines a long-range allosteric network that correlates with auto-phospho and dephosphorylation. This is only partially convincing for me as - the comparisons of the different structures is not clear from the text/ what is X-ray what cryo-EM in terms of oligomeric states/what is crystal packing what in solution dodecameric structure/deletion of the coiled-coil is a bit rough-what about point mutations or deletion of only the last 30aa?

We agree with the referee that our findings regarding the long-range allosteric network mediated by the coiled-coil domain are an integral part of the manuscript and that a careful comparison of the different structures is indeed important. We apologize if our original manuscript was not written clearly enough to convey that we do know the oligomeric states in solution for all states discussed in this manuscript based on size-exclusion chromatography and analytical ultracentrifugation (see Extended Data Fig. 3d-f; information now also added to Extended Data Tables 1 and 2). We have revisited our text and (Extended Data) figures to clarify these points in our revised manuscript.

Regarding the choice of the KaiC_{RS-Δcoil} construct: our choice for the location of the stop codon was guided by (i) the phylogenetic tree (see Extended Data Fig. 1) and (ii) the residues that are visible for the C-terminus in the crystal structure of KaiC_{SE}. In retrospect that turned out to be an appropriate choice as Glu⁴⁹⁰ is near the beginning of the coiled-coil domain; deleting only the last 30 residues seems an arbitrary choice and we think that it would likely not provide additional insight compared to the KaiC_{RS-Δcoil} construct.

Having shown that the KaiC from RS is able of both auto phospho and dephosphorilation the authors go on showing how the ATP-ADP ratios characteristic of the day and night status modulates the phosphorylation of KaiC. Moreover, they go on investigating the

mechanism by which KaiB helps maintaining a night de-phosphorylated state. They consider 3 hypothesis and thoroughly test them (nice and clear ED Figs 7 and 8) thus finding a mechanism of phosphate transfer, similar to the ATP-synthase, followed by KaiB-accelerated ATP hydrolysis. I find this part of the results very interesting and solid. Finally the authors also solve the structure of the Kai-BC complex and suggests a mechanism for increased ATPase activity via higher solvent accessibility in the complex.

B. The work, to my knowledge, is quite novel as it presents a primordial circadian clock made only of KaiBC and thus giving information about basis of cyclic systems. I am not an expert in the field, but, for what I can see, the authors referenced well the existing literature, mainly based on KaiABC, including the studies available on pre-print.

C. This work includes a constant interplay and corroboration steps between biochemical and structural results, which makes it solid. However, while it is really impressive that the authors determined so many different structures, I think that their analysis should be more thorough and better presented in the manuscript to make a really strong structural contribution to the work. For example, one or two extended figures describing the cryo-EM procedures and images of the 2D classes and the maps obtained should be prepared. We thank the referee for the suggestion and added Extended Data Fig. 4 to provide this additional information on the cryo-EM data processing and analysis.

The comparison between the unphosphorylated (U) and phosphomimic seems to be done between the crystal structure (dodecamer?) with the cryo-EM map phosphomimic. However, is the crystal structure a dodecamer in the asymmetric unit or in the crystal packing? Information about the oligomeric state could be added to the X-ray and cryo-EM tables.

Yes, this is correct: the crystal structure of wild-type KaiCRs was solved in P1 and contains the entire dodecamer in the asymmetric unit. We have added the information regarding the oligomeric state of all structures to the Extended Data Tables 1 and 2. We share the reviewer's concern about a possible effect on the oligomeric states in a crystal. Therefore, we directly measured the oligomeric states in solution by size-exclusion chromatography and analytical ultra-centrifugation (data in Extended Data Fig. 3a-c). Our results clearly demonstrate that both the unphosphorylated (U) and phosphomimetic KaiC are dodecameric in solution and is not an artifact caused by crystal packing or EM sample preparation. Hence, we can directly compare two the structures that were solved by cryo-EM and X-ray crystallography.

In the material and methods section the authors explain that the cryo-EM dodecameric structures were actually obtained upon combination of more refined C6 symmetries hexamers because the flexibility of the coiled-coil mediated C-terminal interaction limited the attainable resolution of the dodecameric structure. Now several questions arise:

1. What happens if and no symmetry is applied, even at the level of hexamer analysis? How does a C1 reconstruction of KaiCRs compare with the recent reconstruction by Swan et al. (<https://www.biorxiv.org/content/10.1101/2021.09.14.460370v1.full>)? Any

interesting insights about the ATP pockets in C1 reconstructions of the U versus phosphomimic?

We agree that this is an interesting question. We have in the past tried to calculate C1 reconstructions, starting from the ab-initio 3D step, and found no differences in structure apart from lower resolution. Based on the manuscript by Swan *et al.* we also tried relaxing the structure to C2 and once again found no difference. We have now added the following text into the Methods section:

“In an attempt to find deviations from C6 symmetry we also calculated reconstructions for both structures assuming C1 symmetry, starting from the ab-initio 3D step. The resulting refined structures did not exhibit detectable departures from C6 symmetry, we thus present symmetrized volumes as our final result.”

2. The authors say that “the hexamer reconstruction were overlaid on top of each other and combined by taking the voxel with the maximum absolute value from the two overlaid copies”. How was this done? Did they use the max voxel procedure in chimera? Or with map combination using Phenix? In any case, a combination a posteriori of the two hexamers might make the analysis of the specific changes in the coiled-coil shift of the phosphorylated (cryo-EM with hexamer combination) vs the unphosphorylated (X-ray crystal packing?) states less strong. It is therefore absolutely required that the authors show in supplementary data the look of the naturally occurring dodecameric structures even if at low resolution (how low? How interpretable at the helical level?).

The hexamer reconstructions were combined using a custom program which simply takes the maximum absolute value - this is equivalent to the max voxel procedure implemented in UCSF Chimera, only in this case the voxel with the max absolute value is taken. We fully agree with the reviewer to process the structures as the full dodecameric unit! Thank you for this suggestion. We have now done this for both EM structures and obtained ~3.3 Å resolution for both. Notably, these structures match our combined structure to the achieved resolution. We have added a figure demonstrating these reconstructions (Extended Data Fig. 4).

3. Even at low resolution what about scanning the conformational spaces using cryoDRG or 3D variability for the dodecameric assemblies? Then I would expect that multiple possible variants with the two hexamers oriented differently compared to one another should be done. This could be very interesting especially by looking at the effect of different bending of the coiled-coils on the A-loops. In summary, further structural analysis or at least a better presentation of it in the extended and supplementary materials would strengthen the results about the allosteric network at the moment only validated by coiled-coiled deletion. I would argue that a full deletion is a harsh thing to do and that some point-mutant analysis should be done as complement or at least the deletion of only the last 30 aa.

Thank you for this suggestion, and yes, this would have been indeed interesting if we would have seen lower resolution for the dodecamer reconstruction. However, our analysis of dodecamers yielded ~3.5-Å resolution reconstructions, indicating the absence of large structural variabilities for which cryoDRGN was designed. Due to this, and the fact that symmetry relaxation revealed no differences, we think that this analysis will not

reveal additional insight into the mechanism of KaiC and so have decided not to perform this experiment at this time.

4. An additional clarification is required: the crystal structures were used instead of an ab-initio? So what happens if an ab initio is generated from the by splitted hexamers with the coiled-coil most likely "floating" around or less resolved?

For all the EM maps shown, including C6 and D6 reconstructions, the *cis*TEM ab initio procedure was used to generate a model without any external reference. The C6 ab initio reconstructions are now included in the new workflow panels (Extended Data Fig. 4) to clarify our workflow.

D. I have partially answered this in the section above regarding the cryo-EM data. Would be important to have a workflow with numbers of particles and pictures of the different structured obtained and how.

Agreed, and this information is now included in Extended Data Fig. 4 describing the processing.

Regarding the biochemical assays, I noticed that often there are no multi points or sigma bars nor is stated in the figure captions how many measurements were performed. I assume that at least triplicates were done for each experiment presented, but this must be stated each time.

Thank you for pointing out; indeed, most experiments were performed at least in triplicate, and we have added this information in the figure legends and/or Methods section. Exceptions are the SDS-page gels and radiometric experiments to follow the phosphorylation state of KaiC_{RS}. The experiments have been performed several times and give consistent results; it is, however, not doable to perform exact replicates (and thus show them in the same figure as "mean \pm SD") because it is impossible to prepare the sample in a manner to give identical initial phosphorylation states.

E/F. Overall I think that this work is fairly robust and interesting, I would however required the revisions/experiment/presentation of data suggested in sections C and D.

G. To my knowledge the authors take onto consideration the existing literature.

H. The title does not really represent a condensation of the findings described in the manuscript. The abstract is clear. The "temperature factor", i mean the reference to the temperature compensation aspect is mentioned here and there in the text. I find it confusing, and I do not understand its relevance. It might be more obvious for people in the field. Some reference could be added in the introduction such as after "mechanism that is different from the widely studied circadian oscillator in *Synechococcus elongatus* PCC 7942" or after the referral to the dynein coiled-coil regulation.

Thank you for pointing out that this point is not well enough described for the broader readership. Temperature compensation (e.g., for the ATPase activity) is one of the requirements for a true circadian oscillator and has been demonstrated for KaiC_{SE}. In contrast, the KaiC_{RS} system described here shows a temperature dependence, suggesting that this system is likely not a true circadian oscillator. We have now added a

few references to provide some additional context for readers less familiar with the field as suggested by the referee.

The text could be clearer for readers outside the field in the part where the extended A-loop conformation is explained. What is the importance of the 422-loop? The role of the 422-loop has been described before by Egli *et al.*¹. We added a sentence paraphrasing their explanation: “The loss of interaction between the A-loop and 422-loop (just 10 residues apart from the phosphorylation sites), results in closer proximity between the hydroxyl group of Ser⁴³¹/Thr⁴³² and the γ -phosphate of ATP, thereby, facilitating the phosphoryl-transfer step.”

Line 185: When defining the ATP:ADP ratio there is a typo.

We did not intend to define an ATP-to-ADP ratio but rather indicated the amount of ATP present during day- and night-time, respectively. That sentence was indeed a bit confusing, and thus rephrased it to describe the experimental setup more clearly – thank you for pointing this out.

The regulatory role of the KaiB is very well explained in the ED Figs 7 and 8, while in the main text is not as clear, more specifically the conclusions are not spelt out clearly. In the Discussion, this sentence “The dodecameric KaiCRS shows constitutive kinase-activity due to its extended C-terminal tail that forms a coiled-coil bundle with the opposing hexamer, and auto-phosphorylation occurs within half an hour” should be tuned down as stated as it is now it almost says that the coiled-coil itself has kinase activity.

We expanded Fig. 4g to illustrate more precisely the role of the coiled-coil domain and KaiB_{RS} binding in regulating the phosphorylation state of KaiC_{RS}. We adapted the sentence as follows: “The dodecameric KaiC_{RS} shows constitutive kinase-activity due to its extended C terminal tail that forms a coiled-coil bundle with the opposing hexamer *and elicits a conformation akin to the exposed A-loop conformation in KaiAC_{SE}*, and auto-phosphorylation occurs within half an hour.”

The comparison with the KaiABC system is very clear. The conclusions about the role of the coiled-coil in the allosteric network is not very obvious and convincing to me (from the data), even though likely and tempting to imagine. I would leave it more as an open question that requires further studies. Probably further analysis of the dodecamers in the cryo-EM datasets could give more insights as well as point mutations e.g. for W419. Maybe, in the discussion, a sentence explaining more in details the analogy with the dynein register shift could help.

Thank you for these suggestions, and the question about W419. We edited the main text for the first point, added a movie for better visualization, and added a sentence to the discussion explaining the analogy of the dynein register shift.

We indeed analyzed our data in more detail, and the comparison to the powerful structural findings in: “Coupling of distant ATPase domains in the circadian clock protein KaiC. Swan JA*, Sandate CR*, Chavan A, Freeberg AM, Etwaru D, Ernst DC, Palacios JG, Golden SS, LiWang A, Lander GC, Partch CL, *Nat Struct Mol Biol* 29: 759-766”

strengthens our hypothesis and the prediction that a W419 mutation would be detrimental to the protein:

Our hypothesis that KaiC_{RS} (residues in bold, dephosphorylated state in green, and phosphorylated in blue) uses the allosteric network to relay information between opposite monomers in the dodecamer is strengthened by the fact that KaiC_{SE} uses the same spot to convey structural rearrangements between monomers in the hexamer for its different states (residues in italics; black: night-time compressed PDB 7S65; gray: night-time extended PDB 7S66; white: day-time extended PDB 7S67). The interplay between L438 and W419 in KaiC_{RS} resembles that of residues F419 and I425 in KaiC_{SE}.

Comments on figures:

In many figures the spatial order of the panels is strange and not always the same. I would expect to have always the b panel following a on the top right not bottom left.

The spatial ordering of the panels is chosen as to make most efficient use of the available space. The layout of Figures 1 and 2 indeed do not completely follow the expectations of the referee (the ordering in Figure 3 has now changed with the replacement of panel C); we have opted to keep our proposed layout for now, but defer to the editor for guidance on this point in regard to space usage.

Fig. 1e: It would be good to indicate the 422-loop also in the zoomed view. Moreover, not all KaiC seem to have the same 422-loop.

Good suggestion, we added the 422-loop for the KaiC_{SE} structures in the zoom-in together with the corresponding residues of KaiC_{RS}.

Fig. 2a: Are the colours chosen ok for color-blind people?

We thank reviewer for the reminder: we have adapted the colors to be more colorblind-friendly and additionally introduced different markers, fill-textures, and labels to aid in the visualization in the revisions.

Fig. 2b: I find this panel a bit confusing. I think it would be better to show one picture as it

is now in the left but the details of the phosphorylated and unphosphorylated forms separately.

Agreed, we have changed this figure panel as suggested, and it now much easier to see that the register shift is achieved by changing the helix partner, and which amino acids are present in the interfaces.

Fig. 3a: Please state here all the meaning of the acronyms U, S, D used in the figure. Thank you for pointing out this omission – we have added it to the figure legend.

Fig. 3d: Here is the only time where it is stated that two replicates were done. Should not be three replicates for all the biochemical experiments?

We have included more experimental data for this experiment and added the information regarding replicates (for this experiment and elsewhere) in the figure legends and/or Methods section.

Fig. 3f: Unclear; where are the point values coming from?

We edited the legend to specify the source of experiment: “The decay of phosphorylated ^{32}P -KaiC_{RS} bound with 4 mM ADP in the absence (gray circles) and presence (red diamonds) of KaiB_{RS} at 30 °C is obtained from autoradiography quantification (see Extended Data Fig. 7).

Fig. 4g: Would be good to have here also an additional summarizing schematic showing all that is happening in CI CII and the coiled coils upon binding of KaiB and with ATP/ADP ratios changing.

We thank reviewer for this suggestion and have extended Fig. 4g to provide a comprehensive overview on what is happening in the CI and CII domains during day- and nighttime, together with the regulatory roles of the coiled-coil domain and binding of KaiB_{RS}.

ED Fig. 2a: Why in this experiment up to 4 mM ATP is used? Looks very high to me.

We used the same ATP concentration as was described in the study of KaiC_{SE} (reference ²).

ED Fig. 4f: If the authors have a gel with more separated bands they should use it instead.

We repeated the experiment for the revision and included the resulting gel that shows slightly better separation between the unphosphorylated- and singly-phosphorylated states. We note that it is intrinsically more difficult to obtain a clear separation of bands for samples that start with a roughly 1:1 mixture of unphosphorylated- and Ser⁴¹³-phosphorylated KaiC_{RS} compared to gels with a more equal distribution over three phosphorylation states (*cf.* ED Fig. 2e).

ED Fig. 5d: A movie might help the visualisation (might, it is not a requirement). This figure seems to suggest that a single mutant of W419 could be made and characterised to strengthen the mechanism.

Nice suggestion, we added such a movie to help visualization to the supplemental material. For the W419 mutation see Figure 1 above.

ED Fig. 10d: Unclear to me. I do not understand the point the authors want to make. This panel was another visualization of the thermal denaturation data shown in ED Fig. 10c; however, we agree with the referee that this panel was redundant and thus removed it.

Referee #3

(Remarks to the Author):

The manuscript from Dorothee Kern's group is a tour-de-force work from protein sequence mining to the biochemistry and structural biology of ancient KaiBC proteins. The breadth and depth of work are unparalleled, contributing to the fundamental mechanistic question in biological timing – what do these circadian (in this case, Kai) proteins do before they came together to produce temperature-compensated circadian oscillators?

The biochemical experiments along with various deletions and mutations examining autophosphorylation of KaiC and the role of KaiB in dephosphorylation are meticulously carried out. In addition, x-ray crystal structure and CryoEM add volume to understanding the regulation of phosphorylation and the phosphorylation-dependent changes in interactions. These are seminal contributions.

What a beautiful endorsement of the manuscript, our joint forces between biochemistry and structural biology, and a compelling description of the *importance of biological timing* in general. Thank you for your thorough review and excellent suggestions below for additional experiments and editing the text and figures for the revisions! They indeed made the revised manuscript stronger.

My major issue with the manuscript is the interpretation and the title. I will go over it in the next few paragraphs.

The authors argue that the strength of the manuscript lies in the result interpretation that the KaiBC system in R.s. is a primordial clock and is a precursor to the self-sustaining KaiABC-based circadian oscillator system. The authors claim the self-sustained circadian gene expression profile in RS may not be due to KaiBC, because there is no definitive genetic experiment knocking out the Kai component or biochemical explanation, yet in the very following sentence, they argue their KaiBC system may explain a rudimentary hourglass like the behavior of the organism. The authors don't present any genetic data, and their biochemical explanation is insufficient.

The authors mostly present a series of in vitro experiments with wildly varying conditions (temperature, ATP conc., protein conc.) that demonstrate the KaiBC system shows some aspects of phosphorylation-dephosphorylation that are not self-sustaining and yet can be modulated by an abrupt change in ATP concentration.

The kinase activity of KaiC-RS is almost an order of magnitude higher than that of KaiC-SE and is not temperature compensated. Therefore, it is possible that at a certain stoichiometry of KaiBC complex, ATP-ADP ratio, and temperature, this kinetics can be slow enough for an unstable or ~12h rhythm. However, the authors don't make enough attempts to address this fundamental aspect of the KaiBC primordial clock.

Previous reports have shown the R.s. has a primitive circadian clock mechanism. This clock can be entrained to 24 h light-dark and temperature cycles. Upon release to constant darkness and constant temperature, a self-sustained 24 h rhythm in gene expression under aerobic condition and a 10-12h rhythm in gene expression under anaerobic condition is observed. Hence, a clock in this species is likely to respond to changes that mimic natural conditions, such as temperature, light, and the resultant changes in the intracellular biochemical environment. Therefore, to illustrate that the KaiBC system in Rs underlies a primitive clock, at least two conditions should be met – the phosphorylation-dephosphorylation cycle should respond to a temperature cycle and be self-sustained – albeit with a non-24h periodicity. The authors argue that the change in ATP concentration may represent such an environmental resetting cue. However, in the absence of any *in vivo* measurements of ATP/ADP concentration in the organism under light/dark or high-low temperature conditions, the results are preliminary at best.

The authors show the phosphorylation changes to the KaiBC complex when the ATP levels fall, while there is no experiment on what happens when the ATP-ADP ratio flips to ATP levels begin to rise again. This is a crucial experiment to support the argument that the KaiBC system can respond to changes in ATP concentration in both directions.

In summary, the manuscript is misleading, as is with the current title. The readers expect to see enough experimental evidence that the KaiBC system in R.s. is a primordial clock or hourglass that can respond to light:dark, high:low temperature, or ATP-ADP cycle in both directions. At least such a set of experimental data, along with the extensive structure-function studies, will pave the way for future experiments to “evolve” the KaiBC-RS to KaiABC-S.E. clock.

The paragraph above summarizes very well what has been reported in the literature for *Rhodobacter sphaeroides*. The proposed *in vitro* experiment following a full 48-hours rhythm while mimicking the ATP-to-ADP ratios at day- and nighttime ratios (in both directions) compared to our original data in Fig. 3c was a great suggestion. We performed the new experiments suggested by this reviewer (as well as reviewer 1) and acquired time points for 48 hours. Protein stability becomes an issue at longer time points thus we were able to extend the data points up to 36 hours. However, our new experiments now clearly demonstrate that KaiBC_{RS} is required and sufficient for an hourglass-like timer (see new Fig. 3c): First, during day time, with high ATP concentrations, fast and efficient phosphorylation is observed. Second, under night conditions (25% ATP), KaiB_{RS} is needed for the dephosphorylation of KaiC_{RS} and results in a much higher fraction of the dephosphorylated state than in the absence of KaiB_{RS} (*cf.* Fig. 3c; around 20- to 24-hour time points). Third, when the ATP-to-ADP ratio is flipped again to mimic the daytime, KaiC_{RS} is able to phosphorylate again (*cf.* Fig. 3c; around the 28-hour mark). We were therefore able to measure the ATP/ADP cycle in both directions.

As suggested by the reviewer, we had already used ATP-to-ADP ratios that roughly correspond to *in vivo* concentrations in the organism (see our response to your question on this topic below), but now changed and maintained these ratios mimicking day and night by means of dialysis in the new set of experiments. We thank the reviewer for the suggestion, as the new experimental data strengthens our conclusion that the switch in ATP-to-ADP ratio between day and night is indeed an environmental cue that regulates the phosphorylation state of KaiC_{RS} and is involved in resetting the clock.

Evolution of a more sophisticated, sustained circadian rhythm by the addition of KaiA in, for example, *Synechococcus elongatus* made the system less dependent of the ATP/ADP ratio as was shown by Rust *et al.*².

Minor comments.

The authors acknowledge this phosphorylation-dephosphorylation is also observed in the KaiBC-SE system. However, the authors claim those reports are controversial without explaining why these past reports were controversial and how their experiments rise above those controversies.

We apologize for causing any confusion and assume that you are referring to the KaiBC-RS system. We have expanded the paragraph in the introduction to clarify this: “Surprisingly, the organism shows sustained rhythms of gene expression *in vivo*, but whether *kaiBC* is responsible for this observation remains inconclusive in the absence of a *kaiC* knockout³. A more recent study of the closely-related *Rhodospseudomonas palustris* demonstrated using a KO strain causality between the proto-circadian rhythm of nitrogen fixation and expression of the *kaiC* gene⁴. We discover using *in vitro* experiments that KaiBC_{RS} is indeed a primordial circadian clock with a mechanism that is different from the widely studied circadian oscillator in *Synechococcus elongatus* PCC 7942 (hereafter referred to as KaiABC_{SE})⁵⁻⁸.”

Ext Figure 1. This is an interesting phylogenetic tree with a lot of information. But unfortunately, the font size is less than 1. I hope the authors provide a high-resolution figure and the journal enables downloading such a high res figure. In addition, it will be useful to provide a table of accession numbers that were used in this phylogenetic tree. We fully agree with the referee that these data are important and should be made available to the community. Due to the sheer number of sequences, it is impossible to display the full tree and maintain readability of the labels. Therefore, we added the alignment- and tree-files that contain all pertinent information, including the accession codes, as Supplementary Datasets 1 and 2. We only referenced those additional files in the Materials and Methods section and the legend of ED. Fig 1; to alert the readers to the existence of these datasets we have now added a reference in the main text as well.

Extended Figure 3 should be incorporated into the main figure 1 or 2.

Thank you for the suggestion, we have included these panels (originally in Extended Data Fig. 3a,b) into Figure 2 of the main text in the revisions (panels a and b).

Ext Figure 6. The authors state the nighttime ATP to ADP ratio is 25% ATP but does not cite how they measured the nighttime total ATP+ADP and percentage of ATP *in vivo*. This information is critical. Ideally, this should be measured in the lab from the same strain from where the KaiB and KaiC were cloned for this study.

This information is indeed critical. Thank you for pointing out to cite where these concentrations were previously measured. The rationale for the ATP-to-ADP ratio at day- and night-time comes from two earlier literature reports. The change in ATP-to-ADP ratio at daytime and nighttime were directly measured *in vivo* in the strain *Rhodobacter*

*sphaeroides*⁹, ATP is between 2.0-2.4 mM during day and drops to 0.5-0.6 during night, and it is well known that the total nucleotide concentration stays constant. We chose the total nucleotide concentration of 4 mM in our in vitro work to be identical to the described in vitro experiments performed for the canonical KaiC_{SE} and because of photosynthesis in daylight virtually all nucleotide is ATP². We note that a slightly higher amount of ATP will not affect our results as the affinity of KaiC_{RS} for ATP is higher than for ADP. We have now added the rationale and references given above to the Methods section.

Extended figure 7 beautifully shows the potential ATP synthase mechanism underlying dephosphorylation of KaiC. Part of the Ext Figure 7 (a and b) should be accommodated in the main figures.

We thank reviewer for suggestion and have incorporated part of Extended Data Fig. 7 into Figure 3 in the main text.

There are some figures where mean and SEM are shown, but the replicate number is only 2. I don't see the utility of presenting the mean and SEM for these experiments.

Thank you for pointing this out – we have added an additional data set for this experiment and all experiments where error bars are shown were performed at least in triplicate.

References

- 1 Egli, M. *et al.* Loop-loop interactions regulate KaiA-stimulated KaiC phosphorylation in the cyanobacterial KaiABC circadian clock. *Biochemistry* **52**, 1208-1220 (2013).
- 2 Rust, M. J., Golden, S. S. & O'Shea, E. K. Light-driven changes in energy metabolism directly entrain the cyanobacterial circadian oscillator. *Science* **331**, 220-223 (2011).
- 3 Min, H., Guo, H. & Xiong, J. Rhythmic gene expression in a purple photosynthetic bacterium, *Rhodobacter sphaeroides*. *FEBS Lett* **579**, 808-812 (2005).
- 4 Ma, P., Mori, T., Zhao, C., Thiel, T. & Johnson, C. H. Evolution of KaiC-Dependent Timekeepers: A Proto-circadian Timing Mechanism Confers Adaptive Fitness in the Purple Bacterium *Rhodospseudomonas palustris*. *PLoS Genet* **12**, e1005922 (2016).
- 5 Cohen, S. E. & Golden, S. S. Circadian Rhythms in Cyanobacteria. *Microbiol Mol Biol Rev* **79**, 373-385 (2015).
- 6 Golden, S. S. Principles of rhythmicity emerging from cyanobacteria. *Eur J Neurosci* **51**, 13-18 (2020).
- 7 Partch, C. L. Orchestration of Circadian Timing by Macromolecular Protein Assemblies. *J Mol Biol* **432**, 3426-3448 (2020).
- 8 Rust, M. J., Markson, J. S., Lane, W. S., Fisher, D. S. & O'Shea, E. K. Ordered phosphorylation governs oscillation of a three-protein circadian clock. *Science* **318**, 809-812 (2007).
- 9 Abee, T., Hellingwerf, K. J. & Konings, W. N. Effects of potassium ions on proton motive force in *Rhodobacter sphaeroides*. *J Bacteriol* **170**, 5647-5653 (1988).

Reviewer Reports on the First Revision:

Referees' comments:

Referee #1 (Remarks to the Author):

I appreciate the thorough response of the authors to all of the reviewer comments. The changes in text and additional experiments and structural analyses presented here have significantly improved this already strong manuscript. In fact, I feel that this is a landmark study in that it provides the first molecular description of a so-called hourglass timer—being able to examine the differences between this timer and a self-sustaining circadian oscillator could help bring a whole new level of insight into biological timekeeping. I think this manuscript and its technical achievements would be of interest to the broad readership of this journal.

Minor comments

1. Reference 19 should be updated to the published paper (Swan, Sandate et al. 2022 NSMB).
2. The summary cartoon in Fig. 4g has very small text that is hard to read.

Referee #2 (Remarks to the Author):

The authors have thoroughly answered all my comments and implemented changes in the text and figures to address them. They have performed the additional data-processing experiments that were asked for and shared all their structural data, which are of high quality. I can see that they have additionally added biochemical experiments that further strengthen their conclusions. The figures have been extensively re-visited and extended data added. In the current form, this manuscript is in my opinion very solid, complete, multidisciplinary, well written and documented.

Referee #3 (Remarks to the Author):

The manuscript has addressed all critical concerns I had raised in the first round of review. The revised manuscript with new data, text and reorganized structure is much stronger in supporting the central message and will make a seminal contribution to the field.

-Satchin Panda

Author Rebuttals to First Revision:

Referees' comments:

Referee #1 (Remarks to the Author):

I appreciate the thorough response of the authors to all of the reviewer comments. The changes in text and additional experiments and structural analyses presented here have significantly improved this already strong manuscript. In fact, I feel that this is a landmark study in that it provides the first molecular description of a so-called hourglass timer—being able to examine the differences between this timer and a self-sustaining circadian oscillator could help bring a whole new level of insight into biological timekeeping. I think this manuscript and its technical achievements would be of interest to the broad readership of this journal.

Minor comments

1. Reference 19 should be updated to the published paper (Swan, Sandate et al. 2022 NSMB).
2. The summary cartoon in Fig. 4g has very small text that is hard to read.

Thank you for pointing this out! We have updated the reference to the published paper and reorganized the text in Fig. 4g to improve the readability.

Referee #2 (Remarks to the Author):

The authors have thoroughly answered all my comments and implemented changes in the text and figures to address them. They have performed the additional data-processing experiments that were asked for and shared all their structural data, which are of high quality. I can see that they have additionally added biochemical experiments that further strengthen their conclusions. The figures have been extensively re-visited and extended data added. In the current form, this manuscript is in my opinion very solid, complete, multidisciplinary, well written and documented.

Referee #3 (Remarks to the Author):

The manuscript has addressed all critical concerns I had raised in the first round of review. The revised manuscript with new data, text and reorganized structure is much stronger in supporting the central message and will make a seminal contribution to the field.

-Satchin Panda